# Mitochondrial DNA mosaicism in normal human somatic cells

Jisong An [1], Chang Hyun Nam[1], Ryul Kim[1,2], Yunah Lee [1], Hyein Won[1], Seongyeol Park[1,2], Won Hee Lee[1], Hansol Park [1,2], Christopher J. Yoon [1,3], Yohan An [1], Jie-Hyun Kim[4], Jong Kwan Jun[5], Jeong Mo Bae[6], Eui-Cheol Shin [1], Bun Kim [7], Yong Jun Cha[7], Hyun Woo Kwon [8], Ji Won Oh[9], Jee Yoon Park[10,11], Min Jung Kim[12] & Young Seok Ju [1,2] ✉

Somatic cells accumulate genomic alterations with age; however, our understanding of mitochondrial DNA (mtDNA) mosaicism remains limited. Here we investigated the genomes of 2,096 clones derived from three cell types across 31 donors, identifying 6,451 mtDNA variants with heteroplasmy levels of ≥0.3%. While the majority of these variants were unique to individual clones, suggesting stochastic acquisition with age, 409 variants (6%) were shared across multiple embryonic lineages, indicating their origin from heteroplasmy in fertilized eggs. The mutational spectrum exhibited replication-strand bias, implicating mtDNA replication as a major mutational process. We evaluated the mtDNA mutation rate ($5.0 \times 10^{-8}$ per base pair) and a turnover frequency of 10–20 per year, which are fundamental components shaping the landscape of mtDNA mosaicism over a lifetime. The expansion of mtDNA-truncating mutations toward homoplasmy was substantially suppressed. Our findings provide comprehensive insights into the origins, dynamics and functional consequences of mtDNA mosaicism in human somatic cells.

Genomic alterations accumulate in somatic cells throughout an individual's lifetime[1–5]. Recent sequencing studies have documented mutations in the nuclear genome and frequent clonal competition of normal cells carrying mutations[6–11]. However, the landscape of mitochondrial DNA (mtDNA) mosaicism in normal human tissues remains unexplored.

Mitochondria are organelles involved in energy metabolism, cell signaling, apoptosis and biosynthesis[12–16], carrying their own 16.6 kb-long, circular DNA[17]. mtDNA mutations can be acquired somatically during development and aging[18–22], shaping the genetic mosaicism in somatic tissues[23–25]. Generally, revealing somatic mosaicism is challenging, as most acquired alterations are confined to a single or a tiny fraction of cells in an individual body[26]. Capturing mtDNA mosaicism is more complex than in nuclear DNA (nDNA), as a cell contains hundreds to thousands of mtDNA copies, and newly acquired mtDNA mutations would only be confined to a small fraction of mtDNA copies even in a single cell[12]. Recently, the single-cell assay

[1]Graduate School of Medical Science and Engineering, Korea Advanced Institute of Science and Technology (KAIST), Daejeon, Republic of Korea. [2]Inocras Inc, Daejeon, Republic of Korea. [3]Department of Medicine, Washington University School of Medicine, St. Louis, MO, USA. [4]Department of Internal Medicine, Gangnam Severance Hospital, Yonsei University College of Medicine, Seoul, Republic of Korea. [5]Department of Obstetrics and Gynecology, Seoul National University Hospital, Seoul, Republic of Korea. [6]Department of Pathology, Seoul National University Hospital, Seoul, Republic of Korea. [7]Center for Colorectal Cancer, Research Institute and Hospital, National Cancer Center, Goyang, Republic of Korea. [8]Department of Nuclear Medicine, Korea University College of Medicine, Seoul, Republic of Korea. [9]Department of Anatomy, Yonsei University College of Medicine, Seoul, Republic of Korea. [10]Department of Obstetrics and Gynecology, Seoul National University Bundang Hospital, Seongnam, Republic of Korea. [11]Department of Obstetrics and Gynecology, Seoul National University College of Medicine, Seoul, Republic of Korea. [12]Department of Surgery, Seoul National University College of Medicine, Seoul, Republic of Korea. ✉e-mail: ysju@kaist.ac.kr

for transposase-accessible chromatin using sequencing (scATAC-seq) has been applied to reveal mtDNA mutations in single cells[27,28], but insufficient mtDNA depth per cell has disallowed sensitive profiling[29].

Most of our understanding of acquired mtDNA alterations has been derived from cancer studies[30–35]. In this study, we aimed to investigate the whole-genome sequences (WGSs) from 2,096 colonies expanded from healthy (nontumor) single cells (hereafter referred to as clones)[5–8]. This approach enabled the sensitive and accurate detection of single-cell mtDNA variants in multiple cells of an individual. Using this approach, we traced the origin of heteroplasmic mtDNA variants, the absolute rate of mtDNA mutations and the dynamics of age-dependent changes in heteroplasmy levels in somatic lineages.

## Results

### Landscape of mtDNA heteroplasmy in normal cells

We explored 2,096 WGSs of clones expanded from nonneoplastic healthy single cells collected from the colorectal epithelium (431 crypts from 20 individuals)[6], fibroblasts (334 cells from 7 individuals)[5] and hematopoietic stem and progenitor cells (HSPCs; 1,331 cells from 4 individuals)[7,8] (Fig. 1a and Supplementary Tables 1 and 2). In addition, we analyzed 31 WGSs from tumors, including 19 matched colorectal carcinoma bulk tissues from individuals who donated normal colorectal clones and 12 clones established from adenomatous polyps from one individual with MUTYH-associated polyposis[6].

Using the variant allele frequencies (VAFs) of the somatically acquired mutations in nDNA, we verified the clonality of the clones (Extended Data Fig. 1a). The average mtDNA read-depth was 6,931× from normal clones (188× to 40,421×; Extended Data Fig. 1b,c), allowing for robust assessment of mtDNAs in a single clone to a heteroplasmy level of ~0.3%. For more systematic analysis, we established and applied a locus-specific background noise matrix (Methods; Extended Data Fig. 1d–g and Supplementary Table 3). To trace the developmental origin of mtDNA alterations, we constructed the early embryonic phylogeny of the clones using shared somatic nDNA mutations[3,5]. Of note, the first branching in each phylogeny was close to the first cell division in life, as reported previously[3,5], given the VAFs of lineage-defining variants in the matched bulk blood tissues (Extended Data Fig. 2 and Supplementary Note 1).

Overall, we identified 6,451 mosaic mtDNA base substitutions and insertions and deletions (InDels) from the normal clones, revealing an average of 3.1 mtDNA alterations per clone (Fig. 1b and Supplementary Table 4). Most clones (92.4%; 1,937 of 2,096) exhibited one or more mtDNA alterations, and approximately 18% of the clones (383 of 2,096) carried one or more nearly homoplasmic mtDNA alterations (defined as VAF > 90%). We believe that VAFs of mtDNA alterations in each clone (referred to as clone-VAFs hereafter) are approximate to original levels in the clone's founder cell, as clone-VAFs were overall consistent throughout cell culture (Extended Data Fig. 3a–c and Supplementary Note 2). Additionally, direct genome sequencing of colorectal crypts obtained via laser-capture microdissection (LCM) revealed a highly similar mtDNA mutational landscape, indicating minimal culture-associated bias in mutational diversity (Extended Data Fig. 3d–g and Supplementary Note 2).

The spectrum of mtDNA base substitutions was predominantly composed of transitions (C:G>T:A and T:A>C:G base substitutions; collectively 95%; Fig. 1c). These alterations exhibited an extreme level of replication-strand asymmetry, as previously observed in cancers[31,32]. Generally (outside the heavy strand replication origin; m.192-16,196), mutated cytosine bases of C:G>T:A alterations were predominantly on the heavy strand (92.5%), despite the scarcity of cytosines on the strand ($n_{cytosine}$:$n_{guanine}$ = 1:2.4; Fig. 1d). Similarly, mutated thymine bases of T:A>C:G alterations were predominantly on the light strand (63.4%), despite their relative rarity on the strand ($n_{thymine}$:$n_{adenine}$ = 1:1.3; Fig. 1d). Additionally, the strand asymmetry was reversed within the replication origin (m.16,197-191; Fig. 1c,e), where the bidirectional mtDNA replication

process is operative[36,37]. These collectively suggest that mtDNA variant acquisition is tightly coupled with the strand-asymmetric mtDNA replication process, as speculated previously[31]. However, the strand asymmetry was not completely uniform across cell types ($P = 3.3 \times 10^{-52}$, Pearson's chi-squared test; Fig. 1d), implying that the mtDNA replication processes may be slightly different across cell types.

We occasionally observed localized acquisition of multiple mtDNA variants[32]. For example, 12 substitutions, with similar clone-VAFs (1.1–2.5%), were detected in a fibroblast clone (Fig. 1f). These were predominantly T:A>C:G substitutions (11 of 12), and six of them were enriched in a localized region (m.7,318-8,388) with direct evidence of coclonality in phasing, suggesting that a single mutational hit may create multiple mutations in mtDNA, like kataegis in nDNA[38].

### Two origins of mtDNA alterations

Using shared patterns in the developmental phylogenies and tissues, we categorized the origin of mosaic mtDNA alterations into the following two main groups: (1) heteroplasmy in the fertilized egg (termed $\text{Het}_{FE}$ variants; n = 409 alterations, 153 events when collapsed) and (2) postzygotic mutations acquired in somatic lineages (termed postzygotic mutations; n = 6,042; Fig. 2a,b and Extended Data Fig. 4a). Briefly, consistent with their presence from the first cell of life, $\text{Het}_{FE}$ variants were shared by multiple clones and/or tissues in a particular individual. In contrast, postzygotic mutations were predominantly confined to one or a few clones (n = 5,652; either as singletons (n = 5,276) or coincidentally recurrent mutations (n = 376); referred to as postzygotic simple ($\text{PZ}_{simple}$) mutations). A small subset of postzygotic mutations (n = 390 from 32 mtDNA sites) were recurrent across multiple clones and not confined to a specific donor (referred to as postzygotic recurrent ($\text{PZ}_{recurrent}$) mutations), suggesting a higher mutation rate at these sites compared to other mtDNA loci.

### mtDNA heteroplasmy in the fertilized egg

Annotating mtDNA mosaicism with early developmental phylogenies enabled us to capture $\text{Het}_{FE}$ variants[5]. For example, m.16,400 C>T substitution was shared by 14 fibroblast clones (51.9% of 27 clones) established from DB2 (Fig. 2c). Despite its high prevalence in DB2, the variant was extremely rare in clones from other donors (0.1%; two of 2,069 clones). Similarly, m.7,496 T>C substitution was recurrently but exclusively observed in HC19, including three normal colorectal clones (13.0% of 23 clones) and their matched colorectal cancer tissue (Fig. 2d). In both cases, the mutant clones converged at the first node of each phylogeny (Fig. 2c,d). These patterns strongly suggest that the most recent common ancestor (MRCA) cell, possibly the fertilized egg, carried the heteroplasmic variants. Consistent with their pregastrulation timing, these variants were also found in matched bulk blood tissues with substantial VAFs (0.584 and 0.149, respectively; Fig. 2c,d).

Overall, we categorized 153 variant events as $\text{Het}_{FE}$ variants (Supplementary Table 5). They include 391 shared variants by multiple clones in an individual (6.1% of the total mtDNA variants; 135 events when collapsed) and 18 singleton variants in clones but shared by matched blood tissues. These variants were twofold enriched in the D-loop (m.16,024-576) and 1.5-fold depleted in the rRNA regions (m.648-1,601 and m.1,671-3,229) compared to $\text{PZ}_{simple}$ mutations[39] (P = 0.0031 and 0.0363, respectively, two-sided Fisher's exact test; Extended Data Fig. 4b).

Then, we inferred the original heteroplasmy levels in the fertilized egg of $\text{Het}_{FE}$ variants. Notably, we observed that the average clone-VAF value of a $\text{Het}_{FE}$ variant across all clones from a donor (referred to as clone-averaged VAF (caVAF); Extended Data Fig. 4c) closely correlated with the heteroplasmy level in the matched polyclonal blood tissue (R = 0.967, P = 2.3 × 10⁻¹⁶, Pearson's correlation; Fig. 2e). We speculated that a plausible mechanistic link between these two independent values was the heteroplasmy level in its origin; although clone-VAFs of a $\text{Het}_{FE}$ variant may fluctuate across clonal lineages with aging, the average

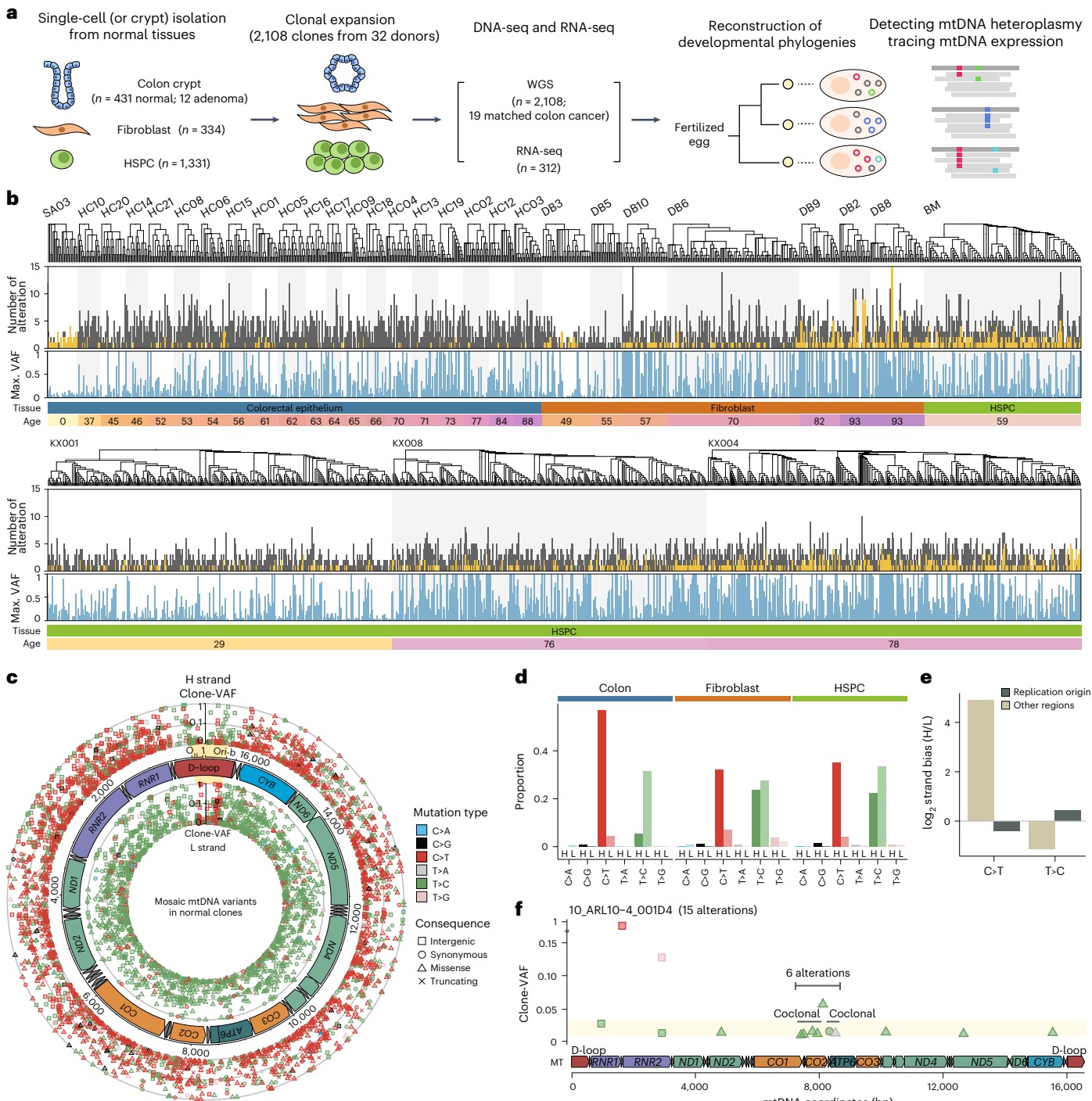

**Fig. 1 | Landscape of mtDNA heteroplasmy in normal cells. a**, Experimental design. **b**, mtDNA mosaicism identified in 2,096 normal clones from 31 donors. Donor, developmental phylogeny, total number of mtDNA variants, maximum clone-VAF, tissue type and donor age for each clone are shown. Shared variants among clones from an individual are shown in yellow. **c**, Landscape of mosaic mtDNA variants. Variants were classified as heavy (external circle) or light (internal circle) strands according to the mutated pyrimidine. Mutation types and consequences are represented by colors and shapes. The heavy strand

replication origin region (Orib-$O_H$) is highlighted in yellow. **d**, Strand-asymmetric mutational spectrum across three tissue types. **e**, Strand bias of C:G>T:A and T:A>C:G base substitutions according to the heavy strand replication origin (Orib-$O_H$; m.16,197-191). The $\log_2$-transformed ratio between the numbers of heavy and light strand mutations is shown. **f**, Distribution of clone-VAFs and phasing of mtDNA alterations in a fibroblast clone, 10_ARL10-4_001D4. H, heavy strand; L, light strand.

(caVAF) would remain overall stable from the original heteroplasmy level, consistent with our computational simulation (Extended Data Fig. 4d). Similarly, as the VAF from the bulk blood tissues (blood-VAF) inherently represents an averaged VAF among many polyclonal blood cells, it should also closely reflect the initial heteroplasmy level.

We extended our speculation to the correlation of VAFs of heteroplasmic mtDNA variants between buccal–buccal and/or buccal–blood tissues in 19 monozygotic twins (Extended Data Fig. 4e,f). Therefore, we used caVAF as a proxy for the heteroplasmy level in the fertilized egg of a $Het_{FE}$ variant (Supplementary Note 3).

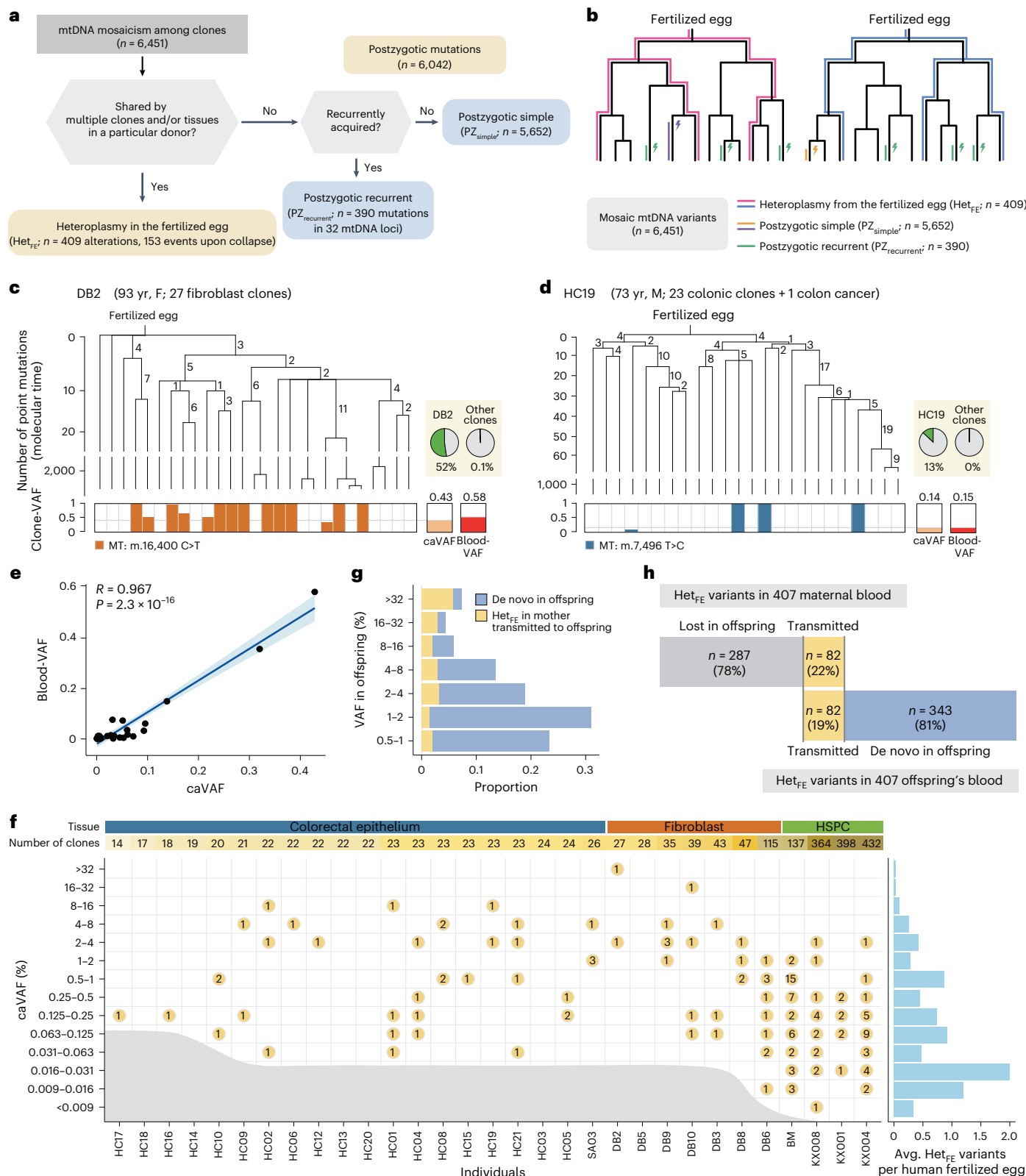

Most donors (80.6%; 25 of 31) carried one or more Het$_{FE}$ variants with caVAFs over 0.03% (Fig. 2f). Twelve individuals (39%) had Het$_{FE}$ variants with substantial caVAFs (>4%). As expected, the statistical power for capturing Het$_{FE}$ variants was associated with the number of clones in a donor. For example, a Het$_{FE}$ variant was identified with caVAF as low as 0.047% from HC02 (22 clones). In contrast, the minimum caVAF of a Het$_{FE}$ variant was sevenfold lower (0.0067%) in KX008 (364 clones). Considering the detection sensitivity, we profiled the average

landscape of Het$_{FE}$ variants, which showed ~2 Het$_{FE}$ variants over 0.5% heteroplasmy level per fertilized egg (Fig. 2f).

Notably, we believe that the actual number of Het$_{FE}$ variants is higher than we observed, as our detection thresholds were ~0.02% for most donors. Given that a fertilized egg typically contains ~100,000 mtDNA copies[40], Het$_{FE}$ variants detectable in this study should be shared by at least 20 mtDNA copies in the first cell, and those restricted to a smaller number of mtDNA copies would likely be undetectable.

**Fig. 2 | Fertilized egg-originated variants in mtDNA. a**, A simple diagram showing how mtDNA alterations were categorized. **b**, Schematic diagram illustrating different shared patterns of mtDNA alterations according to their origin. **c,d**, Examples of Het$_{FE}$ variants—m.16,400 C>T (**c**) and m.7,496 T>C (**d**). Early clonal phylogenies, reconstructed using somatic nDNA mutations, are shown. Branch lengths are proportional to the number of somatic nDNA mutations. Clone-VAF in each clone, caVAF and blood-VAF are represented by bar plots at the bottom. Two pie charts in **c** and **d** indicate the proportions of mutant clones among clones of the individual (left) and among clones of other individuals (right). **e**, Linear correlation between the caVAF and blood-VAF. The blue line represents the regression line, and the shaded area indicates its 95% confidence interval. Pearson's correlation coefficient and *P* value are provided.

Two-sided Pearson's correlation. **f**, Landscape of Het$_{FE}$ variants detected. The colors in 'number of clones' represent the number of clones for each individual with corresponding values indicated alongside. The numbers within the yellow circles indicate the count of detected Het$_{FE}$ variants with the caVAF within the specified ranges. The gray area represents the range of caVAF that cannot be detected when considering the number of clones for each individual. The average landscape of Het$_{FE}$ variants is shown on the right. **g**, The VAF distribution of heteroplasmic mtDNA variants in offspring obtained from bulk blood of 407 mother–offspring pairs. **h**, A bar plot categorizing Het$_{FE}$ variants found in the mother's bulk blood (Het$_{FE}$ variants in 407 maternal blood) and the offspring's bulk blood (Het$_{FE}$ variants in 407 offspring's blood).

In addition, we speculate that the origin of most Het$_{FE}$ variants found in this study was the maternal germline rather than new acquisitions in the fertilized egg, as newly acquired mutations would be restricted to a single mtDNA copy.

To validate our findings, we explored WGSs from bulk blood tissues of 294 families (including 407 mother–offspring pairs)[41]. We discovered 425 heteroplasmic variants (>0.5% VAF) in the polyclonal blood of offspring, which are most likely Het$_{FE}$ variants in offspring (Fig. 2g). We further found that ~20% of the variants were heteroplasmic in the polyclonal blood of the mother (likely Het$_{FE}$ variants of the mother; Fig. 2g,h, Extended Data Fig. 4g and Supplementary Note 4). Our findings collectively indicate that (1) mtDNA heteroplasmy in the fertilized egg is not rare, likely being continuously generated in the germline; (2) a substantial fraction of Het$_{FE}$ variants are transmitted to the next generation[39], despite the purification process during oogenesis in the maternal germline lineage[42,43] and (3) these variants are one of the sources of mtDNA mosaicism observed in aged somatic cells.

## mtDNA turnover and drift in somatic lineages

The distribution of clone-VAFs of a Het$_{FE}$ variant among the clones exhibited pressures that were shifting them to both extremes (0% or 100%) from the initial heteroplasmy level (Fig. 3a; two examples in Fig. 2c,d). For instance, the m.16,256 C>T mutation in DB10 (39 clones), which had a caVAF of 0.32, was observed as homoplasmic in 11 clones (28.2%) and almost wild type in 25 clones (64.1%; Fig. 3b). Two underlying possible scenarios include the following: (1) early embryonic bottleneck during progressive mtDNA copy number reduction in the cleavage of early embryogenesis[44,45] and (2) lifetime drift through the continuous mtDNA turnovers in each somatic lineage for a lifetime[46,47] (Fig. 3c).

The foundation of the early embryonic mtDNA bottleneck is caused by the lack of mtDNA replication until a certain stage of embryogenesis[48,49] (Fig. 3c). If each embryonic cell has one or only a few mtDNA copies at a certain stage, the heteroplasmy level can be quantized according to the composition of founder mtDNAs in each embryonic cell.

In parallel, mtDNAs are lost and newly replicated in somatic lineages[12,50,51] (for example, cell-cycle-dependent mtDNA duplication and random segregation by half in two daughter cells in dividing cells (mitotic turnover) or cell-cycle-independent homeostatic mtDNA replacement in nondividing cells (homeostatic turnover); Fig. 3c). The processes can slightly drift heteroplasmy levels continuously over time, generating a substantial impact in a lifetime. Of these two nonexclusive scenarios, our observations indicate that the lifetime drift is dominant.

First, purification of Het$_{FE}$ variants was age-dependent or much weaker in clones from young donors (for example, clones established from an aborted 19-week-old fetus; Fig. 3d,e and Extended Data Fig. 5a). This suggests that purification was not fixed in the early stages of human life. Second, sister clone pairs that branched out at a later time point did not exhibit more similar heteroplasmy levels of a Het$_{FE}$ variant than clone pairs that diverged earlier (Fig. 3f). For example, clone pairs that had an MRCA cell at the ~30th cell generation, which was much later than the early embryonic bottleneck, showed tremendous heterogeneity in clone-VAFs of a Het$_{FE}$ variant (Fig. 3f).

Finally, the computational simulation suggested that the lifetime drift model alone was sufficient to explain the skewed distribution of clone-VAFs in a Het$_{FE}$ variant. Simulation studies using the mitotic turnover model (Extended Data Fig. 5b,c) indicated that 1,440 rounds of mtDNA mitotic turnovers shifted a Het$_{FE}$ variant with 10% initial heteroplasmy level (caVAF) to homoplasmy (100%) in ~10% of the clones when clones had 750 basal mtDNA copy numbers (a turnover was defined as replication of an mtDNA for *n* times, where *n* is the basal mtDNA copy number in a somatic cell; Fig. 3g). Likewise, simulations assuming homeostatic turnover (Extended Data Fig. 5d,e) suggested a similar conclusion, but ~50% of rounds were necessary for a similar effect under the same conditions (Supplementary Note 5).

Based on the clone-VAF distributions of Het$_{FE}$ variants, the maximum likelihood mtDNA turnover rates across cell types were inferred (14.3, 20.8 and 17.9 mitotic turnovers per year, or 6.5, 11.5 and 9.4 homeostatic turnovers per year for the colon epithelium, fibroblasts and HSPCs, respectively; Fig. 3h). Although we believe that mitotic and homeostatic mtDNA turnovers are predominant mechanisms

**Fig. 3 | mtDNA turnover leading to mtDNA drift over a lifetime. a**, Clone-VAFs of Het$_{FE}$ variants across clones. Clones with a complete absence of corresponding Het$_{FE}$ variants (that is, clone-VAF = 0) are not shown. The colors in age and 'number of clones' represent age and the number of clones for each individual, respectively, with corresponding values indicated alongside. Each column represents each Het$_{FE}$ variant, and a vertical line connects clones carrying the same Het$_{FE}$ variant. Gray bars (bottom) show caVAF values for each Het$_{FE}$ variant. **b**, Clone-VAF distributions of a Het$_{FE}$ variant, m.16,256 C>T, in DB10. A gray dashed line indicates the caVAF. **c**, Schematic diagram illustrating clone-VAF dynamics by early embryonic bottleneck and lifetime drift. Expected mtDNA copy numbers per cell (bottom) are shown. Two alternative drift models (mitotic and homeostatic turnovers) are depicted in different colors. **d**, Clone-VAF distributions of Het$_{FE}$ variants with similar caVAF values in four individuals of different ages. **e**, Linear correlation between age and fixation index for Het$_{FE}$ variants of caVAF > 0.01. Circles represent variants colored by caVAF values.

A gray line and the shaded area represent the regression line and its 95% confidence interval. Pearson's correlation coefficient and *P* value are presented. Two-sided Pearson's correlation. **f**, Clone-VAF difference of a Het$_{FE}$ variant between all possible clone pairs based on their embryonic branching time. Differences were computed when at least one clone-VAF exceeded 0.5 from each pair. Cell generations were calculated using a fixed mutation rate previously reported[5]. **g**, Average mitotic turnover counts to reach homoplasmy according to caVAF from the simulation studies. **h**, Estimated turnover rates for each Het$_{FE}$ variant in mitotic and homeostatic turnover models. Error bars represent the range of 50 simulated results with the lowest MSEs of 1,000,000 simulations per Het$_{FE}$ variant, with circles representing average values. Het$_{FE}$ variants with caVAF > 0.005 were included. Dark gray lines and shaded areas represent average turnover rates and their 95% confidence intervals in each tissue type. EEM, early embryonic mutation; CG, cell generation.

for colorectal epithelium and fibroblasts, respectively, their relative balance between two turnover models in each cell type is uncertain.

## Postzygotic mtDNA mutations

Of the 2,096 clones, 6,042 mtDNA variants (93.7% of all the variants) were categorized as postzygotic mutations, newly acquired from each somatic lineage. As mentioned above, 32 mtDNA loci showed an elevated mutation rate with 390 $PZ_{recurrent}$ variants (Supplementary Table 6 and Supplementary Note 6). These mutations were predominantly located in the hypervariable regions of the D-loop, homopolymer sequences or both[33,52] (Extended Data Fig. 6a). Interestingly, mutations in a hotspot (m.414 T>G) were recurrently found in clones with

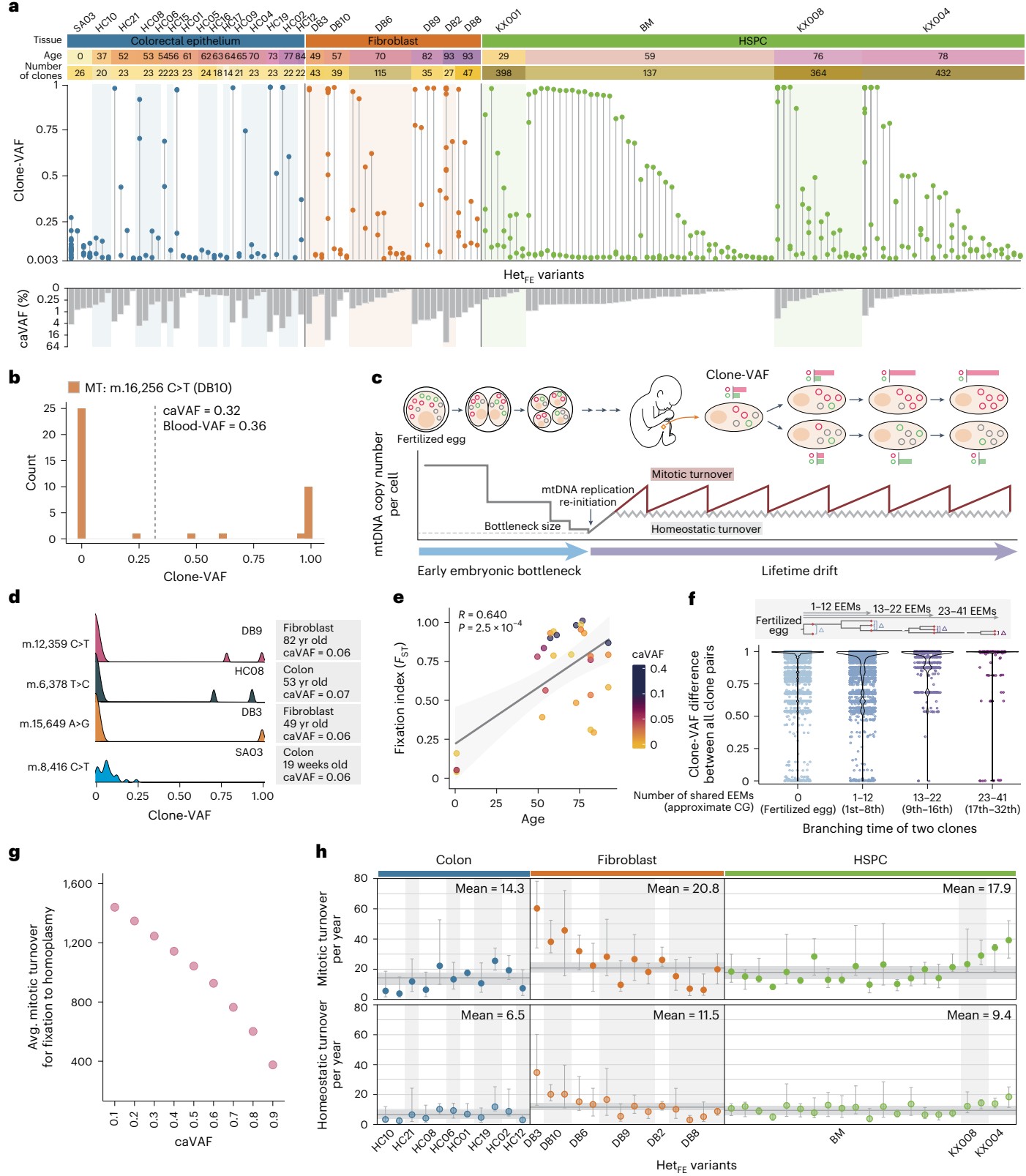

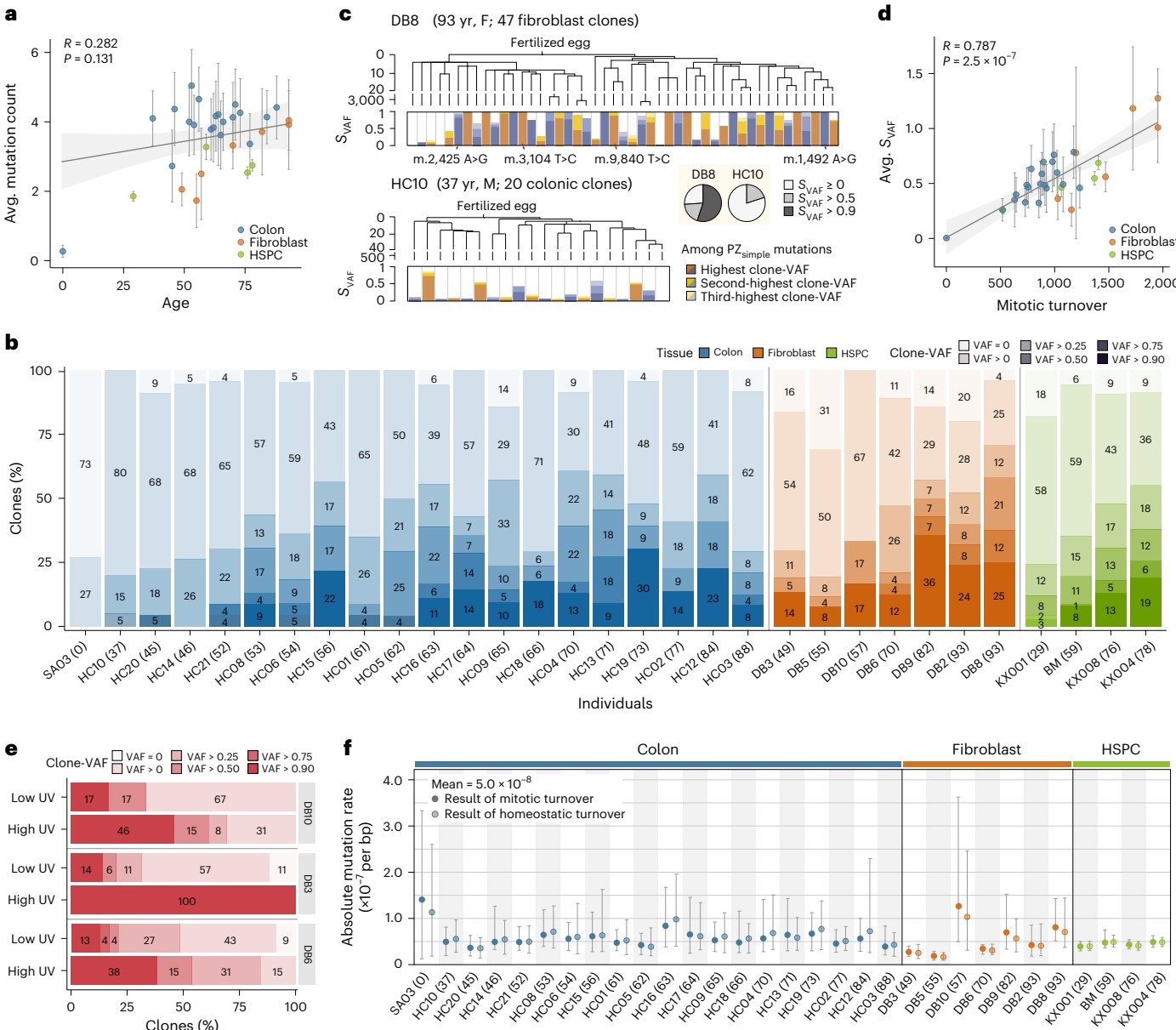

**Fig. 4 | Postzygotic mtDNA variants toward homoplasmy in aged cells.**
**a**, Linear correlation between the average $PZ_{simple}$ mutation count and age across 31 individuals. The gray line and the shaded area represent the regression line and its 95% confidence interval. One individual aged 0 was not included in the regression. Vertical lines indicate the range of $PZ_{simple}$ mutation counts per clone in an individual. **b**, Proportions of clones with maximum clone-VAFs across 31 individuals (individual ages indicated in parentheses). Individuals are sorted by tissues, then by age, in ascending order. **c**, Bar plots of $S_{VAF}$ for each clone in two individuals, DB8 (93 years old; top) and HC10 (37 years old; bottom), with developmental phylogenies. Bar plots include up to the top three clone-VAF $PZ_{simple}$ mutations. Pie charts categorize clones based on $S_{VAF}$. **d**, Linear correlation between average $S_{VAF}$ and total mitotic turnovers across 31 individuals. The total number of mitotic turnovers was calculated using the rates estimated by

$Het_{FE}$ variants for each tissue type. The gray line and shaded area represent the regression line and its 95% confidence interval. One individual aged 0 was not included in the regression. Vertical lines indicate the range of $S_{VAF}$ per clone in an individual. (**a**,**b**,**d**, Clones with high UV exposure were excluded to remove UV radiation's impact. Pearson's correlation coefficient and $P$ value are provided. Two-sided Pearson's correlation.) **e**, Comparison of clone proportions with maximum clone-VAFs of $PZ_{simple}$ mutations in fibroblast clones with low and high UV-derived nDNA mutation burdens in three donors. **f**, Estimated mtDNA mutation rate in 31 individuals under mitotic and homeostatic turnover models (individual ages indicated in parentheses). Error bars represent the range of 50 simulated results with the lowest MSEs of 10,000 simulations per individual, with circles representing average values.

ultraviolet (UV) light exposure (estimated using UV-associated somatic mutations in the nDNA of a clone[53]), suggesting UV-dependent acquisition[54,55] (Extended Data Fig. 6b).

Except for $Het_{FE}$ and $PZ_{recurrent}$ mutations, we detected 5,652 $PZ_{simple}$ mtDNA alterations. Unlike somatic mutations in nuclear genomes, it is challenging to absolutely count $PZ_{simple}$ mutations, as mutations with clone-VAFs below our detection threshold (~0.3%) would remain

undetected. Indeed, the crude number of $PZ_{simple}$ mutations detected in clones was not substantially correlated with age ($R = 0.282$, $P = 0.131$, Pearson's correlation; Fig. 4a). Instead, the overall heteroplasmy levels of $PZ_{simple}$ mutations in clones displayed stronger clock-like properties—$PZ_{simple}$ mutations with higher clone-VAF were more frequent in aged donors than young donors (Fig. 4b), and the sum of the clone-VAFs of all detected $PZ_{simple}$ mutations in a clone (referred to as $S_{VAF}$) showed

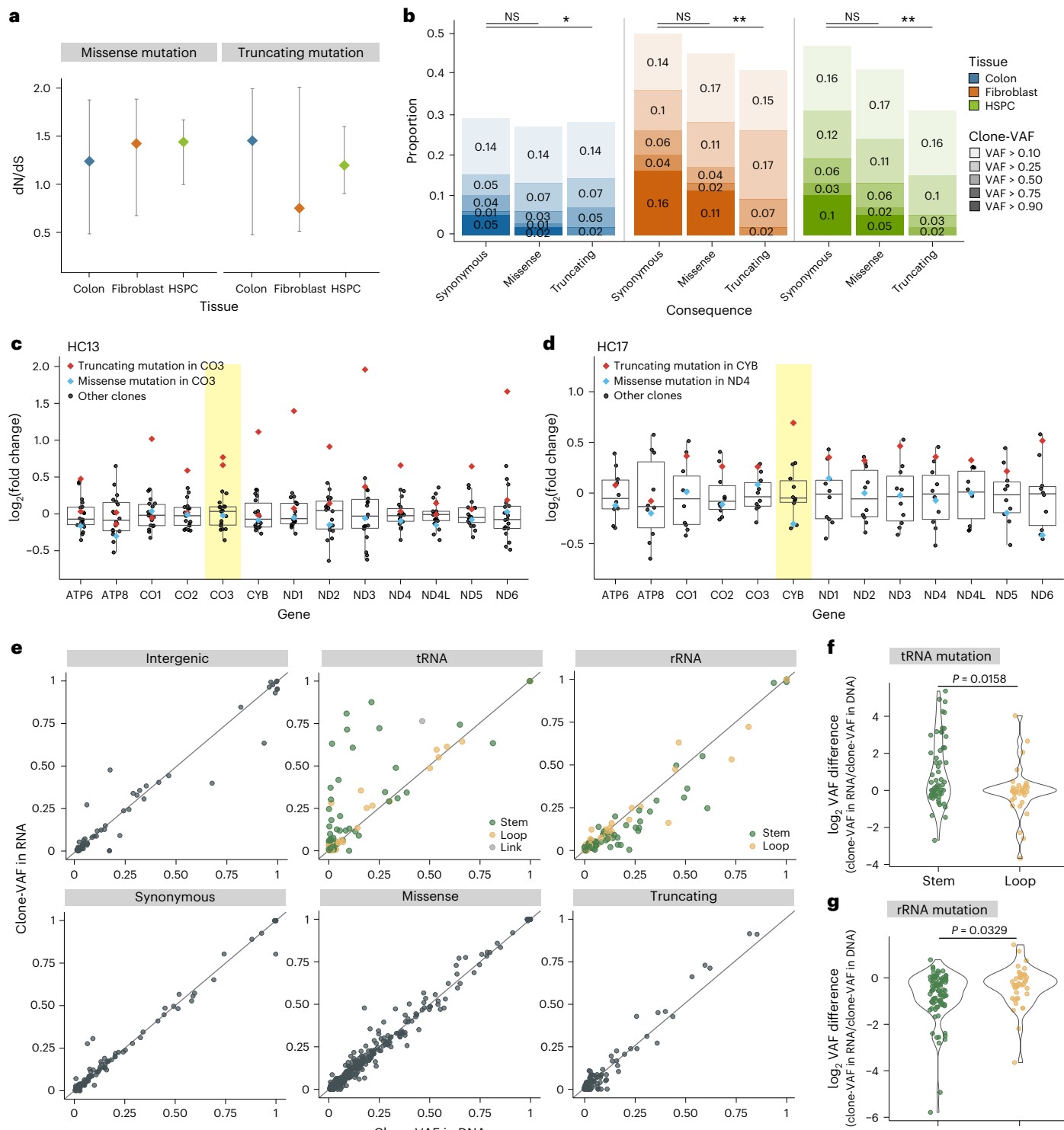

**Fig. 5 | Selection and transcription of mtDNA variants. a**, dN/dS ratios for missense and truncating mutations in each tissue type with simulated null distributions. Error bars represent 25th and 75th percentiles from simulations (10,000 simulations for each donor). **b**, Clone-VAF distribution of synonymous, missense and truncating mutations. Low clone-VAF variants (<0.1) were not included. Two-sided Fisher's exact test, *P < 0.05, **P < 0.01; NS, not significant. Exact P values are 0.0211, 0.0017 and 0.0013 for the colon epithelium, fibroblasts and HSPCs, respectively. **c,d**, log₂-transformed fold changes of expression levels. Normalized read counts from a clone were compared to the average normalized read counts among wild-type (other) clones in HC13 (**c**; 22 clones) and HC17 (**d**; 14 clones). Red and blue diamonds represent clones with truncating and missense mutations, respectively. Black circles indicate wild-type clones. Boxplots

illustrate log₂-transformed fold change variation in wild-type clones with median values, IQRs and whiskers (1.5× IQR). Yellow box highlighting the gene with truncating mutations. **e**, Scatter plots delineating clone-VAFs in genome versus transcriptome sequences according to functional consequences of mutations. Mutations in rRNA and tRNA are further subcategorized based on the secondary structure of the RNA they impact (color-coded). The gray lines represent the diagonal line y = x. **f,g**, Violin plots illustrating how clone-VAF between the genome and transcriptome varies based on tRNA (**f**) and rRNA (**g**) secondary structures. The y axis represents the log₂-transformed ratio of clone-VAF in RNA to clone-VAF in DNA. Mutations with clone-VAF lower than 0.01 were excluded. One-sided Wilcoxon test. IQRs, interquartile ranges.

more measurable characteristics. For example, in an older individual (DB8; 93 years old), 55% of clones (26 of 47) had an $S_{VAF}$ of ~1.0 by 1–3 clone-specific $PZ_{simple}$ mutations (Fig. 4c). In contrast, in a young individual (HC10; 37 years old), all clones exhibited an $S_{VAF}$ far below 1.0 (0.55 versus 0, $P = 5.2 \times 10^{-6}$, two-sided Fisher's exact test; Fig. 4c). Of note, there was no significant difference in the crude number of $PZ_{simple}$ mutations between the clones of the two individuals (Extended Data Fig. 6c). The average $S_{VAF}$ in the clones of an individual exhibited a strong positive correlation with age (Extended Data Fig. 6d). The correlation became stronger when the age of individuals was converted to turnover numbers from birth using the cell-type-specific turnover rates estimated from $Het_{FE}$ variants ($R = 0.787$, $P = 2.5 \times 10^{-7}$, Pearson's correlation; Fig. 4d).

Interestingly, we observed that a few fibroblast clones with a higher amount of lifetime UV-light exposure exhibited a higher $S_{VAF}$ of $PZ_{simple}$ mutations than those with a lower amount of lifetime UV-light exposure ($P = 7.7 \times 10^{-4}$, two-sided Fisher's exact test; Fig. 4e). This indicates that UV exposure accelerated mtDNA turnovers in the cellular lineage. We speculate that UV exposure damages mtDNA, followed by mtDNA degradation and triggering additional mtDNA replications for their replacement[21]. Of note, the mtDNA mutational signatures in clones with a higher UV exposure were similar to the other clones (Extended Data Fig. 6e), indicating that UV light does not directly lead to $PZ_{simple}$ mutations fixed in mtDNA.

With the mtDNA turnover rates estimated using $Het_{FE}$ variants and the landscape of detectable $PZ_{simple}$ mutations, we estimated the absolute number of mtDNA alterations that are newly appearing in every mtDNA replication. In all individuals and both turnover models, the absolute mtDNA mutation rates converged to $5.0 \times 10^{-8}$ alterations per base pair (bp) replication (Fig. 4f). Interestingly, our estimate was within the range of error rates of polymerase γ (*POLG*), the mitochondrial genome's DNA polymerase[56,57]. The converged rate reassures that (1) endogenous mtDNA replication is the dominant process for mtDNA mutation acquisition in somatic cells[31,58,59] and (2) both turnover models (and their turnover rates) are reliable. Given the ~750 mtDNA copies in a single somatic cell, our absolute mutation rate implies an average of 0.31 de novo $PZ_{simple}$ mtDNA alteration is acquired per daughter cell per cell division.

## Selective pressure of mtDNA mutations in normal cells

To understand the selective pressure on $PZ_{simple}$ mutations, we calculated the dN/dS ratio[60–62]. The ratio of missense or truncating mutations to synonymous mutations was not substantially higher than mtDNA mutations randomly generated according to the mtDNA mutational signature, indicating general neutrality in mutation acquisition (Fig. 5a). However, truncating mutations exhibited lower clone-VAFs than synonymous mutations in all three cell types, with no mutations exceeding 90% clone-VAFs, suggesting constrained expansion of mtDNAs carrying inactivating mutations due to functional disadvantage when reaching homoplasmy ($P = 0.0211$, $0.0017$ and $0.0013$ for the colon epithelium, fibroblasts and HSPCs, respectively, two-sided Fisher's exact test; Fig. 5b). These observations were consistent with previous observations in cancer tissues[31,32].

Despite the expansion constraint, 15 truncating mutations displayed high clone-VAFs among the clones (clone-VAF > 0.6), accompanied by upregulated RNA expression levels of mtDNA genes (Fig. 5c,d). This phenomenon is likely attributable to a compensatory response where transcript degradation is inhibited when the protein product is dysfunctional[63,64]. The similarity in clone-VAFs between genome and transcriptome sequences indicates that this inhibitory effect does not distinguish between wild-type and truncated mtDNA (Fig. 5e).

We further compared the clone-VAFs of $PZ_{simple}$ mutations in genome and transcriptome sequences (Fig. 5e). Although most mtDNA mutations showed similar clone-VAFs in both, a subset of tRNA mutations exhibited elevated clone-VAFs in transcriptomes, which is

consistent with a previous report[65]. In contrast, a subset of rRNA mutations showed reduced clone-VAFs in transcriptomes. These mutations were predominantly clustered within stem regions of tRNA and rRNA ($P = 0.0158$ and $0.0329$ for tRNA and rRNA mutations, respectively, one-sided Wilcoxon test; Fig. 5f,g). We speculated that these mutations influence the stability and regulation of these RNAs, leading to tRNA accumulation and rRNA degradation[65,66].

## mtDNA copy number and structural variations (SVs) in normal cells

The average mtDNA copy number was ~750 per cell (per diploid nuclear genome), but large variations in mtDNA copy number were observed across clones, even in an individual (Fig. 6a). For example, mtDNA copy numbers among the clones of HSPCs from KX004 ranged from ~20 to 3,700. There was no apparent correlation between median mtDNA copy number and age ($R = 0.127$, $P = 0.381$, Pearson's correlation). Notably, interclonal mtDNA copy number variations were less substantial in colorectal clones (Fig. 6a). Despite these variations, gene expression levels of mtDNA and nDNA genes were not substantially altered among the clones, suggesting that the mtDNA copy number is not a bottleneck for the transcription of mtDNA genes, at least at the resting stage (Extended Data Fig. 7a,b).

Two colorectal clones had notable SVs within their mtDNA (Fig. 6b and Extended Data Fig. 7c), with deletions of 10,951 bp and 3,389 bp, respectively, at approximately 45% heteroplasmy levels. As expected, gene expression levels in the deleted loci were lower than in the flanking regions ($P < 0.05$, Wald test; Extended Data Fig. 7d). Notably, these large deletions have been observed in cancers at a similar frequency[32]. Our findings illustrate that SVs can occur in normal clones[67]; however, these rare events involve only approximately 0.1% of normal cells.

## Accelerated mtDNA turnover in tumorigenesis

In 19 matched colorectal cancer tissues, we observed, on average, more detectable mutations (5.3 versus 3.8; $P = 0.0301$, Wilcoxon signed rank exact test; Fig. 6c) and higher $S_{VAF}$ values ($P = 8.5 \times 10^{-4}$, Wilcoxon signed rank exact test; Fig. 6d) than normal clones from the same donor. Our findings suggest an elevated mtDNA mutation rate, turnover rate or both during tumor initiation and clonal evolution[68]. Consistent with this speculation, in 12 clones established from MUTYH-associated adenomatous polyps[6], homoplasmic mtDNA mutations were more frequently observed in lineages with more driver mutations (Extended Data Fig. 7e).

We further investigated detectable $PZ_{simple}$ mutations in 70 colorectal carcinomas (19 matched and 51 unrelated colorectal cancers[69]; Supplementary Table 7). Qualitatively, colorectal cancers exhibited a notably higher prevalence of truncating mutations with >0.6 VAFs than normal clones (0.0203 versus 0.0026, $P = 1.5 \times 10^{-4}$, two-sided Fisher's exact test; Fig. 6e). This finding suggests increased accumulation of deleterious mutations in colorectal cancers, as observed previously[32].

Finally, compared to the mtDNA copy numbers in normal clones, 19 matched colon cancer tissues demonstrated biased copy number changes (per diploid nuclear genome) toward either gain or loss of mtDNA copies at face value (Fig. 6f). To gain insights into the mtDNA copy numbers in pure colon cancer cells without co-existing tumor microenvironmental cells, such as infiltrating lymphocytes, we correlated mtDNA copy numbers of cancer tissues with their tumor cell fractions estimated from genome sequences[70] and found a strong positive linear relationship ($R = 0.715$, $P = 5.7 \times 10^{-4}$, Pearson's correlation; Fig. 6g). Extrapolation of the regression line suggested ~1,266 mtDNA copies per diploid nuclear genome at 100% tumor cell fraction, which is 70% higher than in normal colorectal clones. Indeed, we confirmed an mtDNA copy number increase in colon cancer cells by WGSs of 14 colon cancer organoids (100% tumor cell fraction; 1,224 mtDNA copies per diploid cancer cell; Extended Data Fig. 7f). The underlying reason for the mtDNA copy number gain in cancer cells is uncertain.

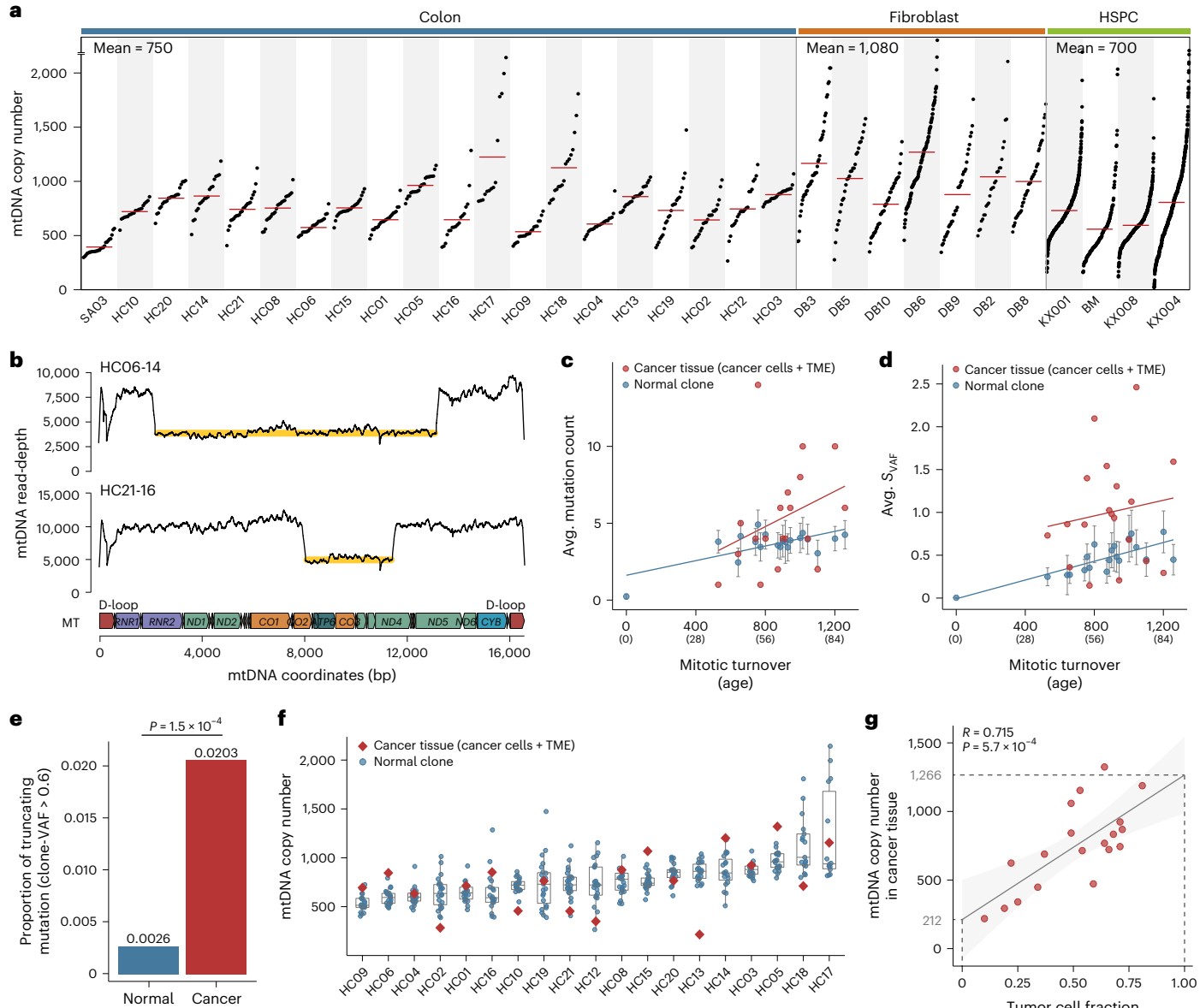

**Fig. 6 | mtDNA copy number in somatic cells and mtDNA in cancer. a**, mtDNA copy number distributions among 31 individuals, sorted by tissue type, then by age in ascending order. Black dots and red bars represent clones (*n* = 2,096) and mean values, respectively. **b**, Read-depth of the mitochondrial genome showing large deletions in two colorectal clones (HC06-14 and HC21-16). Yellow lines represent the deleted regions. **c**,**d**, Mutation number (**c**) and $S_{VAF}$ (**d**) in normal colorectal clones (blue) and matched colon cancer tissues (red) are correlated with the number of mitotic turnovers across 19 donors. The age of donors is shown in parentheses. Vertical lines indicate the range across clones in each donor. Red and blue lines represent regression lines. **e**, The proportion of truncating mtDNA mutations within normal colorectal clones and colorectal cancer tissues. Two-sided Fisher's exact test. **f**, Distributions of mtDNA copy numbers in normal colorectal clones and matched cancer tissues among 19 individuals. Boxplots illustrate median values with IQRs and whiskers (1.5× IQR). **g**, The linear correlation between tumor cell fraction and mtDNA copy numbers per diploid nuclear genome in cancer tissues. The gray line and the shaded area represent the regression line and its 95% confidence interval, respectively. Copy number values at 0% and 100% tumor cell fractions are shown by extrapolation. Pearson's correlation coefficient and *P* value are provided. Two-sided Pearson's correlation. TME, tumor microenvironment.

Similarly, mtDNA copy numbers in cancer tissues were negatively correlated with the amount of infiltrating CD3⁺ T cells (Extended Data Fig. 7g). Genome sequencing of T cells sorted from the peripheral blood suggested that there were ~123 mtDNA copies per T cell (Extended Data Fig. 7f), which was close to the value extrapolated from the regression line (Fig. 6g).

## Discussion

By leveraging WGSs derived from 2,096 healthy normal clones encompassing three different tissues, we elucidated the landscape of mtDNA mosaicism across single cells. Our system allowed the tracing of the embryonic origin of the mtDNA variants. Unlike the conventional wisdom of homogeneous mtDNA in fertilized eggs[42,43], we conclude that human fertilized eggs frequently harbor heteroplasmic mtDNA variants, often showing substantial heteroplasmic levels (that is, VAF > 30%).

The detection of Het_FE variants allowed the determination of one of two essential parameters contributing to the landscape of mtDNA mosaicism—the mtDNA turnover rates in somatic cells. Then, by applying the turnover rate to the landscape of PZ_simple mutations, the other critical parameter, the absolute mtDNA mutation rate per mtDNA replication, was elucidated. Despite their importance in understanding

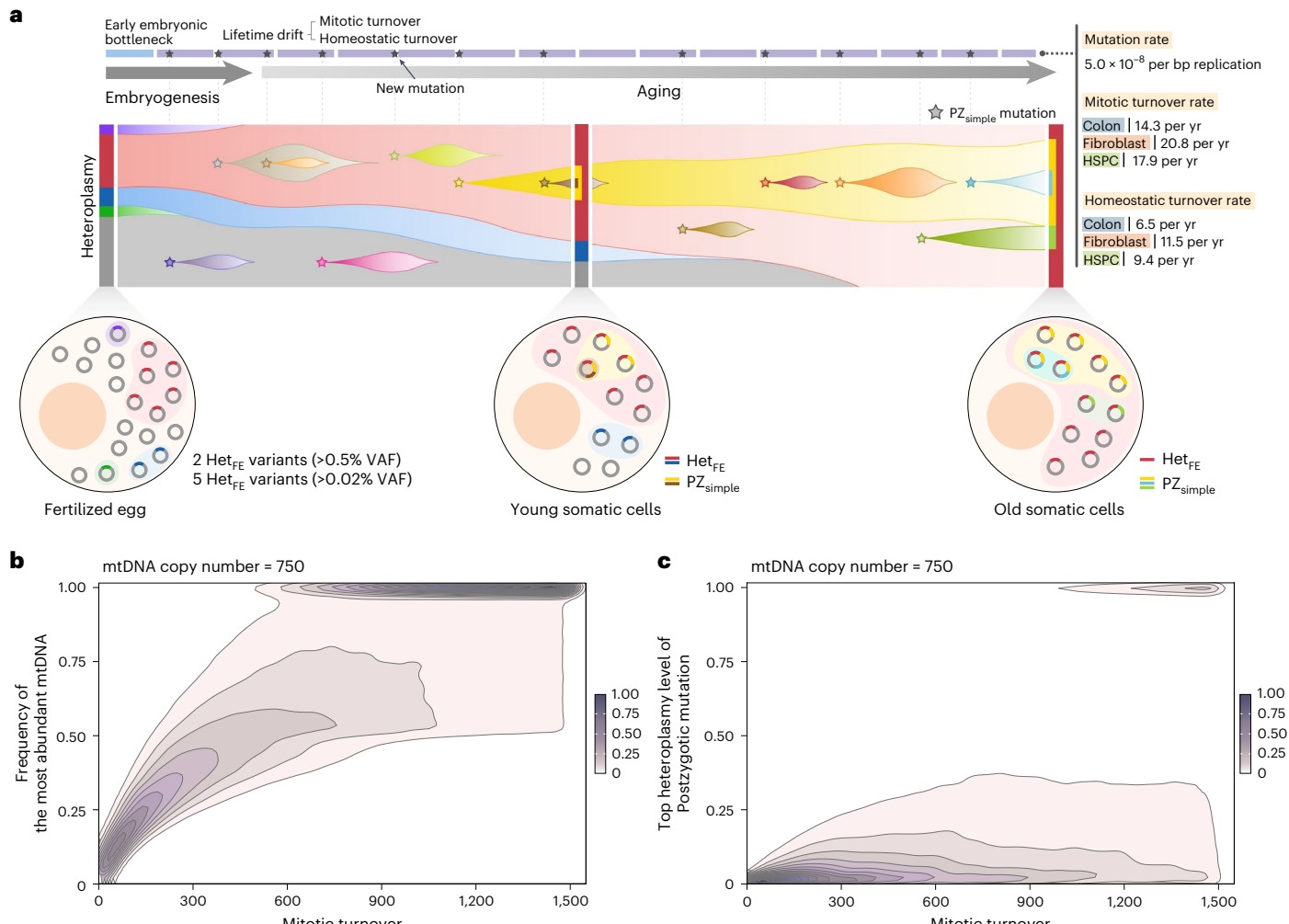

**Fig. 7 | Model for mtDNA dynamics in somatic lineages over a lifetime.**
**a**, Schematic diagram illustrating the origin and dynamics of mtDNA alterations across a lifetime. Heteroplasmic variants in the fertilized egg and somatically acquired postzygotic mutations undergo drift in somatic lineages. Mutation and turnover rates (mitotic and homeostatic turnover rates) are shown at the top right. Stars indicate $PZ_{simple}$ mutations. Bars on the left, middle and right depict clone-VAF distribution within the cell at different time points. **b**, A contour plot representing how an mtDNA population changes with continuous mitotic turnover from simulation studies assuming a baseline mtDNA copy number of 750. The $x$ axis shows the mitotic turnover count and the $y$ axis shows the frequency of the most prevalent mtDNA, regardless of mutations. **c**, A contour plot representing how clone-VAF of $PZ_{simple}$ mtDNA mutations changes with continuous mitotic turnover from simulation studies assuming a baseline copy number of 750 and absolute mutation rate of $5.0 \times 10^{-8}$ per bp replication. The $x$ axis shows the mitotic turnover count and the $y$ axis shows the top heteroplasmy level of postzygotic mutation.

mtDNA mutational dynamics in somatic cells, it has been challenging to decompose these two parameters individually, as both are intermingled. For example, mtDNA mutations without mtDNA expansion cannot be detected, and mtDNA expansion cannot be tracked without mtDNA mutations.

Our findings suggest that stochastic lifetime drift alone can shape the mtDNA heteroplasmy landscape observed in this study (Fig. 7a). The replication-strand-asymmetric mutational spectrum and constant mutation rate suggest that $PZ_{simple}$ mutations arise primarily through replication-associated mechanisms. Our lifetime drift model illustrates that 1,000 mitotic turnovers induce one of 750 mtDNA copies to have a completely purified mtDNA composition in ~30% of somatic cells (Fig. 7b and Supplementary Table 8) and result in homoplasmic $PZ_{simple}$ mutations in ~5% of somatic cells (Fig. 7c and Supplementary Table 9). The extent of lifetime drift generally decreases as the basal mtDNA copy number increases (Extended Data Figs. 8 and 9).

The acquisition of truncating mutations was not substantially constrained, but their drift to homoplasmy was repressed in somatic lineages due to functional disadvantages. The mtDNA-truncating mutations were more common in colorectal cancers than in normal tissues, indicating that cancer cells may depend less on functional mitochondria. Increased mutation counts, expansion and mtDNA copy numbers in colorectal tumors suggest that mtDNA dynamics may change with a potential impact during colorectal tumorigenesis.

When certain mtDNA mutations exceed specific heteroplasmy levels, they can cause mitochondrial dysfunction, a hallmark of aging[71]. Our study meticulously outlines the general landscape of mtDNA mosaicism in apparently normal cells and the forces shaping it throughout life. Similar but more comprehensive analyses with diseased and aged cells are warranted to provide more specific evidence of how mtDNA mutations contribute to phenotypic changes and disease development.

## Online content

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

## Methods

### Sample cohort

Publicly available WGSs of single clones from four previous datasets were used—one for colorectal epithelium from our previous study (405 normal clones and 19 matched colorectal carcinomas from 19 individuals, 12 MUTYH-associated adenomatous clones from 1 individual)[6], one for mesenchymal fibroblasts from our previous study (334 normal clones from 7 individuals)[5] and two for HSPCs (1,331 clones from 4 individuals)[7,8]. WGSs of colorectal epithelium were established from single-crypt-derived organoids, and the others were generated from single-cell expanded clones. We included only clones with >0.4 VAF and >10 average depths in the nuclear genome to ensure clonality and quality.

To deeply understand the mtDNA heteroplasmy in early embryogenesis, we established 26 single-crypt-derived organoids of colorectal epithelium from one 19-week-old aborted fetus, as previously reported[6]. Genomic DNA materials were extracted using the DNeasy Blood and Tissue Kit (Qiagen). DNA libraries were generated using TruSeq DNA PCR-Free Library Prep Kits (Illumina) and sequenced on the NovaSeq 6000 platform. All the procedures in this study were approved by the Institutional Review Board of Korea Advanced Institute of Science and Technology (approval: KH2021-096), and informed consent was obtained from the parents of this individual.

In addition, to assess the prevalence of mtDNA variants across normal clones, we included 52 in-house WGSs of 13 individuals generated from organoids of various tissue types and 432 WGSs of 42 individuals generated from LCM patches of colorectal tissues[72]. To validate heteroplasmy profiles in the fertilized egg, we further explored 938 WGSs of bulk blood from 275 families[41]. To confirm that the observed VAF in bulk tissues matches the VAF in the fertilized egg, we examined 108 in-house WGSs obtained from cord blood and buccal swabs of 19 monozygotic twin families.

To understand the association between gene expression and mtDNA mutations, 312 whole-transcriptome sequences of colon clones from our previous study were also included[6].

To compare mtDNA mutations between tumor and normal samples, we further explored 51 WGSs of colorectal carcinomas from the Pan-Cancer Analysis of Whole-Genome Consortium of the International Cancer Genome Consortium (ICGC) and The Cancer Genome Atlas (TCGA)[69]. Findings from colorectal cancer genome sequences, including tumor cell fraction, tumor ploidy and driver mutations, were analyzed by CancerVision (Inocras)[6,73]. Additionally, we used 17 in-house WGSs to compare the mtDNA copy numbers of normal colon cells, T cells and colorectal cancer cells. For T cells ($n = 3$), we used samples obtained by sorting and clustering T cells followed by bulk sequencing. For colorectal cancer cells ($n = 14$), we sequenced colorectal cancer organoids with a tumor cell fraction of 1 to ascertain the mtDNA copy number exclusively from cancer cells.

### Calling and filtering of mtDNA mutations

Sequenced reads were aligned to the human reference genome build 37 (GRCh37) using the BWA-MEM algorithm[74]. Duplicated reads were removed by Picard (available at https://broadinstitute.github.io/picard/), and reads mapped to the mitochondrial genome were extracted by SAMtools[75]. To be aware of misaligned reads due to nuclear-mitochondrial DNA segments (NUMTs), we only included paired reads that were (1) both mapped to mtDNA, (2) not chimeric aligned and (3) correctly oriented. mtDNA mutations were called using HaplotypeCaller2 (ref. [76]) and VarScan2 (ref. [77]), and any mutation detected by either one was added to the mutation sets for high sensitivity.

Mutations were then filtered out using the following criteria: (1) low mapping quality (<25); (2) low base quality (<15); (3) skewed average mutation position (<15% or >85% of supporting reads); (4) unbalanced ratio between forward and reverse supporting reads (<10% or >90%) and (5) five or more mismatches in supporting reads. Mutations in the regions with low complexity or a gap in the reference genome (m.3,107N) were explicitly discarded[31]:

(1) Misalignment due to ACCCCCCCTCCCCC (rCRS 302-315)
(2) Misalignment due to GCACACACACACC (rCRS 513-525)
(3) Misalignment due to 3107N in rCRS (rCRS 3,105-3,109)
(4) Misalignment due to ACCCCC (rCRS 16,182-16,187)

More strict criteria were applied to InDel mutations, so mutations with a high proportion of additional InDels in supporting reads (>50%) were filtered out. Furthermore, InDels within noisy regions, primarily attributed to C homopolymers, were excluded from the analysis—m.567, 955 and 5,894. When visually inspected using Integrative Genomics Viewer[78], although long-read sequences had precise profiles, various types of InDels were detected at these loci in short-read sequences, making it challenging to identify mutations clearly.

A background noise matrix was generated for each locus of alternate alleles using all WGSs of normal clones to establish high-confidence mutation sets. Due to mtDNA's repetitive nature, background noise rates vary across loci. We systematically measured VAFs within every normal clone and constructed VAF distributions for each locus and alternate allele, considering the background noise matrix. Then, we overlaid the called variant set onto the background noise matrix.

To determine the background noise criteria, we first calculated the average and s.d. of the VAFs in each locus, computing the one-sided 95% confidence interval. Clones with VAF beyond the interval were considered mutants in the locus. In parallel, VAFs in a specific position were sorted in ascending order, and gaps between adjacent VAFs were examined. We considered the quantum jump of the gap as the cutoff value between true signals and background noises. To this end, we calculated the relative gap between adjacent VAFs as follows:

$$\text{relative gap} = \frac{\text{VAF}' - \text{VAF}}{\text{VAF}},$$

where VAF and VAF′ denote the lower and higher adjacent VAFs from an mtDNA locus. When calculating the relative gap, we only use VAFs between 0.05% and 15%. After calculating the relative gap, we identified the adjacent VAFs, which showed the largest gap among those with a relative gap of 0.33 or higher. We considered this gap the boundary between the background noises and true signals. The actual threshold value was set as a smaller value between (1) the average of VAF and VAF′ or (2) VAF × 1.33. If a variant did not exceed this threshold even after being called, it was considered a false-positive and excluded from the variant set. Conversely, if a variant was not called but exceeded this threshold, it was considered a false-negative and rescued.

### Classification of mtDNA alterations

mtDNA alterations shared in at least two clones of the same individual were classified into $\text{Het}_{FF}$ variants, $\text{PZ}_{recurrent}$ mutations and $\text{PZ}_{simple}$ mutations. If a shared mtDNA alteration was identified only in one individual, it was considered a $\text{Het}_{FF}$ variant. When an alteration was detected in two or more clones within a single individual but found in just a few clones within other individuals, the binomial test was used to provide a statistical framework for classifying the mutation, whether a $\text{Het}_{FF}$ variant or a $\text{PZ}_{simple}$ mutation. Concerning these mutations of interest, we used a maximum likelihood estimation method to estimate the probability of their occurrence by chance within each clone:

$$L\left(p|x_{exp}\right) = \binom{n}{x_{exp}} p^{x_{exp}}(1-p)^{n-x_{exp}}$$

$$\hat{P}_{ML} = \text{argmax}_p L\left(P|x_{exp}\right),$$

where $n$ denotes the total number of normal clones, excluding the individual harboring the shared mutation, and $x_{exp}$ denotes the subset of these clones where the mutation was detected. The resultant $\hat{P}_{ML}$

reflects the estimated rate at which the mutation spontaneously arises. By using a binomial distribution, we subsequently computed the probability that the shared mutation emerged by random chance within the individual:

$$P(x \geq x_{\mathrm{obs}}) = \sum_{x=x_{\mathrm{obs}}}^{n_{\mathrm{obs}}} \binom{n_{\mathrm{obs}}}{x} \hat{p}_{\mathrm{ML}}^{\;x} (1 - \hat{p}_{\mathrm{ML}})^{n_{\mathrm{obs}}-x}$$

where $n_{\mathrm{obs}}$ and $x_{\mathrm{obs}}$ represent the number of clones within the individual and the number of clones harboring the shared mutation in the individual, respectively. If the calculated probability was below 0.01, the mutation was categorized as a $\mathrm{Het_{FE}}$ variant; otherwise, it was classified as a $\mathrm{PZ_{simple}}$ mutation.

A mutation was categorized as a $\mathrm{PZ_{recurrent}}$ mutation when observed in two or more individuals, each exhibiting a minimum of two clones carrying the identical mutation. Moreover, mutations found in at least ten individuals within the cohort of 86 individuals (31 individuals in the study and 55 other individuals from in-house WGSs or WGSs of LCM patches[72]) were designated as $\mathrm{PZ_{recurrent}}$ mutations. Identifying $\mathrm{PZ_{recurrent}}$ mutations within a specific tissue was contingent upon their notable prevalence among older individuals (Wilcoxon rank-sum test) and their simultaneous occurrence in multiple individuals within the same tissue.

Mutations exclusive to late-branched clones were designated as $\mathrm{PZ_{simple}}$ mutations. Late-branched clones were defined by their common ancestor, accumulating at least 100 mutations before diverging. Particularly within the context of HSPCs, many clones exhibited branching events during postearly embryogenesis development. This necessitated a meticulous mutation assessment to avoid misclassifications as $\mathrm{Het_{FE}}$ variants.

### Fixation index of $\mathrm{Het_{FE}}$ variants

Fixation index ($F_{\mathrm{ST}}$), or Wright's $F$ statistics, is a statistical metric for quantifying genetic differentiation[79]. Using this, we computed the diversity of clone-VAF for each $\mathrm{Het_{FE}}$ variant to assess the impact of the lifetime drift effect:

$$F_{\mathrm{ST}} = \frac{\sigma_S^2}{\sigma_T^2} = \frac{\sigma_S^2}{\bar{P}(1 - \bar{P})},$$

where $\sigma_S^2$ represents the variance in clone-VAFs among clones within an individual, $\sigma_T^2$ represents the variance in clone-VAFs across the entire cells of the individual and $\bar{P}$ signifies the average clone-VAF across clones in the individual (caVAF), which approximates the average frequency in the entire cell population under the assumption of Hardy–Weinberg equilibrium. Notably, $F_{\mathrm{ST}}$ computations were exclusively conducted for $\mathrm{Het_{FE}}$ variants with $\bar{P}$ values exceeding 0.01.

### Simulation framework for the mitochondrial turnover

We developed a computational algorithm to simulate lifetime drift in mtDNA heteroplasmy. mtDNA undergoes continuous random degradation and replication processes independently of cell division, even without cell division. If cell division occurs during this process, random segregation of mtDNA accompanies it. This series of processes, whereby mtDNA undergoes 100% refreshment within a cell, is defined as mitochondrial turnover. To reflect this nature of mtDNA and intuitively mimic mitochondrial turnover, we developed the following two turnover models: the mitotic turnover model and the homeostatic turnover model.

Both models involve random replication of mtDNA copies. In the mitotic turnover model, mtDNA copies increase one by one through random replication until the original amount is doubled and then halved randomly through mitosis. We define this series of processes as one mitotic turnover. In the homeostatic turnover model, random replication is iterated whenever one mtDNA copy undergoes degradation, ensuring constant mtDNA copy numbers in a cell. When

this iteration occurs for the number of times equivalent to the total amount of mtDNA, we define it as one homeostatic turnover. Thus, while both models involve random replication, the mitotic turnover model integrates cell division. In contrast, the homeostatic turnover model focuses solely on mtDNA replacement. The scheme of two turnover models is illustrated in Extended Data Fig. 5b–e and discussed in detail in Supplementary Note 5.

### Estimation of average turnover to fixation and turnover rate

We simulated how $\mathrm{Het_{FE}}$ variants change as turnover repeats by lifetime drift. We set up wild-type and mutant mtDNA to co-exist within a single cell. Then, using the mitotic and homeostatic turnover models, we aimed to infer the average turnover count until fixation into a clone-VAF of 100% (homoplasmy) and the turnover rate.

While specific parameters such as mtDNA copy number ($n$) and VAF in the fertilized egg ($P$; caVAF) differ depending on the inference being made, the foundational structure of the model remains unchanged. The central assumptions of this model encompass (1) the constancy of the average mtDNA copy number within a cell per turnover, (2) the neutrality of mutant mtDNA about selection and (3) simultaneous turnovers occurring in 10,000 cells. In the case of the mitotic turnover model, one more assumption, the random segregation of mtDNA into two daughter cells with an equal amount, is added. At the commencement of the simulation, each cell initiated with a VAF of $P$, mirroring the condition observed in the fertilized egg. We conducted the mitotic and homeostatic turnover model simulations with the same simulation parameters. Each simulation persisted until 10,000 turnovers were executed, with clone-VAF per cell documented at each turnover.

To infer the average turnover count for fixation to homoplasmy, $n$ was set to 750, and $P$ spanned the range of (0.1, 0.2, ..., 0.8, 0.9). Subsequently, in each of the 10,000 individual cells, the number of turnovers required for mtDNA variants to attain homoplasmy was determined. The average number of turnovers required for fixation at each parameter $P$ was computed across all 10,000 cells.

Regarding the inference of turnover rate, we selected $\mathrm{Het_{FE}}$ variants with the caVAF exceeding 0.005 for simulation. The value of $n$ was established as the observed average mtDNA copy number for the individual, while $P$ was determined as the caVAF. Both mitotic and homeostatic turnover models underwent 10,000 simulations for each unique parameter set. In each iteration, cells were randomly selected 100 times to replicate the sequencing process, considering the specific number of clones within the individual. This yielded a total of 25 summary statistics for each simulation—the count of cells within specified clone-VAF ranges (0.5–5%, 5–10%, ..., 90–95%, 95–100%), mean clone-VAF, s.d. of clone-VAF, the proportion of cells categorized as wild type (clone-VAF < 5%), heteroplasmic (5% < clone-VAF < 90%) and homoplasmic (clone-VAF > 90%). We then compared the summary statistics derived from each simulation to the observed summary statistics. The mitotic and homeostatic turnover rates were estimated by minimizing the mean squared error (MSE). The estimation of the turnover rate was performed for each specific tissue type.

### Estimation of mtDNA mutation rate

We used the estimated turnover rate to infer the mtDNA mutation rate. The mtDNA mutation rate was inferred through simulations conducted separately for the mitotic and homeostatic turnover models. The core algorithm parallels the model elucidated earlier but is oriented toward simulating $\mathrm{PZ_{simple}}$ mutations. This model comprises the following three key parameters: mtDNA copy number ($n$), total turnover count ($g$) and mutation rate ($r$). For each individual, this simulation was performed with $n$ set to the individual's observed average mtDNA copy number and $g$ calculated as the product of the estimated tissue-specific turnover rate and the individual's age. The tissue-specific turnover rate for the

parameter $g$ is determined based on whether this simulation corresponds to the mitotic or homeostatic turnover models. The logarithm (base 10) of $r$ was sampled from a uniform distribution spanning the range of $(-9, -3)$, corresponding to a minimum $r$ of $1 \times 10^{-9}$ mutations per bp and a maximum $r$ of $1 \times 10^{-3}$ mutations per bp. The number of mutations occurring within a cell was drawn from a Poisson distribution with a parameter lambda set to $r \times 16{,}569$ for each replication event. These mutations were then introduced into the mtDNA of each cell. Following $g$ turnovers, cells were randomly chosen ten times to simulate the sequencing process, considering the specific number of clones within the individual.

This entire simulation was iterated 1,000 times, resulting in 10,000 simulation outcomes per individual. For each simulation, a set of 22 summary statistics was generated—the count of cells with the maximum clone-VAF falling within specified clone-VAF ranges (0.5–5%, 5–10%, …, 90–95%, 95–100%), the count of cells with two homoplasmic mutations (clone-VAF > 90%) and the count of cells with three homoplasmic mutations (clone-VAF > 90%). Subsequently, the summary statistics from each simulation were compared to the observed summary statistics. The mutation rates of each mitotic and homeostatic turnover were estimated by minimizing the MSE.

## Modeling the mtDNA dynamics

We conducted simulations in the mitotic and homeostatic turnover models to explore how mtDNA and heteroplasmy levels of $PZ_{simple}$ mutations change due to lifetime drift. Initially, we determined the mtDNA copy number ($n$) in a single cell and labeled each $n$ mtDNA molecule differently to allow for individual tracking. Subsequently, we simulated turnovers, recording the changes in the frequency of each mtDNA with each turnover. We then identified the most abundant mtDNA frequency in a cell. This process was simultaneously performed in a total of 10,000 cells, and the distributions were plotted for each turnover based on the most abundant mtDNA frequency in each cell.

Then, we induced $PZ_{simple}$ mutations in the abovementioned process using the mtDNA mutation rate we inferred. Subsequently, we tracked how the clone-VAF of $PZ_{simple}$ mutations changed with each turnover and determined the highest clone-VAF of $PZ_{simple}$ mutations in a cell. This process was also conducted simultaneously in 10,000 cells, and distributions were plotted for each turnover based on the highest $PZ_{simple}$ mutation clone-VAF in each cell.

Each simulation was conducted using five different mtDNA copy numbers (500, 750, 1,000, 1,500 and 2,000) in two turnover models—the mitotic and homeostatic turnover models.

## Selective pressure on mtDNA

mtDNA's evolutionary history in germline cells exhibits a notable bias toward missense mutations[31]. We aimed to compute the dN/dS ratio for each individual's unique mtDNA sequence and probabilistically assess the likelihood of these mutations occurring randomly[80,81].

To this end, we simulated the null neutrality hypothesis by introducing random mutations into individual-specific mtDNA sequences. These sequences accounted for the interindividual variability in germline mutations. We then quantified the possible occurrence of synonymous, missense and truncating mutations within each mtDNA. Using computational simulation, we generated random mutations with the exact mutation count in the individual's observed data and then annotated the functional consequences to calculate the simulated dN/dS ratio[60–62]. This simulation was iterated 10,000 times across all individuals, yielding a null distribution of dN/dS values for each individual under the neutrality assumption. These outcomes were further aggregated for individuals within the same tissue types. Ultimately, we compared the dN/dS ratios observed for missense and truncating mutations to the null distribution of simulated dN/dS values for each tissue type. We subsequently evaluated the probability of the observed dN/dS ratios occurring by chance.

## Inference of mtDNA copy number

We estimated the mtDNA copy numbers per clone using the below formula:

$$\text{mtDNA copy number} = \frac{\text{coverage}_{mtDNA}}{\text{coverage}_{nDNA}} \times 2,$$

where $\text{coverage}_{mtDNA}$ and $\text{coverage}_{nDNA}$ denote the mean coverage depth of mtDNA and the mean coverage depth of nDNA, respectively. The ploidy was fixed at two regardless of normal clone or cancer tissue to obtain more reliable mtDNA copy numbers regardless of tumor cell fraction and ploidy values. The mean mtDNA and nDNA coverage depths were computed using mosdepth[82] and an in-house script.

## Statistics and reproducibility

No statistical method was used to predetermine the sample size. No data were excluded from the analyses. The experiments were not randomized. The investigators were not blinded to allocation during experiments and outcome assessment.

## Reporting summary

Further information on research design is available in the Nature Portfolio Reporting Summary linked to this article.

## Data availability

Whole-genome sequencing data used in the study are publicly available[5–8] at the European Genome-Phenome Archive (EGA) with accessions EGAD00001007032, EGAD00001010183, EGAD00001004086 and EGAD00001007851. Whole-genome sequencing data of normal colorectal epithelium and fibroblast clones, extracted from the mitochondrial genome, are deposited in the EGA with accession EGAS50000000254 and available for general research use. The base substitutions and InDels identified in the mtDNA are available in Supplementary Table 4. The human reference genome, GRCh37, is available at https://www.ncbi.nlm.nih.gov/data-hub/genome/GCF_000001405.13. Source data are provided with this paper.

## Code availability

The open-source software and tools used in this study are detailed in Methods. Custom scripts for simulations and figures, written in Python (v3.7.0) and R (v4.1.3), are available on GitHub at https://github.com/jisong-an/mtDNA_mosaicism.

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

## Acknowledgements

This work was supported by the Suh Kyungbae Foundation (SUHF-18010082 to Y.S.J.) and the National Research Foundation of Korea funded by the Korean Government (NRF-2022R1A5A102641311 and Leading Researcher Program NRF-2020R1A3B2078973 (both to Y.S.J.)).

## Author contributions

J.A. and Y.S.J. conceived the study. H.W.K., J.W.O., Y.L. and H.W. developed the entire protocol of clonal expansion of a single cell and conducted experiments. J.-H.K., J.K.J., E.-C.S., B.K., Y.J.C., J.Y.P. and M.J.K. collected samples and clinical histories from patients. J.A. conducted most genome and statistical analysis, with contributions from C.H.N., R.K., S.P., W.H.L., H.P., C.J.Y., Y.A., J.M.B. and Y.S.J. J.A. and Y.S.J. wrote the manuscript with contributions from all authors. Y.S.J. supervised the overall study.

## Competing interests

Y.S.J. is a cofounder and chief executive officer of Inocras. The remaining authors declare no competing interests.

## Additional information

**Extended data** is available for this paper at https://doi.org/10.1038/s41588-024-01838-z.

**Correspondence and requests for materials** should be addressed to Young Seok Ju.

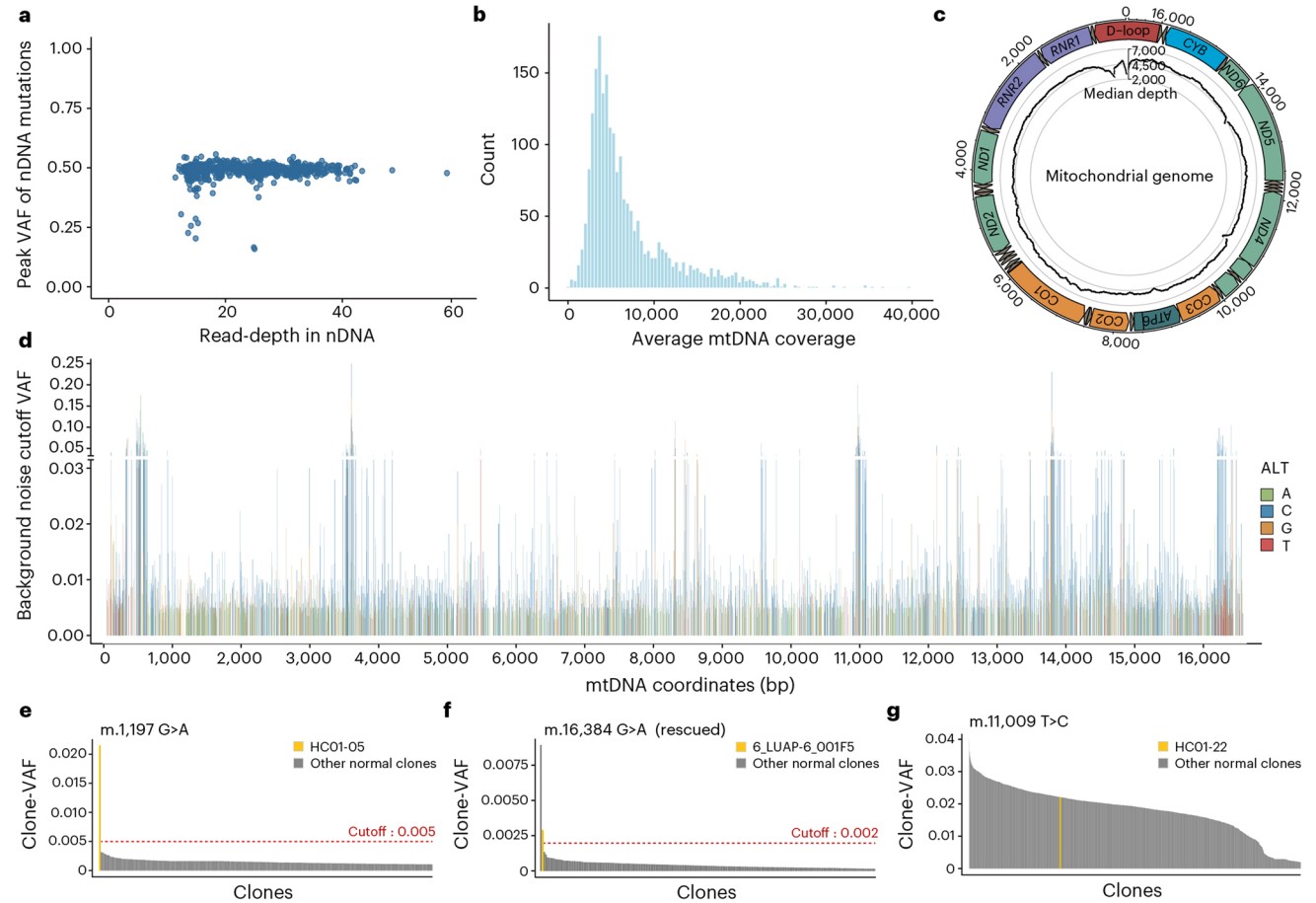

**Extended Data Fig. 1 | Coverage of mtDNA and background noise matrix.**
**a**, A scatter plot showing each clone's mean sequencing coverage and peak VAF of somatic nDNA mutations. **b**, Histogram depicting the average mtDNA coverage in 2,096 normal clones. **c**, Median read-depth per mtDNA locus across 2,096 normal clones. **d**, A bar graph showing cumulative background noise cutoff for each mtDNA locus. **e**–**g**, Examples of background noise matrix. Clone-VAFs for each mutation in normal clones are sorted in descending order. The yellow bar denotes the clone-VAF of the called mutation in a clone, and the gray bars indicate clone-VAFs of the same mutation in other clones. **e**, An example of a true-positive call (m.1,197 G>A). **f**, An example of a rescued variant (m.16,384 G>A). **g**, An example of a false-positive call (m.11,009 T>C). ALT, alternate allele.

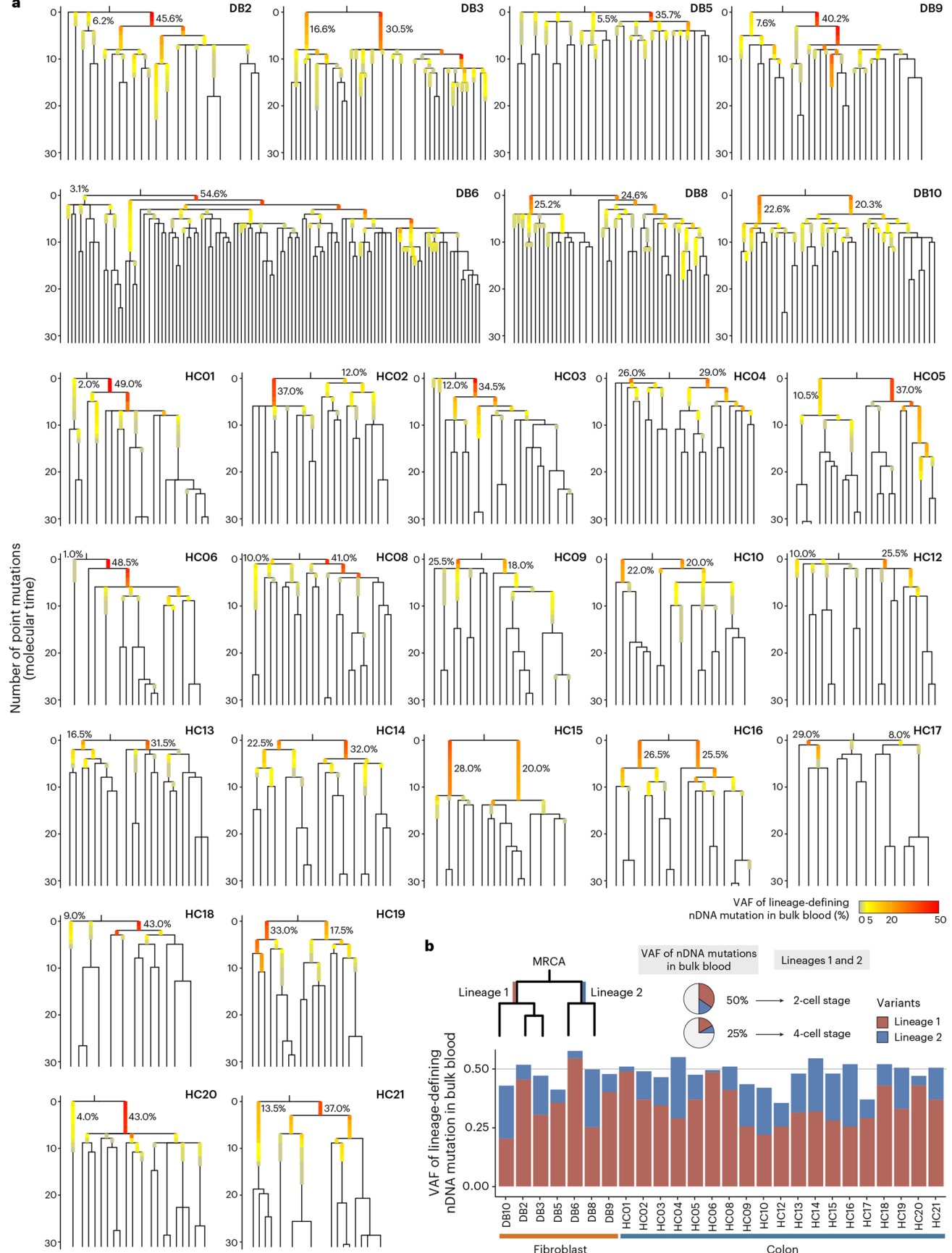

**Extended Data Fig. 2 | Phylogenetic trees annotated with VAF in bulk blood.**
**a**, Phylogenetic trees of 26 donors colored by the VAFs of lineage-defining EEMs in bulk blood tissues. The average VAFs of EEMs constituting the first two branches diverging from the MRCA (referred to as lineage 1 and lineage 2) for each phylogeny were shown. The phylogenetic trees display up to 30 EEMs, and each donor's name is shown at the top right of each panel. **b**, Cumulative bar graphs showing the average VAFs of lineages 1 and 2 EEMs in bulk blood.

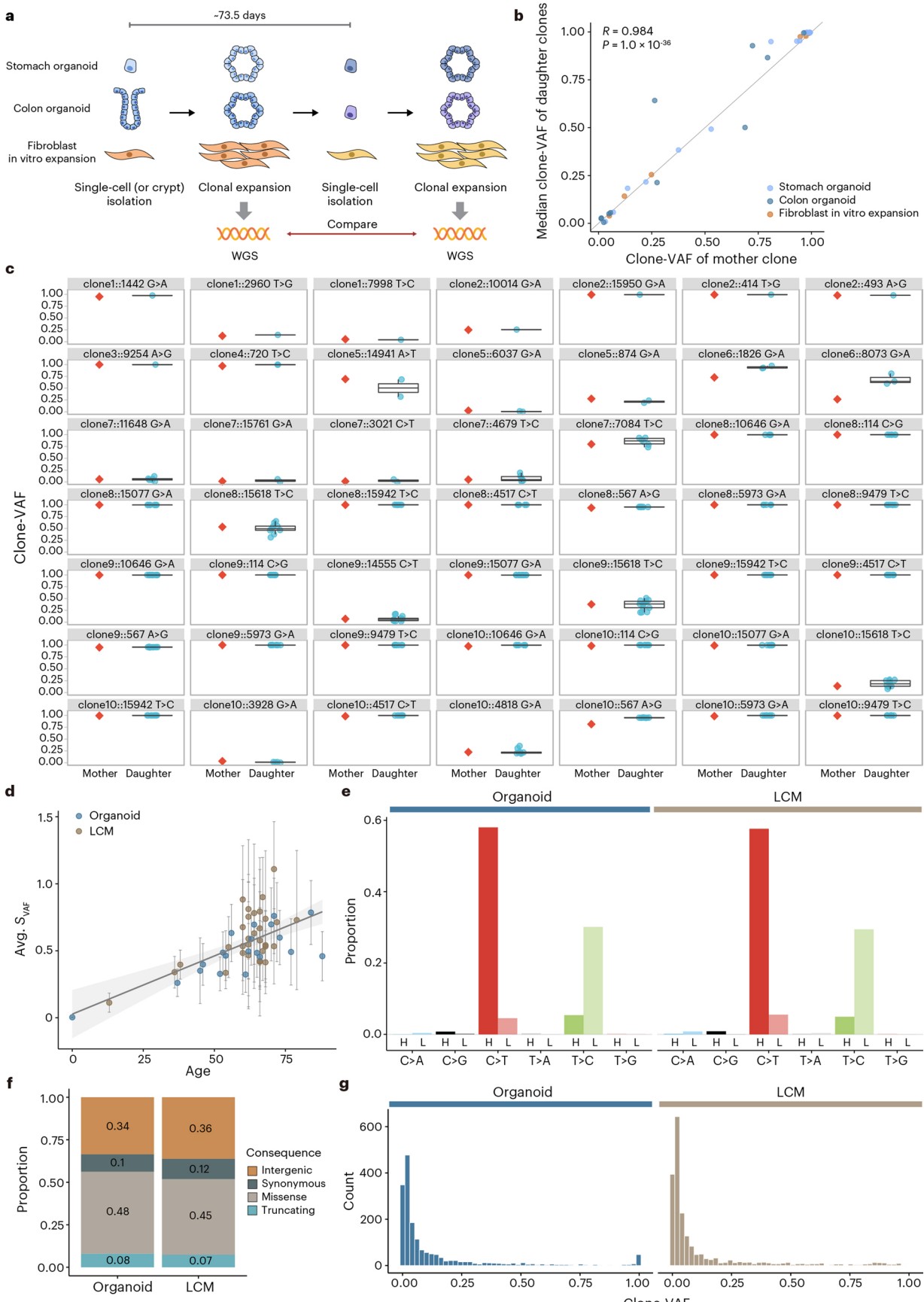

**Extended Data Fig. 3 | See next page for caption.**

**Extended Data Fig. 3 | Culture-associated events in mtDNA. a**, Experimental design for assessing culture-associated events[5,6]. A total of 47 pairs of mother–daughter clones were obtained from ten mother clones. **b**, Linear correlation between clone-VAFs of mother clones and median clone-VAFs of daughter clones. The gray line represents the diagonal line $y = x$. Pearson's correlation coefficient and $P$ value are provided. Two-sided Pearson's correlation. **c**, Comparison of clone-VAFs between mother and daughter clones for 49 variants. The mother clone and variants are provided for each panel. Boxplots illustrate median values with IQRs and whiskers (1.5× IQRs). **d**–**g**, Comparison between 431 clones from single-crypt-derived organoids (20 individuals) and 432 patches from colon crypts obtained via LCM (42 individuals)[72], including linear correlation between age and average $S_{VAF}$ of clones from the individual (**d**), mutational spectrum (**e**), the proportion of mutations classified based on functional consequences (**f**) and clone-VAF distribution (**g**). **d**, The gray line and shaded area represent the regression line and its 95% confidence interval. Vertical lines indicate the range of $S_{VAF}$ per clone in an individual.

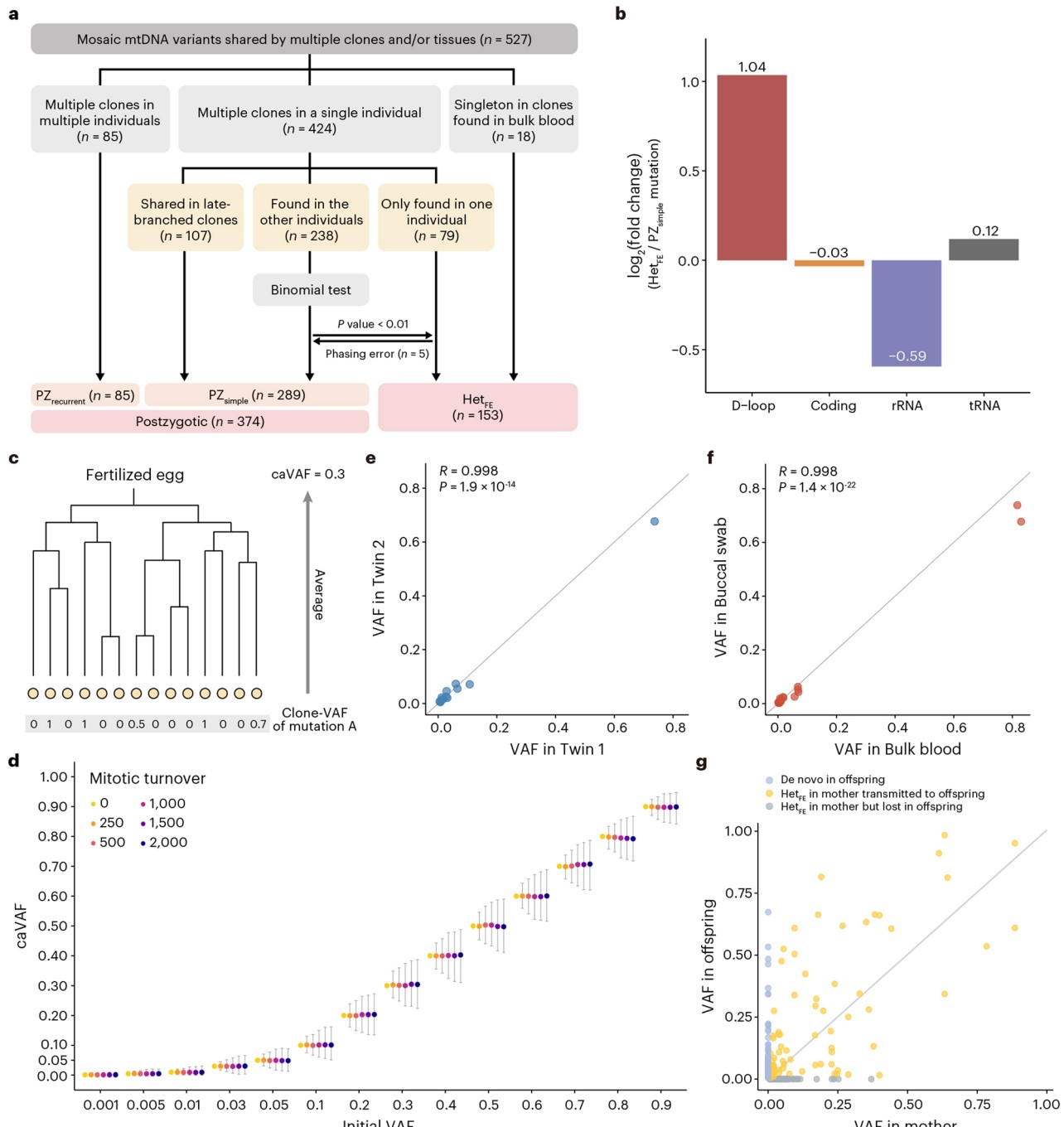

**Extended Data Fig. 4 | Features of Het$_{FE}$ variants and caVAF. a**, A diagram showing the classification of shared mtDNA alterations, with respective variant counts in parentheses. **b**, The region preference of Het$_{FE}$ variants to PZ$_{simple}$ mutations. The *y* axis indicates the log$_2$-transformed ratio of the prevalence of the Het$_{FE}$ variant to PZ$_{simple}$ mutation. **c**, A schematic diagram illustrating caVAF calculation using clone-VAFs within an individual. The gray box contains clone-VAFs of the Het$_{FE}$ variant. **d**, The relationship between initial VAF and caVAF from computational simulations. Simulated caVAFs were calculated using 100 simulated cells, with 1,000 iterations per mitotic turnover and initial VAF. Circles and vertical lines denote average caVAF values and 95% confidence intervals. **e,f**, Scatter plots comparing VAFs from bulk tissues between monozygotic twins (**e**) and within one twin (**f**). The gray lines represent the diagonal line *y* = *x*. Pearson's correlation coefficient and *P* value are presented. Two-sided Pearson's correlation. **g**, A scatter plot depicting the VAF of mtDNA variants in 407 mother–offspring pairs. The gray lines represent the diagonal line *y* = *x*.

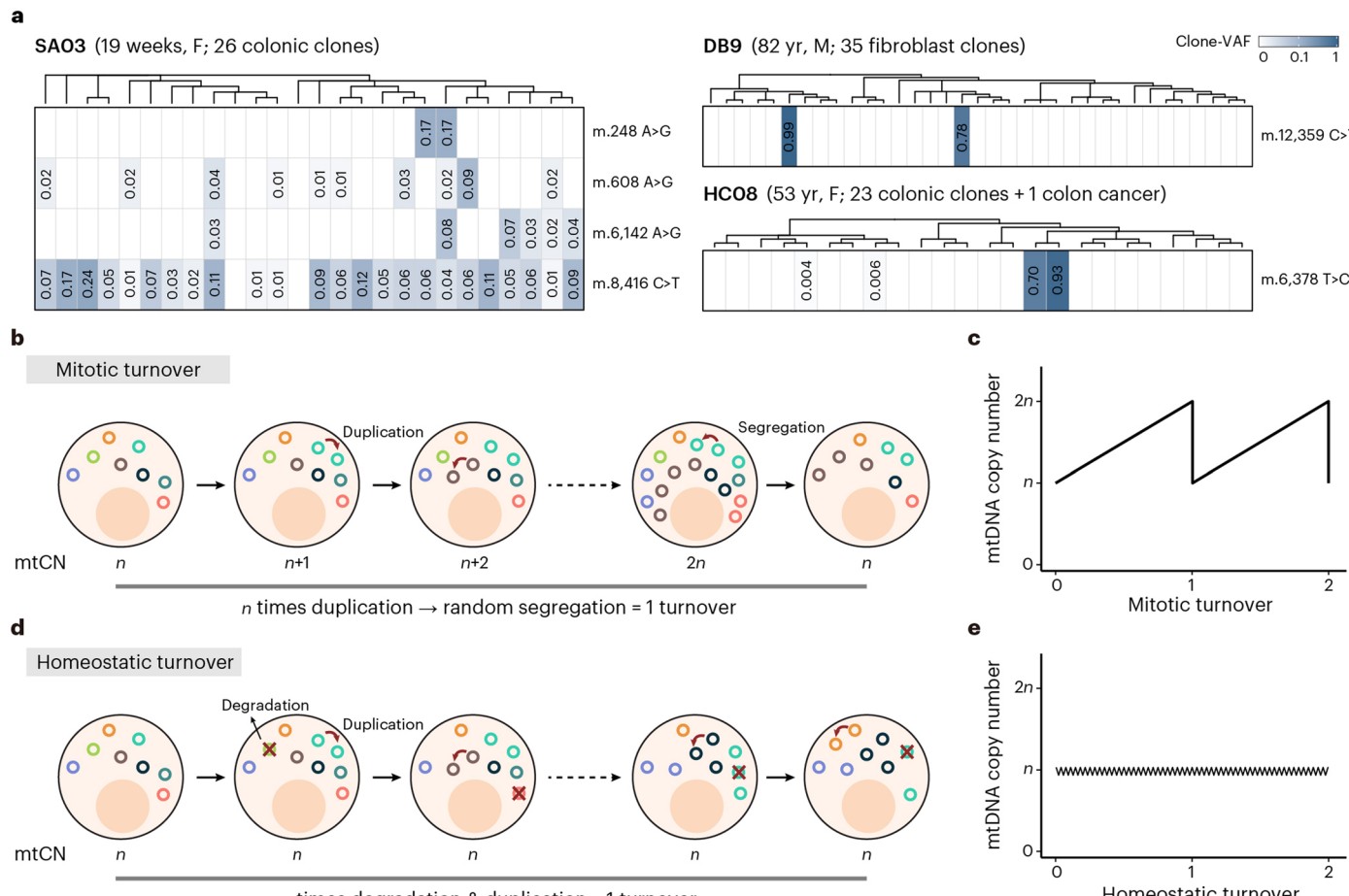

**Extended Data Fig. 5 | Clone-VAF distribution of Het_FE variants and the overview of the turnover model. a**, Heatmaps representing clone-VAFs of Het_FE variants in each clone for three donors. The columns correspond to individual clones, ordered according to their phylogenetic relationships. **b,c**, Schematic diagrams illustrating the concept of mitotic turnover. Examples of changes in mtDNA frequency during one turnover (**b**) and mtDNA copy number (**c**) are depicted. **d,e**, Schematic diagrams illustrating the concept of homeostatic turnover. Examples of changes in mtDNA frequency during one turnover (**d**) and changes in mtDNA copy number (**e**) are depicted.

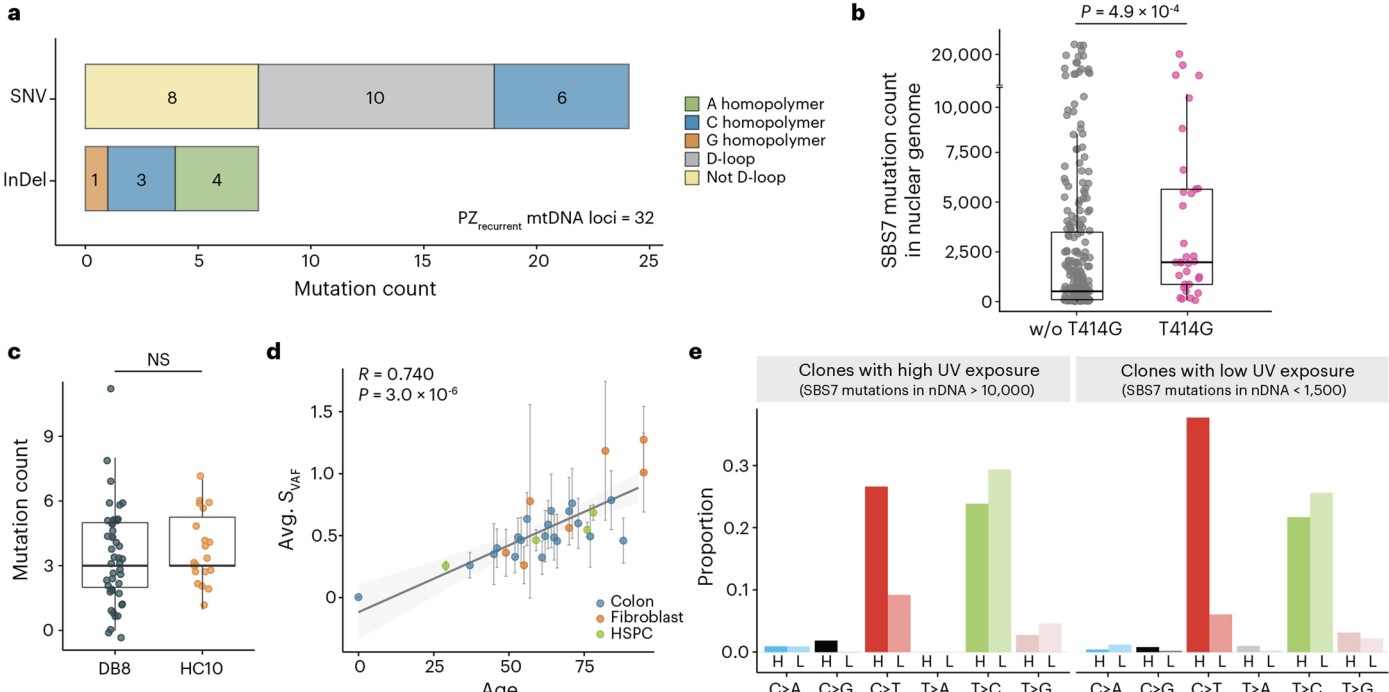

**Extended Data Fig. 6 | Mutational profiles of post-zygotic mutations.**
**a**, Categories of $PZ_{recurrent}$ mutations ($n = 32$) based on types of homopolymer and region. **b**, Boxplots illustrating the median SBS7 mutation counts in 301 clones without m.414 T>G and 33 clones harboring m.414 T>G. Boxplots illustrate median values with IQRs and whiskers (1.5× IQRs). Two-sided Wilcoxon test. **c**, Boxplots illustrating $PZ_{simple}$ mtDNA mutation count in clones of DB8 ($n = 47$; 93 years old) and HC10 ($n = 20$; 37 years old). Boxplots illustrate median values with IQRs and whiskers (1.5× IQRs). Two-sided Wilcoxon test; NS, not significant. **d**, Linear correlation between $S_{VAF}$ and age across 31 individuals. The gray

line and the shaded area represent the regression line and its 95% confidence interval. Vertical lines crossing each dot indicate the range of $S_{VAF}$ per clone in an individual. Clones with high UV exposure were excluded to remove UV radiation's impact. Pearson's correlation coefficient and $P$ value are provided. Two-sided Pearson's correlation. **e**, $PZ_{simple}$ mutation spectrum between clones with high and low UV exposure. UV exposure levels are determined by SBS7 mutation counts in the nuclear genome, with high exposure defined as over 10,000 mutations and low exposure as under 1,500 mutations. H, H strand; L, L strand.

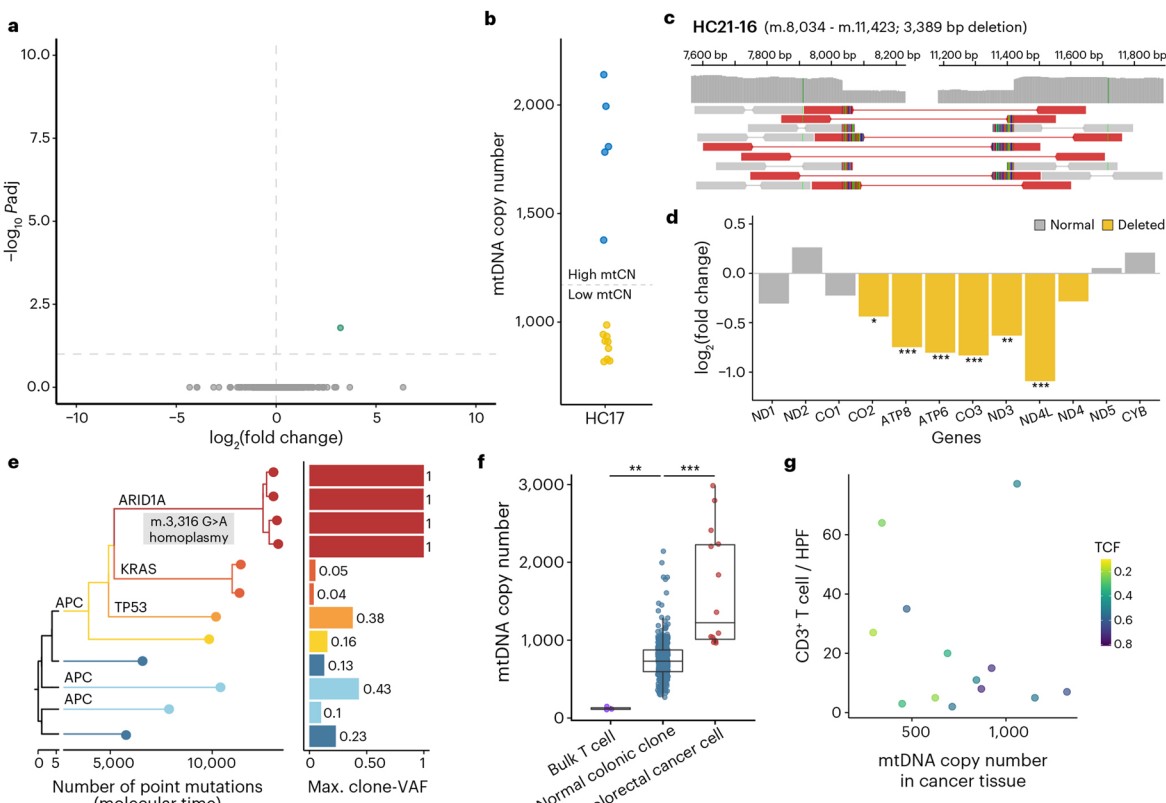

**Extended Data Fig. 7 | Characteristics of mtDNA copy number. a**, A volcano plot showing differentially expressed genes between clones with high and low mtDNA copy number (mtCN) in HC17. The *x* axis represents $\log_2$-transformed fold changes, and the *y* axis represents adjusted *P* values ($-\log_{10}$-transformed). Gray and green dots indicate nonsignificant differences and significantly downregulated genes in high mtCN clones. Two-sided Wald test with Benjamini-Hochberg correction. **b**, A dot plot of mtCNs in 14 clones of HC17, with high and low mtCNs distinguished by a gray dashed line at 1,200. **c**, A screenshot of Integrative Genomics Viewer showing mtDNA SV in HC21-16. **d**, $\log_2$-transformed fold changes comparing normalized read counts of HC21-16 with SV to other HC21 clones per mtDNA gene. Two-sided Wald test, *P < 0.05, **P < 0.01, ***P < 0.001. Exact *P* values are 0.0468, $3.5 \times 10^{-4}$, $1.4 \times 10^{-5}$, $8.5 \times 10^{-6}$, $2.3 \times 10^{-3}$, $1.9 \times 10^{-11}$, for *MT-CO2, MT-ATP8, MT-ATP6, MT-CO3, MT-ND3* and *MT-ND4L*, respectively. **e**, Phylogenetic tree of MUTYH-associated adenomatous clones with maximum clone-VAFs. **f**, Boxplots comparing mtCNs in 3 bulk T cells, 431 normal colonic clones and 14 colorectal cancer cells. Boxplots illustrate median values with IQRs and whiskers (1.5× IQRs). Two-sided Wilcoxon test, **P < 0.01, ***P < 0.001. Exact *P* values are 0.0029 for bulk T cells and $5.7 \times 10^{-12}$ for colorectal cancer cells. **g**, The relationship between mtCN per diploid nuclear genome in cancer tissue and the number of CD3$^+$ T cells. TCF, tumor cell fraction.

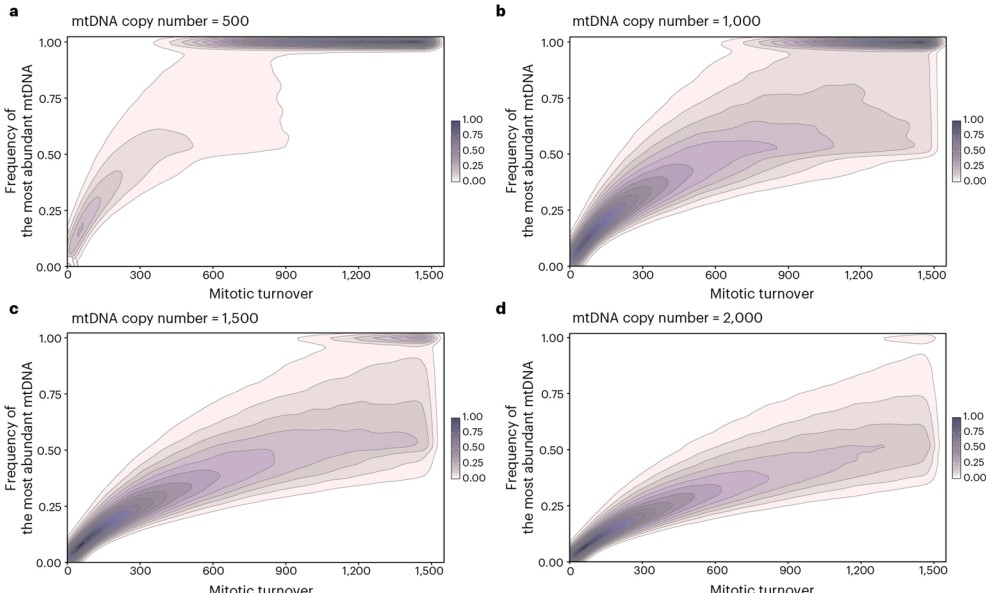

**Extended Data Fig. 8 | Model for mtDNA population change. a–d**, Contour plots representing how an mtDNA population changes with continuous mitotic turnover from simulation studies assuming four different baseline mtDNA copy numbers, including 500 (**a**), 1,000 (**b**), 1,500 (**c**) and 2,000 (**d**). The *x* axis shows the mitotic turnover count and the *y* axis shows the frequency of the most prevalent mtDNA, regardless of mutations.

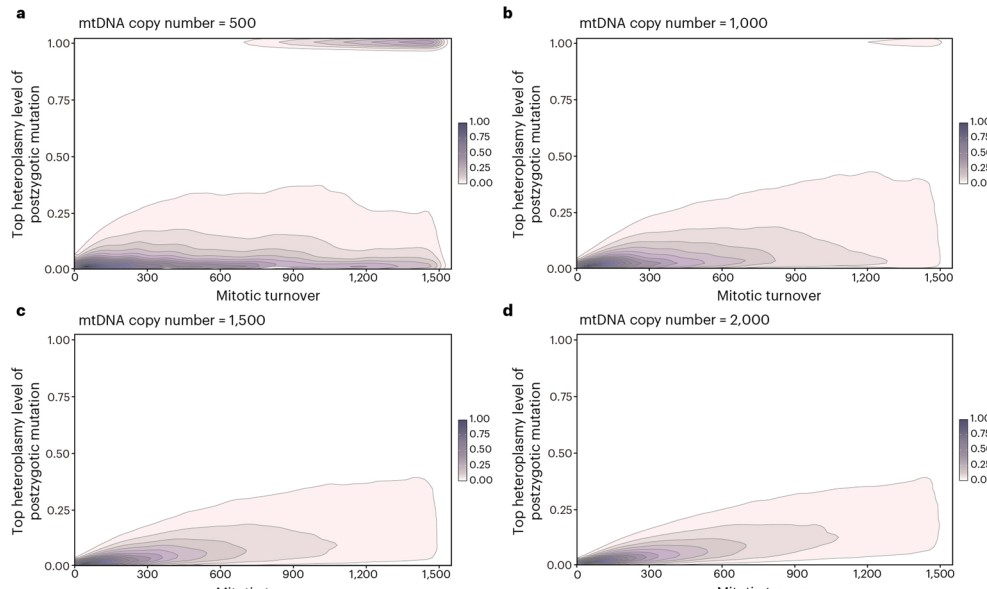

**Extended Data Fig. 9 | Model for mtDNA heteroplasmy change. a–d,** Contour plots representing how clone-VAF of $PZ_{simple}$ mtDNA mutations changes with continuous mitotic turnover from simulation studies assuming four different baseline mtDNA copy numbers, including 500 (**a**), 1,000 (**b**), 1,500 (**c**) and 2,000 (**d**) and absolute mutation rate of $5.0 \times 10^{-8}$ per bp replication. The $x$ axis shows the mitotic turnover count and the $y$ axis shows the top heteroplasmy level of postzygotic mutation.

# Reporting Summary

## Statistics

For all statistical analyses, confirm that the following items are present in the figure legend, table legend, main text, or Methods section.

| n/a | Confirmed | |
|---|---|---|
| ☐ | ☒ | The exact sample size (*n*) for each experimental group/condition, given as a discrete number and unit of measurement |
| ☐ | ☒ | A statement on whether measurements were taken from distinct samples or whether the same sample was measured repeatedly |
| ☐ | ☒ | The statistical test(s) used AND whether they are one- or two-sided *Only common tests should be described solely by name; describe more complex techniques in the Methods section.* |
| ☐ | ☒ | A description of all covariates tested |
| ☐ | ☒ | A description of any assumptions or corrections, such as tests of normality and adjustment for multiple comparisons |
| ☐ | ☒ | A full description of the statistical parameters including central tendency (e.g. means) or other basic estimates (e.g. regression coefficient) AND variation (e.g. standard deviation) or associated estimates of uncertainty (e.g. confidence intervals) |
| ☐ | ☒ | For null hypothesis testing, the test statistic (e.g. *F*, *t*, *r*) with confidence intervals, effect sizes, degrees of freedom and *P* value noted *Give P values as exact values whenever suitable.* |
| ☒ | ☐ | For Bayesian analysis, information on the choice of priors and Markov chain Monte Carlo settings |
| ☒ | ☐ | For hierarchical and complex designs, identification of the appropriate level for tests and full reporting of outcomes |
| ☐ | ☒ | Estimates of effect sizes (e.g. Cohen's *d*, Pearson's *r*), indicating how they were calculated |

*Our web collection on statistics for biologists contains articles on many of the points above.*

## Software and code

Policy information about availability of computer code

| Data collection | no software was used. |
|---|---|
| Data analysis | Sequenced reads were aligned to the human reference genome (GRCh37) using BWA (v0.7.17) algorithm. The duplicated reads were removed by Picard (v2.1.0) and mitochondrial DNA sequences were extracted by Samtools (v1.10). We identified single-nucleotide variants and short indels using Varscan2 (v2.4.2) and HaplotyperCaller2 in GATK (v4.2.0.0). Detected variants were inspected using IGV (v2.11.9). The mean coverage depths of mitochondrial and nuclear genome were computed using mosdepth (v0.3.1). Custom scripts were written by Python (v3.7.0) and R (v4.1.3) and are available at GitHub (https://github.com/jisong-an/mtDNA_mosaicism). |

For manuscripts utilizing custom algorithms or software that are central to the research but not yet described in published literature, software must be made available to editors and reviewers. We strongly encourage code deposition in a community repository (e.g. GitHub). See the Nature Portfolio guidelines for submitting code & software for further information.

# Data

Whole-genome sequencing data used in the study were publicly available at the European Genome-phenome Archive (EGA) with accession no. EGAD00001007032, EGAD00001010183, EGAD00001004086, and EGAD00001007851. Whole-genome sequencing data of normal colorectal epithelium and fibroblast clones, extracted from the mitochondrial genome, are deposited in the EGA with accession no. EGAS50000000254 and available for general research use. The base substitutions and indels identified in the mitochondrial DNA are available in Supplementary Table 4. Source data are provided with this paper. The human reference genome GRCh37 is available at https://www.ncbi.nlm.nih.gov/data-hub/genome/GCF_000001405.13.

# Research involving human participants, their data, or biological material

Policy information about studies with human participants or human data. See also policy information about sex, gender (identity/presentation), and sexual orientation and race, ethnicity and racism.

| | |
|---|---|
| Reporting on sex and gender | Sex information of individuals in this study was obtained from the original papers of public dataset. The information is summarized in Supplementary Table 1. We did not find any differences in mitochondrial DNA variants between males and females. Sex and gender were not considered in study design. |
| Reporting on race, ethnicity, or other socially relevant groupings | Out of 31 individuals, 27 individuals are Asians and 4 individuals are Europeans. Mitochondrial germline variants and mitochondrial haplogroups differed by race, but no differences were found for other mitochondrial DNA variants. Race, ethnicity, or other socially relevant groupings were not considered in study design. |
| Population characteristics | Out of 31 individuals, 19 were diagnosed with colorectal cancer. The age of the individuals spanned several age groups ranging from 0 to 93. The ratio between males and females were 0.8:1.2 (14 males and 17 females). |
| Recruitment | One individual was an aborted fetus obtained at Bundang Seoul National University Hospital. The parents of this individual, attending the Obstetrics and Gynecology department of Bundang Seoul National University Hospital, were recruited with their consent for an infant miscarried after 12 weeks of gestation. Data for the other individuals were published previously and downloaded for this study. |
| Ethics oversight | All the procedures in this study were approved by the Institutional Review Board of Korea Advanced Institute of Science and Technology (approval number: KH2021-096). |

Note that full information on the approval of the study protocol must also be provided in the manuscript.

# Field-specific reporting

Please select the one below that is the best fit for your research. If you are not sure, read the appropriate sections before making your selection.

☒ Life sciences          ☐ Behavioural & social sciences          ☐ Ecological, evolutionary & environmental sciences

For a reference copy of the document with all sections, see nature.com/documents/nr-reporting-summary-flat.pdf

# Life sciences study design

All studies must disclose on these points even when the disclosure is negative.

| | |
|---|---|
| Sample size | No statistical methods were used to predetermine the sample size. We selected samples from available individuals to describe the mitochondrial DNA mosaicism in normal cells. |
| Data exclusions | No data were excluded from the analyses. |
| Replication | We used 47 pairs of mother-daughter clones (biological replicates) to estimate the rates of culture-associated mitochondrial DNA mutations. We generated 47 daughter clones by serial clonalization of ten mother clones. All attempts at replication were successful, and almost every mtDNA variant identified in mother clones was detected in daughter clones. |
| Randomization | Not applicable since there was no predetermined group of samples. All available samples were used. |
| Blinding | Not applicable since this is a descriptive study |

# Reporting for specific materials, systems and methods

We require information from authors about some types of materials, experimental systems and methods used in many studies. Here, indicate whether each material, system or method listed is relevant to your study. If you are not sure if a list item applies to your research, read the appropriate section before selecting a response.

## Materials & experimental systems

| n/a | Involved in the study |
|-----|----------------------|
| ☒ ☐ | Antibodies |
| ☒ ☐ | Eukaryotic cell lines |
| ☒ ☐ | Palaeontology and archaeology |
| ☒ ☐ | Animals and other organisms |
| ☒ ☐ | Clinical data |
| ☒ ☐ | Dual use research of concern |
| ☒ ☐ | Plants |

## Methods

| n/a | Involved in the study |
|-----|----------------------|
| ☒ ☐ | ChIP-seq |
| ☒ ☐ | Flow cytometry |
| ☒ ☐ | MRI-based neuroimaging |

## Plants

| | |
|---|---|
| Seed stocks | *Report on the source of all seed stocks or other plant material used. If applicable, state the seed stock centre and catalogue number. If plant specimens were collected from the field, describe the collection location, date and sampling procedures.* |
| Novel plant genotypes | *Describe the methods by which all novel plant genotypes were produced. This includes those generated by transgenic approaches, gene editing, chemical/radiation-based mutagenesis and hybridization. For transgenic lines, describe the transformation method, the number of independent lines analyzed and the generation upon which experiments were performed. For gene-edited lines, describe the editor used, the endogenous sequence targeted for editing, the targeting guide RNA sequence (if applicable) and how the editor was applied.* |
| Authentication | *Describe any authentication procedures for each seed stock used or novel genotype generated. Describe any experiments used to assess the effect of a mutation and, where applicable, how potential secondary effects (e.g. second site T-DNA insertions, mosiacism, off-target gene editing) were examined.* |

