## [Peer Review File · Nature Genetics]

Peer Review Information

Manuscript Title: Mitochondrial DNA mosaicism in human normal somatic cells

Corresponding author name(s): Professor Young Seok Ju

Reviewer Comments & Decisions:

Decision Letter, initial version:

17th Jan 2024

Dear Professor Ju,

Your Article, "Mitochondrial DNA mosaicism in human normal somatic cells" has now been seen by 3 referees. You will see from their comments below that while they find your work of interest, some important points are raised. We are interested in the possibility of publishing your study in Nature Genetics, but would like to consider your response to these concerns in the form of a revised manuscript before we make a final decision on publication.

We therefore invite you to revise your manuscript taking into account all reviewer editor comments. Please highlight all changes in the manuscript text file. At this stage we will need you to upload a copy of the manuscript in MS Word .docx or similar editable format.

*2) If you have not done so already please begin to revise your manuscript so that it conforms to our Article format instructions, available here.

*3) Include a revised version of any required Reporting Summary:

Please be aware of our guidelines on digital image standards.

[redacted]

We hope to receive your revised manuscript within four to eight weeks. If you cannot send it within this time, please let us know.

Nature Genetics is committed to improving transparency in authorship. As part of our efforts in this direction, we are now requesting that all authors identified as 'corresponding author' on published papers create and link their Open Researcher and Contributor Identifier (ORCID) with their account on the Manuscript Tracking System (MTS), prior to acceptance. ORCID helps the scientific community achieve unambiguous attribution of all scholarly contributions. You can create and link your ORCID from the home page of the MTS by clicking on 'Modify my Springer Nature account'. For more information please visit please visit www.springernature.com/orcid.

Sincerely,

Safia Danovi
Editor
Nature Genetics

Referee expertise:

Referee #1: somatic evolution

Referee #2: mtDNA

Referee #3: mtDNA

Reviewers' Comments:

Reviewer #1:

Remarks to the Author:

In their manuscript "Mitochondrial DNA mosaicism in human somatic cells," An and colleagues analyze sequencing data from single-cell derived colonies from multiple tissues to identify mtDNA variants and place them along developmental phylogenies. The authors describe substantial and pervasive heteroplasmy in the fertilized egg, explore the evolution of heteroplasmic variants in the soma, investigate the origins of de novo mtDNA mutations (mostly replication) and study how their frequencies are shaped by selective pressures. Finally, they also quantify mitochondrial turnover and copy number in normal and malignant cells.

This manuscript is interesting and rigorously constructed. It uses state-of-the-art experimental and computational methods in somatic genetics to explore how mutations arise in mitochondria and how they are propagated during cell turnover. The insights that are presented here, based on extremely high-quality sequencing data from many single cells derived from different individuals, are very valuable and I think they will be very useful for everyone who is interested in mitochondria and their somatic evolution. Much of the writing is a speculation-free enumeration of the observed quantities, so I feel that there really isn't much to argue with. Below I have a few questions and also a suggestion to make the paper more user-friendly, but in general I think that the manuscript requires only minor revisions.

- I was surprised by how similar the inferred turnover per year was across tissues (Fig. 3i). Colonic stem cells should be dividing a lot more frequently than HSCs, and fibroblasts should be very slow. The authors suggest in line 277 that this could be due to mitochondrial turnover independently of cell proliferation, but if that is a serious possibility, I feel that it would be good to model this separately from the Fisher Wright model which couples the mtDNA duplication with cell division. If major mtDNA turnover happens in the absence of cell division, then the Fisher Wright model and its presented results may not adequately represent the underlying process and this should be clarified in the text.
- Given that we now know that many nuclear genomic mutations occur independently of cell division as a function of time (e.g. SBS1 mutations in neurons), is there a reason to report the mtDNA mutation rate exclusively as "per turnover" as opposed to "per year"? In particular, given that the turnover estimate is a bit of a question (see point above)?
- Is the plot in Figure 4d somewhat circular, given that turnover (x-axis) is estimated by heteroplasmy levels and Svaf is a related measure?
- Since the measurements in Figure 4c are linked by common ancestry, I am not sure it is fair to treat them as independent observations in a Fisher exact test (if I understand correctly what the authors are doing).
- Figure 5a is a bit misleading visually as it almost looks like the mutation rate "precedes" the turnover rate in time, perhaps this could be redesigned.
- I do not understand the argument in lines 244-251. Perhaps this part could be rewritten for clarity (although it could also be my own personal failure unrelated to the text)
- The finding of extreme mtDNA copy numbers in cancer is very interesting but the authors do not interpret it in any way. Perhaps some interpretation (even if it was a bit on the speculative side) would be helpful.
- The paper would benefit from text revisions to make it more readable. First, the order of topics is not entirely understandable to me (e.g. the description of HetFE variants and the description of their

somatic evolution is interrupted by a section on hotspot mutations), some restructuring may improve the text flow. The writing is very dense and technical, enumerating many statistics in most paragraphs (e.g. lines 150-156 and many other examples), but I am not sure that this is necessary to convey precise points. Some of the numerical details could presumably be presented in a supplementary note or table. Instead, it might be useful to focus a little bit more on explanation and interpretation of the results in the main text. There are some language issues which I think somewhat obfuscate the precise meaning of several important conclusions outlined in the text. Re-review after text editing might be helpful.

Reviewer #2:

Remarks to the Author:

In the manuscript by An and colleagues, the authors investigate patterns of somatic mtDNA mosaicism through an impressive whole genome sequencing experiment covering over two thousand clones. The authors make a number of observations, some of which relate to the evolutionary origin of mtDNA mutations within an individual back to the developing embryo. The manuscript is expansive and covers significant ground. While I find some of the results exciting, there are some results which have been described previously in the literature and which may (in my subjective opinion) distract from the key messages of the manuscript. I detail my comments below, separating according to the thematic nature of the feedback.

Broad comments:

1. While I find the dataset that the authors produce impressive, I do not think the authors did a strong job communicating to the reader what the major gap in knowledge is that could be addressed by this new data. The evolutionary origins of somatic mtDNA mosaicism are to some extent covered by earlier work of a different flavor, e.g. in Wei...Chinnery Science 2019, where the transmission of heteroplasmies was investigated. So, I would ask the authors directly: what is the major gap in knowledge that this work addresses? And what is the conceptual advance that the reader should walk away with, how should this work make us see the world differently?
2. It felt several times throughout the manuscript as if key concepts were not clearly communicated or visualized. The most critical example is the precise definitions of the different origins of mtDNA alterations on line 150, i.e. heteroplasmies from a fertilized egg, recurrently acquired hotspot mutations (which I think are coincidental mutations?) on lines 150-159. This is the bedrock of the remainder of the paper, why not make a simple figure illustrating the point?
3. Could the authors elaborate on how they draw the conclusion that mtDNA mutations observed across all or most clones must necessarily derive from the fertilized egg? If the sampled tissues are colorectal epithelium, fibroblast, and HSC (and matched blood potentially, if I understand correctly), what is the most recent common ancestor of those cell types? How late in development is it?
4. Related to the above, but now in a more speculative mode (I don't expect the authors to be able to address this question, but if they can it would be nice to add to the manuscript): is there evidence to conclude whether the Het_FE variants are predominantly de novo or alternatively transmitted from the mother?
5. In general, there are a number of findings in this manuscript which seem to have been reported in prior work. In some cases, the authors go further than the prior work, but in other cases it is less clear. Some examples which could be clarified: (a) the existence of heteroplasmic variants in the early embryo, which is similar to the cited study from the UK Biobank (ref 50, line 198); (b) the description of truncating variants in colorectal cancer, (c) differences in RNA and DNA VAFs in certain tRNA

mutations and their implications for tRNA processing

6. Perhaps I missed it, but I saw neither a link to the code necessary for reproducing the analysis in this manuscript, nor (more importantly) the data. Data of this scale would be exceptionally useful for the field at large, and it would be fantastic for the authors to not only deposit the raw data in a suitable repository, but also to provide processed mutation calls as Supplementary Data (or placed in a repository).

Specific But Major Comments:

1. Line 187: The statement or assumption that the caVAFs of the HetFE variants correspond to heteroplasmy in the first cell of life feels like pure speculation. Is there any way to corroborate this? The line of reasoning the authors use is plausible, but so are many others. Can some mathematical modeling (e.g. Wright-Fisher) help?
2. To what extent do the hotspot variants the authors describe overlap with those reported in cancer, i.e. at specific homopolymeric loci in Complex I genes? Does the overlap or lack thereof imply something about evolutionary origins?
3. Line 249: "This indicates that the size of the early mtDNA bottleneck was much higher than 1." I'm less familiar with this aspect of mtDNA selection, can the authors elaborate further (it can be in the Methods) as to what a bottleneck size corresponds to (the units?) and provide some intuition?
4. Regarding the SVAF calculations around lines 280-296, I'm confused about the difference between SVAFs in old and young individuals. While there is not a statistically significant association between age and sporadic mutations, it seems like this is still confounded. Is it possible to control for this in a more principled analysis using a linear model with number of sporadic mutations as a covariate?
5. Figure 2f: I found this confusing and was hoping the authors could clarify. The figure annotates m.12417 as a hotspot but in the figure legend it is labeled as HetFE. Not sure what I am looking at, and how this shows a phasing error.
6. I'm having trouble reconciling two ideas about the clone-averaged VAF (caVAF): early in the manuscript, it is used to summarize a smooth unimodal distribution of heteroplasmies across the cells of a clone. But later in the manuscript, it is revealed that sometimes caVAF summarizes a variant which has been pushed to extinction (0% VAF) or homoplasmy (100% VAF). Is the caVAF then useful?
7. Figure 5A: What do the colors represent?
8. Line 411: "...this is the first comprehensive study to reveal mtDNA heteroplasmy in fertilized eggs." This is a vast overstatement, there is no direct profiling of fertilized eggs.

Methodological comments (mostly related to Methods: "Calling and filtering mtDNA mutations")

1. How the locus-specific background noise matrix is constructed is not clear to me. It is mentioned that it is locus specific but based on the supplementary table it appears to me position and alternate allele specific so I would specify this. How do you test specifically for a variant to be "significantly greater than the background noise"?
2. On line 863 it is stated that "any mutations that were not called initially were rescued. Do you mean that if a mutation was not called using HaplotypeCaller2 or VarScan2 but its VAF was higher than the background noise then it was considered a variant? I am curious as to how many mutations were detected in this fashion and the justification of using these when they were not called by neither HaplotypeCaller2 nor VarScan2.

Reviewer #3:

Remarks to the Author:

This manuscript from An and colleagues describes an interesting and unusual approach to studying mitochondrial DNA mosaicism in human somatic mitotic cells. The paper is understandable, but would benefit from reordering and rewriting for lucidity and communicability.

The authors use data from 'clones' mostly derived from healthy tissues (with a few colorectal polyps and carcinomas included), which were cultured, sequenced and the data analysed.

There are a few fundamental issues with the approach the authors conclusions are based upon. Chief among these is the diversity of ways in which the 'clones' were derived and/or cultured. Assuming that single crypt organoid derivation vs hsc culture vs expansion of cancer cells will demonstrate similar selective pressures to one another or to the pressures present in situ is, at best, hopeful. It is widely known that culture conditions alone in 2D monoculture of cytoplasmic hybrids can induce dramatic shifts in heteroplasmy. As such, the unknown selective pressures being applied by the authors choice of sampling weakens the foundation of their conclusions.

A further conceptual issue for me is the assumption that shared variants from an individual must result from such an early stage as egg fertilisation. If e.g. colonic crypts are being derived from a small area of the gut, the authors must also consider it likely that clonal competition and expansion over time is at least as likely a reason for shared variants as these being directly derived from the earliest stage of development. A similar comment could be made of the HSC data presented. Accepting this reality also weakens the foundation of the conclusions, which while still interesting, are not as far-reaching as the authors are suggesting at present.

Minor issues include:

The description of POLG as an error prone polymerase - POLG is the highest fidelity polymerase in humans.

The description of the mtDNA bottleneck occurring during embryo cleavage - this is part of the bottleneck, however the massive expansion of mtDNA copies from primordial germ cell to immature/mature oocyte represents a major part of the bottleneck process.

For clarity, despite the shortcomings I have illustrated and while many of the results presented here have been described elsewhere in other settings, I believe some of the data here are of interest and represent novel findings. However the authors need to robustly address the issues raised for the study to be convincing.

Author Rebuttal to Initial comments

Reviewers' Comments:

Reviewer #1:

Comment 1-1. In their manuscript "Mitochondrial DNA mosaicism in human somatic cells," An and colleagues analyze sequencing data from single-cell derived colonies from multiple tissues to identify

mtDNA variants and place them along developmental phylogenies. The authors describe substantial and pervasive heteroplasmy in the fertilized egg, explore the evolution of heteroplasmic variants in the soma, investigate the origins of de novo mtDNA mutations (mostly replication) and study how their frequencies are shaped by selective pressures. Finally, they also quantify mitochondrial turnover and copy number in normal and malignant cells.

Answer 1-1. It is an excellent summary of our work, and we appreciate the reviewer's time and effort in reviewing our manuscript and providing constructive comments.

Comment 1-2. This manuscript is interesting and rigorously constructed. It uses state-of-the-art experimental and computational methods in somatic genetics to explore how mutations arise in mitochondria and how they are propagated during cell turnover. The insights that are presented here, based on extremely high-quality sequencing data from many single cells derived from different individuals, are very valuable and I think they will be very useful for everyone who is interested in mitochondria and their somatic evolution. Much of the writing is a speculation-free enumeration of the observed quantities, so I feel that there really isn't much to argue with. Below I have a few questions and also a suggestion to make the paper more user-friendly, but in general I think that the manuscript requires only minor revisions.

Answer 1-2. We appreciate the reviewer for the positive comments on our manuscript. To abide by all the reviewers' comments, we have improved our analyses and have revised our manuscript accordingly. While the main message is stable, we feel that our revised manuscript provides more meaningful insights.

Comment 1-3. I was surprised by how similar the inferred turnover per year was across tissues (Fig. 3i). Colonic stem cells should be dividing a lot more frequently than HSCs, and fibroblasts should be very slow. The authors suggest in line 277 that this could be due to mitochondrial turnover independently of cell proliferation, but if that is a serious possibility, I feel that it would be good to model this separately from the Fisher Wright model which couples the mtDNA duplication with cell division. If major mtDNA turnover happens in the absence of cell division, then the Fisher Wright model and its presented results may not adequately represent the underlying process and this should be clarified in the text.

Answer 1-3. We truly appreciate the reviewer for raising the issue, leading us to deeper consideration and understanding of mtDNA dynamics in each cell generation. In the original manuscript, we conceived a single mtDNA turnover model (“mitotic mtDNA turnover”), which assumed continuous mitoses over time and duplications of mtDNAs in every cell cycle, in line with the Wright-Fisher model. To abide by the reviewer’s comment, we have introduced another new model, referred to as “homeostatic mtDNA turnover” in the revised manuscript, where a damaged mtDNA copy is immediately replaced by a new mtDNA copy from one replication event. The concept of these two models is now represented in **Fig. 3c** and **Extended Data Figs. 5b-5e**. In both models, we define an *mtDNA turnover* as ‘replication of an mtDNA for N times, where N is the basal mtDNA copy number in a somatic cell.’ As Reviewer #1 understands, the former mitotic model applies to rapidly dividing cells (such as colon epithelium). The latter homeostatic model applies more to slow- or non-diving cells.

Further, to harmonize the two turnover models, we have refined one of the conditions in the mitotic turnover model. In the original mitotic model (original manuscript), all mtDNA copies were physically duplicated. However, in the modified mitotic model (revised manuscript), all mtDNA copies now have an equal chance of being duplicated. This adjustment was made because we realized the previous assumption was too stringent. Additionally, we have modified the basal mtDNA copy numbers per cell from 1,000 (original manuscript) to 750, a value that aligns more closely with our mtDNA copy number estimation (**Fig. 6a**).

Although we have two different models in the revised manuscript, our conclusion is still robust and is improved. (1) The revised mitotic turnover rates per year are now 14.3 (colon), 20.8 (fibroblast), and 17.9 (hematopoietic stem cell; **Fig. 3i**), which are ~50% of what was initially described. It is caused by the modified condition mentioned above (replication of randomly selected mtDNA). (2) The homeostatic turnover rates also provide stable values across Het_{FE} variants, 6.5 (colon), 11.5 (fibroblast), and 9.5 (hematopoietic stem cells) per year (**Fig. 3i**), which are ~50% of the corresponding mitotic turnover rates. Because these two models describe two extreme situations, the actual turnover rates should be in the middle of these values determined by the relative contributions of the two models in each cell type. (3) Interestingly, we obtained a robust absolute mtDNA mutation rates regardless of the turnover models (5.0×10^{-8} per bp replication; **Fig. 4f**), which implies that both the models are relevant.

We have updated our manuscript accordingly and have introduced “**Supplementary Discussion 4**” to describe two different models. Although we still show ‘mitotic turnover’ as a dominant model, we provide values from the homeostatic turnover model in parallel. We have newly inserted **Fig. 3h** to show the relative differences in turnover values between the two models with the results of mitotic turnover in **Fig. 3g**. In addition, we show the impact of basal mtDNA copy numbers for the final turnover rate and fixation of heteroplasmic mtDNA variants more extensively with **Extended Data Figs. 8, 9, and Supplementary Discussion 4**. These were obtained from additional simulations conducted using four different basal mtDNA copy numbers: 500, 1,000, 1,500, and 2,000. Indeed, the overall trends

were similar between the two turnover models.

Original [#234-#243]

The foundation of the early bottleneck is the lack of mtDNA replication until a certain stage of embryonic cleavage, leading to a continuous reduction in the mtDNA copy number per embryonic cell^{59,60}. If each embryonic cell has one or only a few mtDNA copies at a certain stage, the heteroplasmy level in each embryonic cell (and its downstream somatic lineage) can be quantized, thus becoming biased from the original level (early bottleneck). In parallel, mtDNAs are newly replicated and lost in many circumstances in somatic lineages (for example, mtDNA duplication and random segregation in two daughter cells at every cell division). The process can drift heteroplasmy levels over a lifetime, generating a substantial impact (lifetime drift). Of these two nonexclusive scenarios, our observations indicate that the latter is dominant for the following four reasons.

Revised [Results, page 9, Section “MtDNA turnover and drift in somatic lineages”]

*The foundation of the early embryonic mtDNA bottleneck is caused by the lack of mtDNA replication until a certain stage of embryogenesis (**Fig. 3c**)^{57,58}. If each embryonic cell has one or only a few mtDNA copies at a certain stage, the heteroplasmy level can be quantized according to the composition of founder mtDNAs in each embryonic cell. In parallel, mtDNAs are newly replicated and lost in many circumstances in somatic lineages (for example, cell-cycle dependent mtDNA duplication and random segregation by half in two daughter cells in dividing cells [**mitotic turnover**] or cell-cycle independent homeostatic mtDNA replacement in non-dividing cells [**homeostatic turnover**]; **Fig. 3c**)^{59,60}. The process can slightly drift heteroplasmy levels continuously over time, generating a substantial impact in lifetime. Of these two non-exclusive scenarios, our observations indicate that the lifetime drift is dominant.*

< Revised **Fig. 3c** >

Fig. 3c. Schematic diagram illustrating the dynamics of clone-VAFs attributable to the early embryonic bottleneck and the lifetime drift. Each circle represents mtDNA, with gray circles denoting wild-type mtDNAs and colored circles indicating mutant mtDNAs with various mutations. The expected mtDNA copy number per cell during the early embryonic bottleneck and the lifetime drift are also provided. In the part of the lifetime drift, two simulation models we conducted, namely mitotic and homeostatic turnover, are depicted in different colors.

Original [#262-#277]

Finally, the late drift model alone was sufficient to explain biased VAFs in the clones. Simulation studies using the Wright-Fisher model⁶² suggested that a series of mtDNA turnovers (defined in the study as mtDNA duplication followed by random selection of 50%) purify mtDNA mutations to wild-type, or homoplasmy. For a Het_{FE} variant with 0.1 of the initial heteroplasmy, fixation to homoplasmy occurred in ~10% of the simulation, and an average of 3,688 rounds of mtDNA turnover was necessary until the fixation in homoplasmy (Fig. 3h). Such mtDNA turnover numbers, which is equivalent to ~52 per year or 1 per week for an individual aged 70 years, are possible in human lifetime, as is close to the cell division rate. If the original heteroplasmy in the fertilized egg is high, the required turnover number for homoplasmy decreases (Fig. 3h).

Based on the landscape of VAF distributions of Het_{FE} variants, mtDNA turnover rates across cell types were inferred to be 22.7, 42.9, and 31.7 per year for the colon epithelium, fibroblasts, and HSCs, respectively (Fig. 3i). Although slight inter- and intra-individual differences in the estimated turnover rate were observed, the overall trends were consistent across the different tissues. Fibroblasts displayed the highest turnover rates, contradicting the relatively slow cell cycle observed in other cell types⁶³, suggesting an active renewal of the mitochondrial genome independently of the cell proliferation^{64,65}.

Revised [Results, page 9-10, Section “MtDNA turnover and drift in somatic lineages”]

Finally, computational simulation suggested that the lifetime drift model alone was sufficient to explain the skewed distribution of clone-VAFs of a Het_{FE} variant. Simulation studies using the mitotic turnover model (**Extended Data Figs. 5b, 5c**) indicated that 1,440 rounds of mtDNA mitotic turnovers shifted a Het_{FE} variant of 10% initial heteroplasmy level (caVAF) to homoplasmy (100%) in ~10% of the clones when clones had 750 basal mtDNA copy numbers (**Fig. 3g**; see **Methods** and **Supplementary Discussion 4** for more details; a turnover was defined as replication of an mtDNA for N times, where N is the basal mtDNA copy number in a somatic cell). As expected, smaller rounds of mitotic turnovers were necessary when a Het_{FE} variant had higher initial heteroplasmy levels, or basal mtDNA copy numbers were smaller (**Fig. 3g**). Likewise, simulations assuming homeostatic turnover (a round of homeostatic turnover was defined as mtDNA replacements as many as the baseline mtDNA copy number; **Extended Data Figs. 5d, 5e**) suggested a similar conclusion, but ~50% of rounds were necessary for a similar effect under the same conditions (**Fig. 3h**). Of note, despite the same basal copy numbers per cell, cells in the mitotic turnover model contain a higher number of mtDNA copies until mitosis than the homeostatic turnover model (**Fig. 3c**), leading to a lower drift level per turnover (**Supplementary Discussion 4**).

Based on the clone-VAF distributions of Het_{FE} variants in clones, the maximum likelihood mtDNA turnover rates across cell types were inferred (14.3, 20.8, and 17.9 mitotic turnovers per year, or 6.5, 11.5, and 9.4 homeostatic turnovers per year for the colon epithelium, fibroblasts, and HSCs, respectively; **Fig. 3i**). Although we speculate that colorectal epithelium and fibroblasts have mitotic and homeostatic mtDNA turnovers, respectively, as their predominant mechanism, the relative balance between two turnover models in each cell type is uncertain.

< New Extended Data Figs. 5b-5e >

Extended Data Figs. 5b-5e. **b-c**, A schematic diagram illustrating the concept of mitotic turnover. MtDNA undergoes random duplication within the cell until it reaches twice the original amount (N). Then, it undergoes random segregation into half, defining one turnover. Examples of changes in mtDNA frequency during one turnover (**b**) and changes in mtDNA copy number are depicted (**c**). A different color differentiates each mtDNA. mtCN, mtDNA copy number. **d-e**, A schematic diagram illustrating the concept of homeostatic turnover. Within the cell, mtDNA undergoes random degradation and duplication iteratively for the number of times equivalent to the total amount of mtDNA (N), defining one turnover. Examples of changes in mtDNA frequency during one turnover (**d**) and changes in mtDNA copy number are depicted (**e**). A different color differentiates each mtDNA. mtCN, mtDNA copy number.

< Revised Figs. 3g-3i >

Figs. 3g-3i (3h new; 3g, 3i modified). **g**, Average mitotic turnover counts to reach homoplasmism. Dots indicate simulated results. **h**, A scatter plot comparing turnover counts to reach homoplasmism

simulated from the mitotic and homeostatic turnover models. For each caVAF [0.1-0.9], the x-axis represents the average mitotic turnover counts to reach homoplasmy, and the y-axis represents the average homeostatic turnover counts. The Pearson's correlation coefficient and p-value from Pearson's correlation test are provided. The blue line represents the regression line, and the shaded area indicates its 95% confidence interval. The regression equation was calculated as $y=0.496x$ through univariable linear regression analysis. **i**, Estimated turnover rates for each Het_{FE} variant with 95% confidence intervals derived from simulations of two models, the mitotic and homeostatic turnover models. Mitotic turnover rates were described in the upper panel with the filled circles, and homeostatic turnover rates in the lower panel with the hollow circles. It includes Het_{FE} variants with the caVAF exceeding 0.005. Color bars at the top denote tissue types, and the mean values of estimated tissue-specific turnover rates are provided for each turnover model.

Original [#297-#305]

Given the mtDNA turnover rate estimated from the Het_{FE} variants and the landscape of detectable sporadic mutations, we estimated the absolute mutation rate of mtDNA underlying sporadic mutations. In all individuals, the absolute mtDNA mutation rates converged to 1.01×10^{-7} per bp per turnover (Fig. 4e). Interestingly, this value was close to the error rate of DNA polymerase gamma (POLG), the DNA polymerase for mitochondrial genomes^{57,66}, suggesting that endogenous replication was the major mutational process operating in the mitochondria^{26,30,67}. Given the ~1,000 mtDNA copies in a single cell, our absolute mutation rate implies an average of 0.83 de novo mtDNA base change per cell per mtDNA turnover ($0.83 = 1,000 \times 16,569 \text{ bp} \times 1.01 \times 10^{-7} \text{ per bp} / 2$).

Revised [Results, page 11, Section "Post-zygotic mutations in the mtDNA"]

With the mtDNA turnover rates estimated using Het_{FE} variants and the landscape of detectable sporadic mutations, we estimated the absolute number of mtDNA alterations newly appearing in every mtDNA replication (**Methods**). In all individuals and both the turnover models, the absolute mtDNA mutation rates converged to 5.0×10^{-8} alterations per bp replication (**Fig. 4f**). Interestingly, this value was close to the error rate of POLG, the DNA polymerase for mitochondrial genomes^{55,65}. The converged rate reassures that endogenous mtDNA replication is the dominant process for mtDNA mutation acquisition in somatic cells^{26,31,66}, and both the turnover models (and their turnover rates) are reliable. Given the ~750 mtDNA copies in a single somatic cell, our absolute mutation rate implies that mitosis acquires an average of 0.31 de novo mtDNA sporadic alteration per daughter cell ($0.31 = 750 \times 16,569 \text{ bp} \times 5.0 \times 10^{-8} \text{ per bp} / 2$).

< Revised Fig. 4f >

Fig. 4f. Estimated mutation rate per base pair for each individual with 95% confidence intervals derived from simulations of two models, the mitotic and homeostatic turnover models (individual ages indicated in parentheses). The filled circles represent the mutation rate inferred from the mitotic turnover model, while the hollow circles represent the mutation rate inferred from the homeostatic turnover model. Color bars at the top denote tissue types, and the mean value of the estimated mutation rate per base pair is shown.

Original [#319-#325]

Our random drift model illustrates that 1,000 turnovers induce one mtDNA copy to have ~50% of the mtDNA population in approximately 50% of the somatic cells (Fig. 5b; Supplementary Table 8). An additional 1,000 turnovers generate one mtDNA copy to reach homoplasmy in approximately 25% of the somatic cells (Fig. 5b; Supplementary Table 8). Combined with the absolute mutation rate (1.01×10^{-7} per bp), 2,000 turnovers resulted in ~50% VAF of sporadic mutations in approximately 30% of somatic cells (Fig. 5c; Supplementary Table 9).

→

Revised [Discussion, page 16]

Our lifetime drift model illustrates that 1,000 mitotic turnovers induce one of 750 mtDNA copies to have 100% of the mtDNA population in approximately 30% of the somatic cells (Fig. 7b; Supplementary Table 8). Combined with the absolute mutation rate, 1,000 mitotic turnovers resulted in homoplasmic sporadic mutations in approximately 5% of somatic cells (Fig. 7c; Supplementary Table 9). The rate of lifetime drift tends to decrease as the basal mtDNA copy number increases (Extended Data Fig. 8 and Extended Data Fig. 9).

< Revised Figs. 7b, 7c >

Figs. 7b, 7c. **b,** A contour plot representing how the mtDNA population expands with continuous mitotic turnover. The x-axis and y-axis present mitotic turnover count and top mtDNA frequency, respectively. The top mtDNA frequency represents the frequency of the most prevalent mtDNA within a cell, assuming that each mtDNA can be distinguished within a fertilized egg. The top mtDNA frequency for each mitotic turnover was calculated for 10,000 cells, and its distribution is represented in the plot. The intensity of the color deepens with an expanding ratio of cells at the corresponding top mtDNA frequency and mitotic turnover. The graph is generated with the mtDNA copy number set baseline at 750. **c,** A contour plot representing how clone VAF of the mtDNA sporadic mutation expands with continuous mitotic turnover. The x-axis and y-axis present mitotic turnover count and top mtDNA mutation heteroplasmy, respectively. The top mtDNA mutation heteroplasmy represents the highest clone VAF among mtDNA sporadic mutations within a cell. The top mtDNA mutation heteroplasmy for each mitotic turnover was calculated for 10,000 cells, and its distribution is represented in the plot. The intensity of the color deepens with an expanding ratio of cells at the corresponding top mtDNA mutation heteroplasmy and mitotic turnover. The graph is generated with the mtDNA copy number set baseline at 750. Sporadic mutations occur with the absolute mutation rate of 5.0×10^{-8} per bp replication.

< Revised Extended Data Fig. 8 >

Extended Data Fig. 8. a-d, Contour plots representing how the mtDNA population expands with continuous mitotic turnover. The x-axis and y-axis present mitotic turnover count and top mtDNA frequency, respectively. The top mtDNA frequency represents the frequency of the most prevalent mtDNA within a cell, assuming that each mtDNA can be distinguished within a fertilized egg. The top mtDNA frequency for each mitotic turnover was calculated for 10,000 cells, and its distribution is represented in the plot. The intensity of the color deepens with an expanding ratio of cells at the corresponding top mtDNA frequency and mitotic turnover. The graphs are generated with four different mtDNA copy numbers, including 500 (a), 1,000 (b), 1,500 (c), and 2,000 (d).

< Revised Extended Data Fig. 9 >

Extended Data Fig. 9. a-d, Contour plots representing how clone-VAF of the mtDNA sporadic mutation expands with continuous mitotic turnover. The x-axis and y-axis present mitotic turnover count and top mtDNA mutation heteroplasmy, respectively. The top mtDNA mutation heteroplasmy represents the highest heteroplasmy level among mtDNA sporadic mutations within a cell. The top mtDNA mutation heteroplasmy for each mitotic turnover was calculated for 10,000 cells, and its distribution is represented in the plot. The intensity of the color deepens with an expanding ratio of cells at the corresponding top mtDNA mutation heteroplasmy and mitotic turnover. The graphs are generated with four different mtDNA copy numbers, including 500 (a), 1,000 (b), 1,500 (c), and 2,000 (d). Sporadic mutations occur with the absolute mutation rate of 5.0×10^{-8} per bp replication.

Comment 1-4. Given that we now know that many nuclear genomic mutations occur independently of cell division as a function of time (e.g. SBS1 mutations in neurons), is there a reason to report the mtDNA mutation rate exclusively as “per turnover” as opposed to “per year”? In particular, given that the turnover estimate is a bit of a question (see point above)?

Answer 1-4. We are grateful to Reviewer #1 for this question. We have two reasons for calculating the mutation rate per turnover (or per bp replication) rather than per year. First, as suggested previously (Ju et al., eLife 2014) and also reproduced in this work (Figs. 1e, 1f), we speculate that the predominant process underlying mtDNA mutations is the endogenous mtDNA replication, which should be

associated with the error of DNA polymerase gamma. Therefore, the 'mtDNA turnover' is defined as the 'replication of an mtDNA for N times, where N is the basal mtDNA copy number in a somatic cell' as mentioned above and is a suitable unit for calculating the mutation rate.

Second, as mentioned in the manuscript, it is technically challenging to measure the absolute mtDNA mutation burden in a cell. We can observe only mtDNA mutations that have sufficiently expanded in the cell lineage above the detection threshold (~0.3% in this manuscript). In other words, mtDNA mutations not amplified in the lifetime drift cannot be detected. As a result, the mtDNA mutation burden per year (rate) is not constant (**Fig. 4a**). In contrast, by applying two simple values (the absolute mutation rate (per turnover or bp) and cell-type specific turnover), we can explain the variability of mtDNA mutations nicely (**Fig. 4d**). Therefore, the most suitable method for measuring the mutation rate is to estimate how many mutations occur per turnover or bp replication. In the revised manuscript, we have added a more detailed description of the utility of the absolute mutation rate and mtDNA turnover in explaining mtDNA mutations in somatic cells.

Original [#282-#283]

However, we speculate that we missed many sporadically acquired mutations, as those below our detection threshold (~0.3%) would remain undetected.

→

Revised [Results, page 10, Section "Post-zygotic mutations in the mtDNA"]

Unlike somatic mutations in nuclear genomes, it is challenging to absolutely count mtDNA sporadic mutations, as mutations with clone-VAFs below our detection threshold (~0.3%) would remain undetected.

Original [#294-#296]

The correlation extended regardless of the cell types when the age of individuals was converted into a universal parameter, the accumulated mtDNA turnover numbers ($R=0.758$, $P=1.2 \times 10^{-6}$, Pearson's correlation; Fig. 4d).

→

Revised [Results, page 11, Section "Post-zygotic mutations in the mtDNA"]

*The correlation became stronger when the age of individuals was converted to the turnover numbers from birth using the cell-type-specific turnover rates estimated from Het_{FE} variants ($R = 0.758$, $P = 1.2 \times 10^{-6}$, Pearson's correlation; **Fig. 4d**).*

Newly added [Results, page 11, Section “Post-zygotic mutations in the mtDNA”]

*With the mtDNA turnover rates estimated using Het_{FE} variants and the landscape of detectable sporadic mutations, we estimated the absolute number of mtDNA alterations newly appearing in every mtDNA replication (**Methods**).*

Comment 1-5. Is the plot in Figure 4d somewhat circular, given that turnover (x-axis) is estimated by heteroplasmy levels and S_{VAF} is a related measure?

Answer 1-5. We apologize for the confusion in **Fig. 4d**. For clarification, the turnover rates (x-axis) were estimated by heteroplasmy levels of each ‘ Het_{FE} (heteroplasmy from the fertilized egg)’ variant, and we used the averaged value from each cell type for the figure. On the contrary, the S_{VAF} (y-axis) values in **Fig. 4d** were calculated from ‘sporadic’ mutations in each clone. The turnover rate from Het_{FE} and S_{VAF} from sporadic mutations are independent. We have clarified it in the revised manuscript, as shown below.

Original [#292-#296]

In each tissue, the average S_{VAF} of the clones from an individual exhibited a strong positive correlation with age (Extended Data Fig. 5b). The correlation extended regardless of the cell types when the age of individuals was converted into a universal parameter, the accumulated mtDNA turnover numbers ($R=0.758$, $P=1.2\times 10^{-6}$, Pearson’s correlation; Fig. 4d).

→

Revised [Results, page 11, Section “Post-zygotic mutations in the mtDNA”]

*The average S_{VAF} in the clones of an individual exhibited a strong positive correlation with age (**Extended Data Fig. 6d**). The correlation became stronger when the age of individuals was converted to the turnover numbers from birth using the cell-type-specific turnover rates estimated from Het_{FE} variants ($R = 0.758$, $P = 1.2\times 10^{-6}$, Pearson’s correlation; **Fig. 4d**).*

Comment 1-6. Since the measurements in Figure 4c are linked by common ancestry, I am not sure it is fair to treat them as independent observations in a Fisher exact test (if I understand correctly what the authors are doing).

Answer 1-6. We apologize for the confusion again. Mutations of the same color indicated in **Fig. 4c** do not represent identical mutations sharing common ancestry. Instead, they represent sporadic mutations in each clone, having the highest, second-highest, and third-highest clone-VAFs. As these were all independently acquired by definition, we applied Fisher's exact test. In order to represent that these are not identical mutations, we have updated **Fig. 4c** to show two different color schemes and coordinates of mutations for some clones.

< Original **Fig. 4c** >

< Revised **Fig. 4c** >

Fig. 4c. Bar plots depicting S_{VAF} (the sum of the clone-VAFs of all detected sporadic mutations) for each clone in two individuals, DB8 (upper) and HC10 (lower), with developmental phylogenies. These plots consider only the three sporadic mutations with the highest clone-VAFs. Three sporadic mutations are color-coded differently. Pie charts indicate the clone distribution based on the S_{VAF} . Some examples of sporadic mutations are written on the graph, showing that sporadic mutations with the same color do not represent the same mutation. Het_{FE} variants and hotspot mutations are excluded from the graph.

Comment 1-7. Figure 5a is a bit misleading visually as it almost looks like the mutation rate “precedes” the turnover rate in time, perhaps this could be redesigned.

Answer 1-7. We appreciate the reviewer for the kind comment. We have updated **Fig. 5a** (now **Fig. 7a**) accordingly.

< Original **Fig. 5a** >

< Revised Fig. 7a >

Fig. 7a. Schematic diagram illustrating the origin and dynamics of mtDNA heteroplasmy across the lifetime. The mtDNA variants originating from the fertilized egg undergo drift in each somatic lineage and are often fixed to homoplasmy. In every mtDNA turnover, a new mtDNA mutation can be acquired sporadically, which can also be expanded through mtDNA turnovers. Mutation and turnover rates (mitotic and homeostatic turnover rates), two fundamental elements of lifetime drift, are shown at the top. Stars indicate newly occurred sporadic mutations. The middle panel shows how the heteroplasmy of each mtDNA variant changes over a lifetime. Each Het_{FE} variant and sporadic mutation is represented in different colors, and the bars located on the leftmost, middle, and rightmost depict the clone-VAF distribution within the cell at that moment. This can be reconfirmed by the illustration at the bottom, showing how variants exist within the mtDNA of the cell.

Comment 1-8. I do not understand the argument in lines 244-251. Perhaps this part could be rewritten for clarity (although it could also be my own personal failure unrelated to the text)

Answer 1-8. We appreciate the careful comment. We included this section to indicate that the early embryonic mtDNA bottleneck size is much larger than 1. If the early embryonic bottleneck size were extremely small, VAF in clones (clone-VAF) would be skewed to both extremes regardless of the VAF in the fertilized egg (caVAF). However, we did not observe this pattern in the actual data. We found that larger caVAF values were associated with a higher likelihood of homoplasmy in the clone. Therefore, we used this observation as evidence against the hypothesis of a small bottleneck size.

To abide by the reviewer's comment, we clarified our manuscript with an extensive revision of the paragraphs and the direct description about the bottleneck size has been minimized in the revised manuscript.

Comment 1-9. The finding of extreme mtDNA copy numbers in cancer is very interesting but the authors do not interpret it in any way. Perhaps some interpretation (even if it was a bit on the speculative side) would be helpful.

Answer 1-9. We truly appreciate the reviewer for the insightful comment. To abide by the suggestion, we have deep-dived into the issue and have reached a higher-level understanding (shown in **Figs. 6h-6j**). First, we recognized that the 'mtDNA copy number in cancer' in the original manuscript was not the copy number in cancer 'cells' but in cancer 'tissues,' which include various microenvironment cells, such as endothelial, mesenchymal, and infiltrating immune cells. We recognized that we need to focus on mtDNA copy numbers in cancer cells. Indeed, the 'mtDNA copy numbers in cancer tissues' were positively correlated with 'tumor cell fractions' estimated from the whole-genome sequencing as shown in **Fig. 6h**, a new figure panel that then the revised manuscript. It implies that the mtDNA copy number in cancer 'cells' should be ~1,266, substantially higher than its microenvironmental cells (extrapolation to tumor cell fraction is 100%). To validate the mtDNA copy numbers in pure tumor cells, we newly established 14 clonal cancer organoids (100% tumor cell fraction) from two colorectal cancer tissues. The estimated mtDNA copy number per diploid nuclear genome of tumor (cells) was 1,224 on the median, which was close to our extrapolation. Of note, it is higher than the mtDNA copy number in normal colorectal epithelium (~750 copies). The mtDNA copy numbers measured from colorectal cancer organoids are now shown in **Fig. 6i** below.

Further, when we compared the 'mtDNA copy numbers per in cancer tissues' with the number of tumor-

infiltrating CD3-positive T cells reported by pathologists, we observed a strong negative correlation, implying substantially lower mtDNA copies in tumor-infiltrating T cells. For experimental validation, we whole-genome sequenced sorted T cells. The estimated mtDNA copy number in T cells was 123 on the median, similar to the extrapolation from **Fig. 6h** (y-intercept when tumor cell fraction is 0). We have added the mtDNA copy number of T cells in **Fig. 6i** in the revised manuscript.

To describe our new interpretation of mtDNA copy number in cancer, we revised our manuscript as shown below. To avoid confusing readers, we distinguished mtDNA copy numbers in cancer 'tissues' and 'cells' in the revised manuscript.

Original [#394-#399]

Finally, we observed that mtDNA copy numbers in 19 matched colorectal carcinomas. Compared to the mtDNA copy number variations in normal clones, colorectal carcinoma demonstrated biased copy number changes towards either gain or loss of mtDNA copies (Fig. 7h). Colorectal carcinomas displayed a discernible propensity to shift from a state of mtDNA copy loss relative to normal clones to a state of gain as S_{VAF} or maximum VAF increased, but impact of accelerated turnovers on these observations was uncertain (Extended Data Fig. 6h).

Revised [Results, page 13-14, Section "Accelerated mtDNA turnover in tumorigenesis"]

*Finally, mtDNA copy numbers were observed in 19 matched colorectal carcinomas. Compared to the mtDNA copy number variations in normal clones, matched colon cancer tissues demonstrated biased copy number changes (per diploid nuclear genome) towards either gain or loss of mtDNA copies at face value (**Fig. 6g**). To gain insights into the mtDNA copy numbers in pure colon cancer cells without co-existing tumor microenvironmental cells, such as infiltrating lymphocytes, endothelial cells and other mesenchymal cells, we correlated the mtDNA copy numbers of cancer tissues with their tumor cell fractions estimated from genome sequences⁷⁷ and found a strong positive linear relationship ($R = 0.715$, $P = 5.7 \times 10^{-4}$, Pearson's correlation; **Fig. 6h**). Extrapolation of the regression line suggested that there were $\sim 1,266$ mtDNA copies per colon cancer cell (100% tumor cell fraction; relative to diploid nuclear genome), which is 70% higher than those present in normal colorectal clones. Indeed, we confirmed a mtDNA copy number increase in colon cancer cells by validated WGS of 14 colon cancer organoids (100% tumor cell fraction; 1,224 mtDNA copies per diploid cancer cell; **Fig. 6i**). The underlying reason for the increased mtDNA copy number in colon cancer cells is uncertain.*

*Similarly, mtDNA copy numbers of cancer tissues were negatively correlated with the amount of infiltrating CD3⁺ T cells estimated from the pathology review of the tissues (**Fig. 6j**). Genome*

sequencing of T cells sorted from the peripheral blood suggested that there were ~123 mtDNA copies per T cell (Fig. 6i), which was close to the value extrapolated from the regression line (Fig. 6h).

< New Figs. 6h-6j >

Figs. 6h-6j. *h*, The linear correlation for the tumor cell fraction and mtDNA copy numbers per diploid nuclear genome in tumor tissues. The Pearson's correlation coefficient and p-value from Pearson's correlation test are provided. The gray line represents the regression line, and the shaded area indicates its 95% confidence interval. When tumor cell fractions are 0 and 1, calculated by univariable linear regression analysis, tumor tissue's expected mtDNA copy numbers are represented by dashed lines and written in gray. $mtCN_{tumor}$, the mtDNA copy number of tumor tissue. *i*, Boxplots comparing the mtDNA copy numbers in bulk T cells, normal colon clones, and colorectal cancer organoids. Bulk T cells represent samples obtained after sorting T cells and conducting bulk sequencing. Two-sided Wilcoxon test, $**P < 0.01$, $***P < 0.001$. *j*, A scatter plot depicting the relationship between the mtDNA copy number of tumor tissue and the number of CD3-positive T cells. The x-axis represents the mtDNA copy number of each tumor tissue, while the y-axis represents the number of CD3-positive T cells per high-power field (HPF) observed in each tumor tissue. MtDNA copy numbers in tumor tissues are calculated per diploid nuclear genome. The estimated purity (tumor cell fraction) of each tumor tissue, obtained through sequencing, is indicated by color. $mtCN_{tumor}$, the mtDNA copy number of tumor tissue.

Comment 1-10. The paper would benefit from text revisions to make it more readable. First, the order of topics is not entirely understandable to me (e.g. the description of HetFE variants and the description of their somatic evolution is interrupted by a section on hotspot mutations), some restructuring may

improve the text flow. The writing is very dense and technical, enumerating many statistics in most paragraphs (e.g. lines 150-156 and many other examples), but I am not sure that this is necessary to convey precise points. Some of the numerical details could presumably be presented in a supplementary note or table. Instead, it might be useful to focus a little bit more on explanation and interpretation of the results in the main text. There are some language issues which I think somewhat obfuscate the precise meaning of several important conclusions outlined in the text. Re-review after text editing might be helpful.

Answer 1-10. We appreciate the reviewer for this comment, which makes our manuscript clearer. To abide by the reviewer’s comment, we reconstructed the structure of our manuscript and carefully modified it accordingly. We improved the flow between the topics in the revised manuscript by explaining the lifetime drift immediately after Het_{FE} variants. Additionally, since hotspot mutations are somewhat peripheral to the main points of this study, they are briefly mentioned in the section on “Post-zygotic mutations in the mtDNA,” with most of the explanation moved to **Supplementary Discussion 5**. When explaining the origin of mtDNA variants, we reduced the emphasis on statistics and added new figures to enhance the clarity of the critical points (**Figs. 2a, 2b** as shown below).

Furthermore, we revised paragraphs to focus more on an in-depth analysis of the caVAF calculated from Het_{FE} variants, patterns observed in transmission, and turnover describing mitochondrial dynamics. Lastly, we underwent overall text editing throughout the manuscript to ensure a clear understanding of our conclusions. We have extensively revised the manuscript, so we would appreciate it if the reviewer could review the modifications in the manuscript.

< New **Figs. 2a, 2b** >

Figs. 2a, 2b. **a**, A simple diagram showing how the origins of the identified mosaic mtDNA variants were categorized. **b**, Schematic diagram illustrating different origins of shared mtDNA variants. Different origins of mtDNA variants are color-coded.

Reviewer #2:

Comment 2-1. In the manuscript by An and colleagues, the authors investigate patterns of somatic mtDNA mosaicism through an impressive whole genome sequencing experiment covering over two thousand clones. The authors make a number of observations, some of which relate to the evolutionary origin of mtDNA mutations within an individual back to the developing embryo. The manuscript is expansive and covers significant ground. While I find some of the results exciting, there are some results which have been described previously in the literature and which may (in my subjective opinion) distract from the key messages of the manuscript. I detail my comments below, separating according to the thematic nature of the feedback.

Answer 2-1. We appreciate the reviewer's time for our manuscript and the valuable comments that make our work more meaningful.

Broad comments:

Comment 2-2. While I find the dataset that the authors produce impressive, I do not think the authors did a strong job communicating to the reader what the major gap in knowledge is that could be addressed by this new data. The evolutionary origins of somatic mtDNA mosaicism are to some extent covered by earlier work of a different flavor, e.g. in Wei...Chinnery Science 2019, where the transmission of heteroplasmies was investigated. So, I would ask the authors directly: what is the major gap in knowledge that this work addresses? And what is the conceptual advance that the reader should walk away with, how should this work make us see the world differently?

Answer 2-2. We appreciate the reviewer for allowing us to highlight the importance of our work. Our significant advance is that we explored the mtDNA heteroplasmy in healthy (non-neoplastic) human cells at the absolute "single-cell" resolution. This perspective has not been extensively explored before.

Previous studies on mtDNA heteroplasmy have primarily concentrated on mitochondrial diseases (Goto et al., Nature 1990), or changes in heteroplasmy during germline transmission (Wei et al., Science 2019), and cancer tissues (Ju et al., eLife 2014; Yuan et al., Nature Genetics 2020). However, studies on mtDNA mosaicism in normal cells have been rare, likely due to technical difficulties. Even when conducted, most studies have relied on bulk polyclonal cells. However, an in-depth analysis of mtDNA heteroplasmy from bulk polyclonal cells was challenging due to normal tissue's polyclonal

nature and thousands of mtDNA molecules within a single cell. For example, if one cell has a 10% VAF for a variant and 999 cells have 0% VAFs for the variant, the signal for the variant in the bulk sequencing (encompassing 1,000 cells) will be 0.01% VAF, far lower than the detection threshold.

Due to these limitations, numerous questions regarding mtDNA mosaicism in somatic lineages remain to be addressed. These include, but are not limited to, 'What is the landscape/frequency of mtDNA mutations in healthy aged cells?', 'Is there any cell type specificity?', 'What is the mechanism for mtDNA mutation acquisition in healthy somatic cells?', and 'What is the dynamics of mtDNA turnover in healthy somatic cells?'. In this study, we aimed to answer these questions by leveraging the advantage of having tens to hundreds of single normal cell whole-genome sequencing (WGS) data from each of the 31 individuals. We observed the landscape of mtDNA mosaicism in normal cells. We revealed two different origins of mtDNA variants in healthy somatic cells, i.e., i) heteroplasmy in the fertilized egg (Het_{FE}) and ii) post-zygotic acquired mutations (sporadic and hotspot mutations). We inferred the mtDNA turnover rate by Het_{FE} variants, and using sporadic mutations, we inferred the mtDNA mutation rate for each tissue. Based on these inferences, we explained how mtDNA heteroplasmies are shaped in different cell types over time. We believe that these topics were not covered in previous works.

To provide the aim of this study more clearly, we have updated the Introduction as shown below.

Newly added [Introduction, page 2-3]

Due to the limitations, numerous questions regarding mtDNA mosaicism in somatic lineages remain to be addressed, including the landscape of mtDNA mutations and the dynamics of mtDNA heteroplasmy in healthy somatic cells.

Comment 2-3. It felt several times throughout the manuscript as if key concepts were not clearly communicated or visualized. The most critical example is the precise definitions of the different origins of mtDNA alterations on line 150, i.e. heteroplasmies from a fertilized egg, recurrently acquired hotspot mutations (which I think are coincidental mutations?) on lines 150-159. This is the bedrock of the remainder of the paper, why not make a simple figure illustrating the point?

Answer 2-3. We appreciate Reviewer #2 for the kind suggestion. To abide by the reviewer's comment, we added new figures to clarify our main concepts (**Figs. 2a, 2b**). As the reviewer thinks, there are two different origins for mtDNA mosaic variants in normal cells: i) heteroplasmies from a fertilized egg (Het_{FE}) and ii) post-zygotic acquired mutations. Post-zygotic mutations are subcategorized into sporadic mutations and hotspot mutations. Most of them are sporadic mutations, but mutations recurrently occurring in multiple donors are designated as hotspots. We have clarified our key concepts

in the main manuscript, as shown below. Furthermore, simulation models discussed in this study were illustrated with schemes representing the structure and sufficient explanations (**Extended Data Figs. 5b-5e; Supplementary Discussion 4**).

< New Figs. 2a, 2b >

Figs. 2a, 2b. a, A simple diagram showing how the origins of the identified mosaic mtDNA variants were categorized. **b**, Schematic diagram illustrating different origins of shared mtDNA variants. Different origins of mtDNA variants are color-coded.

< New Extended Data Figs. 5b-5e >

Extended Data Figs. 5b-5e. b-c, A schematic diagram illustrating the concept of mitotic turnover. MtDNA undergoes random duplication within the cell until it reaches twice the original amount (N).

Then, it undergoes random segregation into half, defining one turnover. Examples of changes in mtDNA frequency during one turnover (b) and changes in mtDNA copy number are depicted (c). A different color differentiates each mtDNA. mtCN, mtDNA copy number. d-e, A schematic diagram illustrating the concept of homeostatic turnover. Within the cell, mtDNA undergoes random degradation and duplication iteratively for the number of times equivalent to the total amount of mtDNA (N), defining one turnover. Examples of changes in mtDNA frequency during one turnover (d) and changes in mtDNA copy number are depicted (e). A different color differentiates each mtDNA. mtCN, mtDNA copy number.

Comment 2-4. Could the authors elaborate on how they draw the conclusion that mtDNA mutations observed across all or most clones must necessarily derive from the fertilized egg? If the sampled tissues are colorectal epithelium, fibroblast, and HSC (and matched blood potentially, if I understand correctly), what is the most recent common ancestor of those cell types? How late in development is it?

Answer 2-4. We appreciate the comment and apologize for not conveying complete information about the branching time of lineages from the most recent common ancestral (MRCA) cell. The simple answer is that the MRCA cell is one of the earliest embryonic cells in life, most likely to be the fertilized egg, or at least the top founder cell for our body (soma), as we and other groups described in the previous papers (Coorens et al., Nature 2021; Park et al., Nature 2021; Nam et al., Nature 2023; Lee-Six et al., Nature 2018). To further validate the timing of MRCA of colorectal (or fibroblast) clones, we confirmed the lineage-defining nuclear genome variants in other cell types (matched bulk blood tissues). If the variants were localized in the colon (or fibroblast), they were unlikely to be shared by the blood (different cell types).

In all the phylogenies, the earliest lineage-defining nuclear variants were always shared by the blood (**Extended Data Fig. 2**). This indicates that the branching occurred before the cell fate determination (gastrulation). Moreover, the aggregated VAFs of the lineage-defining variants of the earliest two branches in bulk blood tissues were ~50%, indicating that the entire bulk cells are comprised of cells from the earliest two lineages. This suggests the MRCA cell is the first founder cell, likely the fertilized egg. We have updated our manuscript with new **Fig. 1b** and **Extended Data Fig. 2** and described it in detail in **Supplementary Discussion 1**.

Newly added [Results, page 4-5, Section “Landscape of mtDNA heteroplasmy in normal cells”]

The VAFs of lineage-defining variants in the matched bulk blood tissues suggest that the first branching in each of the phylogenies was early embryonic, likely to the first cell division in life, as

reported previously³ (Fig. 1b; Extended Data Fig. 2; Supplementary Discussion 1).

< New Extended Data Fig. 2 >

Extended Data Fig. 2. The phylogenetic trees of 26 individuals colored by the VAF of lineage-defining early embryonic mutations (EEMs) in bulk blood tissues. Phylogenetic trees of each individual are reconstructed using somatic mutations in the nuclear genome, with branch lengths

proportional to the number of somatic mutations. After identifying the EEMs constituting the first two branches diverging from the MRCA (referred to as Lineage 1 and Lineage 2), their VAF in bulk blood was calculated. Then, the average VAF of each lineage was provided on the phylogenetic tree. In cases where branching from the MRCA resulted in more than two lineages, two with the highest VAFs in bulk blood were selected as Lineage 1 and Lineage 2. The sum of the average VAFs for Lineages 1 and 2 is shown in Fig. 1b. The phylogenetic trees display up to 30 EEMs, and each donor's name is provided at the top right of each panel. nDNA, nuclear DNA.

< New Fig. 1b >

Fig. 1b. Bar graphs showing the VAFs of early lineage-defining nDNA (nuclear DNA) mutations in bulk blood. Lineage-defining mutations refer to the somatic mutations in the nuclear genome that constitute the first two branches (designated as Lineage 1 and Lineage 2) branching from the MRCA (most recent common ancestor) in the phylogenetic tree. Cells carrying Lineage 1 mutations and Lineage 2 mutations are mutually exclusive, and the mutation VAF for each lineage is depicted in a cumulative bar graph. The top illustration explains the interpretation of this graph. If the VAF of mutations corresponding to each lineage in the bulk tissue totals 50%, it suggests that cells carrying Lineages 1 and 2 mutations comprise the entire cell population. This indicates that Lineages 1 and 2 represent lineages at the 2-cell stage. If the combined mutation VAF for Lineages 1 and 2 is 25%, it implies that Lineages 1 and 2 represent lineages at the 4-cell stage,

as they constitute 50% of the entire cell population. Different color bars at the bottom represent the original tissues from which each individual's clone was obtained.

Comment 2-5. Related to the above, but now in a more speculative mode (I don't expect the authors to be able to address this question, but if they can it would be nice to add to the manuscript): is there evidence to conclude whether the Het_{FE} variants are predominantly de novo or alternatively transmitted from the mother?

Answer 2-5. We appreciate the reviewer's insightful comment. Although it is challenging to conclude experimentally, we speculate that most of the Het_{FE} variants we found were transmitted from the mother. Because the lowest caVAF we observed was approximately 0.01% (**Fig. 2f**), we presume that each of the Het_{FE} variants should be shared by at least ten mtDNA copies in the fertilized egg, assuming a total of >100,000 mtDNA molecules in a fertilized egg (Floros et al., Nature Cell Biology 2018). If mtDNA replication is not active in the fertilized egg, we believe a de novo mtDNA mutation in the fertilized egg should be confined to a single mtDNA copy. However, if the mutations were acquired during the oogenesis, the possibility of acquiring heteroplasmy shared by multiple mtDNA copies becomes more plausible.

Furthermore, to deeply understand the transmission of the Het_{FE} variants, we explored bulk tissues from 294 families, which included 407 mother-offspring pairs. Although we were unable to detect the Het_{FE} variants with a small fraction in the fertilized egg using bulk tissues, we identified approximately one Het_{FE} variant with >0.5% VAF per fertilized egg (**Fig. 2g**). Among these, ~20% of Het_{FE} variants in the offspring were also found in mother's bulk blood as heteroplasmic variants (**Figs. 2g, 2h**). This suggests that 20% of Het_{FE} variants observed in the offspring likely originated from the mother's fertilized egg (Het_{FE} in the mother) because these were also mosaic in the mother's somatic cells. The remaining 80% of Het_{FE} variants were not found in the mother's bulk blood, indicating that these were most likely acquired during the mother's oogenesis. Of note, when we focused on the mother's heteroplasmic variants, we found that ~20% of variants were transmitted to the next generation (**Fig. 2i**).

We have updated our manuscript to discuss the insightful issue.

Newly added [Results, page 7-8, Section "MtDNA heteroplasmy in the fertilized egg"]

To validate the landscape of Het_{FE} variants in other cohorts, we explored WGSs in bulk blood tissues from 294 families (including 407 mother-offspring pairs)⁵¹ to discover heteroplasmic mtDNA variants in the offspring (likely Het_{FE} variants). Of note, the VAF cutoff for Het_{FE} variants

here with bulk tissues was much higher than our analysis with clones ($>0.5\%$ vs $>0.047\%$ or lower, respectively). Intriguingly, in the VAF range ($>0.5\%$), we observed similar levels of heteroplasmic variants in the offspring ($n = 425$; **Fig. 2g**), with approximately one Het_{FE} variant with $>0.5\%$ VAF per fertilized egg.

Notably, we believe that the actual number of Het_{FE} variants is higher than observed in this study, as our detection threshold was $\sim 0.02\%$ with clones. Given that a fertilized egg typically contains $\sim 100,000$ mtDNA copies⁵², detectable Het_{FE} variants in this study should be shared by 20 mtDNAs in the cell, and those confined to a smaller number of mtDNA copies should be missed. In addition, we speculate that most of the Het_{FE} variants found in this study were transmitted from the mother, as newly acquired mutations in the fertilized egg could not be shared by multiple mtDNAs.

We further dissected the maternal origins of Het_{FE} variants using the family data again ($n = 294$ families). Interestingly, approximately 20% of Het_{FE} variants in the offspring were also detected as heteroplasmic variants in the mother's bulk blood ($n = 82$; **Figs. 2g, 2h**). This strongly indicates that these variants were pan-body mosaic in the mother and were not confined to the maternal germline; thus, they were present in the mother's first cell (fertilized egg) inherited from the maternal grandmother. The remaining 80% of Het_{FE} variants in the offspring were not found in the mother's bulk blood and were likely acquired during the mother's oogenesis.

Furthermore, we traced the transmission of Het_{FE} variants to the next generation using all heteroplasmic variants in the mothers' blood tissues (Het_{FE} variants of mothers; $n = 369$ from 407 mother-offspring pairs). Overall, $\sim 20\%$ of Het_{FE} variants were found in the next generation (**Fig. 2i**).

< New Figs. 2g-2i >

Figs. 2g-2i. **g**, The VAF distribution of mtDNA variants in offspring obtained from bulk tissues of 407 mother-offspring pairs. If the same variant is present in the mother, it is classified as 'grandmother origin (Het_{FE} in mother)'; if it is present only in the offspring, it is classified as 'de novo' and indicated in different colors. **h**, A scatter plot depicting the VAF of mtDNA variants found

in 407 mother-offspring pairs. The x-axis represents the VAF of variants found in the mother's bulk blood, and the y-axis represents the VAF of variants found in the offspring's bulk blood or buccal swab tissue. Variants present in both the mother and offspring are classified as 'grandmother origin (Het_{FE} in mother),' variants present only in the offspring are classified as 'de novo,' and variants present only in the mother are classified as 'lost,' with each category represented in different colors. The gray diagonal line represents a trendline indicating a one-to-one correspondence between the same x and y values. *i*, A bar plot categorizing mtDNA heteroplasmy found in the mother's bulk blood and the offspring's bulk tissue as 'lost,' 'grandmother origin (Het_{FE} in mother),' or 'lost.' The number of mosaic mtDNA variants belonging to each classification for the mother and the offspring are provided.

Comment 2-6. In general, there are a number of findings in this manuscript which seem to have been reported in prior work. In some cases, the authors go further than the prior work, but in other cases it is less clear. Some examples which could be clarified: **(a)** the existence of heteroplasmic variants in the early embryo, which is similar to the cited study from the UK Biobank (ref 50, line 198); **(b)** the description of truncating variants in colorectal cancer, **(c)** differences in RNA and DNA VAFs in certain tRNA mutations and their implications for tRNA processing

Answer 2-6. We appreciate the reviewer's comments and understand what the reviewer means. Because the topics raised by the reviewer have been previously addressed, we cited relevant previous literature in the initial manuscript. However, from our work, we believe that our understanding substantially advanced by observing the landscape of mtDNA heteroplasmy in healthy human cells at the single-cell resolution.

(a) In the previous study (UK Biobank), the authors qualitatively revealed the presence of heteroplasmic variants in the fertilized eggs. In this manuscript, we have utilized variants observed in somatic cells to quantitatively reconstruct the number and levels of the heteroplasmy in the fertilized eggs. While previous studies have reported transmitting mtDNA variants from mothers to offspring (Wei et al., Science 2019), these studies relied solely on whole-genome sequencing of bulk blood, limiting the analysis to variants with relatively higher VAF. Our study could dissect the heteroplasmy further, being able to detect variants having ~0.01% levels in fertilized eggs. We could systematically explore 31 fertilized eggs.

(b) Previously, it had been observed that colorectal cancer exhibits a higher prevalence of truncating variants compared to other cancer types (Yuan et al., Nature Genetics 2020). Our analysis demonstrates a higher prevalence of truncating variants in colorectal tumors than in colorectal normal cells. Our observation further suggests that mtDNA truncating mutations may have a favorable role in

colorectal tumorigenesis.

(c) Using whole-genome and transcriptome sequences obtained from the same clone, we observed changes in the RNA VAFs of mutations located in the stem regions of tRNA and rRNA. While previous studies demonstrated the impact of mutations in the tRNA stem region on RNA VAF and tRNA processing in cancer cells (Stewart et al., PLoS Genetics 2015), we extend these findings to normal cells. Additionally, to the best of our knowledge, the transcriptional VAF reduction of mtDNA mutations in the rRNA stem region has not been reported previously.

Comment 2-7. Perhaps I missed it, but I saw neither a link to the code necessary for reproducing the analysis in this manuscript, nor (more importantly) the data. Data of this scale would be exceptionally useful for the field at large, and it would be fantastic for the authors to not only deposit the raw data in a suitable repository, but also to provide processed mutation calls as Supplementary Data (or placed in a repository).

Answer 2-7. We appreciate the reviewer's great suggestion. We have provided mutation calls identified in this study in **Supplementary Table 4**. In addition, we have deposited CRAM files of mtDNA in the EGA with accession no. EGAS00001007674, and the related code in GitHub (https://github.com/jisong-an/mtDNA_mosaicism).

Specific But Major Comments:

Comment 2-8. Line 187: The statement or assumption that the caVAFs of the HetFE variants correspond to heteroplasmy in the first cell of life feels like pure speculation. Is there any way to corroborate this? The line of reasoning the authors use is plausible, but so are many others. Can some mathematical modeling (e.g. Wright-Fisher) help?

Answer 2-8. We appreciate the reviewer for the professional comments. As we showed in the original manuscript (**Fig. 2e** in the revised manuscript), the clone-averaged VAFs (caVAFs) calculated from multiple clones from a specific cell type (i.e., colon epithelial cells) were very close to its VAF in the bulk blood (blood-VAF). Of note, two VAFs (caVAF and blood-VAF) were measured by applying different methods to different cell types. The most plausible speculation for the correlation is that these two VAFs share the same root, the initial heteroplasmy level in their MRCA cell (or the fertilized egg). To abide by the reviewer's comment, we further validated our speculation in two ways, discussed in

Supplementary Discussion 3 of the revised manuscript.

First, as the reviewer suggested, we performed computational simulations to track the clone-averaged VAF of mtDNA variants in a founder cell over multiple rounds of mitotic cell generations. The mathematical modeling confirmed that the caVAF indeed maintains the initial VAF in the ancestral cell. The summary of these simulations has been added as **Extended Data Fig. 4d**.

Further, as another experimental validation, we conducted a new analysis on bulk tissues of monozygotic twins. By comparing the VAFs of Het_{FE} variants across multiple tissues of the twins, we found a strong correlation between the buccal swab tissues of the twins and between the bulk blood and buccal swab tissues. This observation suggests that the VAFs for these variants correspond to those in the fertilized egg of the twins. The summary of the twin sequencing has been added as **Extended Data Figs. 4e, 4f**.

Original [#179-#187]

We observed that the clone-averaged VAF (referred to as caVAF; Extended Data Fig. 2d) of a Het_{FE} variant in an individual was close to the heteroplasmy in the matched polyclonal blood tissues ($R=0.967$, $P=2.3 \times 10^{-16}$, Pearson's correlation; Fig. 2c). We speculated that a plausible mechanistic link between these two values was the original heteroplasmy level in the fertilized eggs. Although the VAF of a Het_{FE} variant may fluctuate in each clonal lineage, the average VAF among many clones would be close to the original heteroplasmy level. Likewise, their VAF in the blood should also be similar to the initial level as blood tissue is a mixture of many polyclonal lineages. Therefore, we concluded that caVAFs of the Het_{FE} variants can be interpreted as its heteroplasmy level in the first cell of life.

**Revised** [Results, page 7, Section "MtDNA heteroplasmy in the fertilized egg"]

*Then, we assessed the heteroplasmy levels of Het_{FE} variants in the fertilized egg. Interestingly, we observed that the averaged clone-VAF of a Het_{FE} variant in all the clones of a donor (referred to as clone-averaged VAF [caVAF]; **Extended Data Fig. 4c**) was close to the heteroplasmy level in the matched polyclonal blood tissue ($R = 0.967$, $P = 2.3 \times 10^{-16}$, Pearson's correlation; **Fig. 2e**). We speculated that a plausible mechanistic link between these two independent values was the original heteroplasmy level in the fertilized eggs. Although the clone-VAF of a Het_{FE} variant may fluctuate in each clonal lineage, the average of clone-VAFs among many clones (caVAF) would be close to the original heteroplasmy level, which is consistent with our computational simulation (**Extended Data Fig. 4d**). As the VAF from the bulk blood tissues (blood-VAF) is an averaged VAF among many polyclonal blood cells, it should be also close to the initial heteroplasmy level.*

To further validate our speculation, we compared the heteroplasmic mtDNA variants (Het_{FE} s most likely) from the bulk buccal swab tissues of 19 monozygotic twin siblings. Indeed, VAFs of all 16 variants showed a strong correlation with each other (**Extended Data Fig. 4e**). These variants were also found in bulk blood tissues of the twins with similar VAF levels (**Extended Data Fig. 4f**). These findings collectively indicate that the initial level of a Het_{FE} variant is on average stable among a group of polyclonal cells (blood or buccal tissues or a collection of multiple clones). Therefore, we used the caVAF (or blood-VAF) as a proxy for the heteroplasmy level of a Het_{FE} variant in the fertilized egg (**Supplementary Discussion 3**).

< New Extended Data Fig. 4d >

Extended Data Fig. 4d. A graph showing the relationship between the initial VAF and caVAF (average clone-VAFs across clones in an individual), as confirmed through computational simulations. After specifying the initial VAF per cell (VAF in the fertilized egg), simulations were conducted to repeat mitotic turnovers in 10,000 cells. The process of sampling 100 cells and calculating their average was repeated 1,000 times. The process of sampling 100 cells represents the sequencing process, and the average among these cells represents the caVAF we calculated. The results are depicted on a graph based on the number of mitotic turnovers, where the x-axis represents the initial VAF, and the y-axis represents the calculated caVAF. Each point denotes the average of the calculated caVAF values, and the vertical lines crossing the points indicate 95% confidence intervals of caVAF. Different colors indicate the number of mitotic turnovers.

< New Extended Data Figs. 4e, 4f >

Extended Data Figs. 4e, 4f. e-f, Scatter plots comparing the VAFs obtained from bulk tissues of monozygotic twins. After identifying variants in monozygotic twins, variants shared by twins were selected to compare. VAFs from the buccal swab tissues were compared between monozygotic twins (e), while VAFs from the different tissues were compared within one twin (cord blood and buccal swab tissue; f). The Pearson's correlation coefficient and p-value from Pearson's correlation test are presented. The gray diagonal lines represent trendlines indicating a one-to-one correspondence between the same x and y values.

Comment 2-9. To what extent do the hotspot variants the authors describe overlap with those reported in cancer, i.e. at specific homopolymeric loci in Complex I genes? Does the overlap or lack thereof imply something about evolutionary origins?

Answer 2-9. We appreciate the comment. To abide by the reviewer's comment, we checked six mtDNA hotspot mutations reported in cancers (Gorelick et al., Nature Metabolism 2021). Of these, four homopolymeric hotspots in Complex I genes were included in the list of hotspot variants in our study (m.10947-10952, m.11032-11038, m.11867-11872, m.12418-12415). The four hotspots were short indels in the homopolymeric sequence contexts, suggesting their independent acquisition by DNA polymerase errors. We have added the observation in the revised manuscript.

Newly added [Supplementary Discussion, page 7-8, Section "mtDNA hotspot mutations"]

Furthermore, we explored the resemblance of these hotspot mutations to six hotspot mutations found in cancers⁸. Among these, four homopolymeric hotspots in Complex I genes were included in our list of hotspot mutations (m.10,947-10,952, m.11,032-11,038, m.11,867-11,872, m.12,418-

12,415). These four hotspots comprised short indels within the homopolymeric sequences, suggesting their independent acquisition via DNA polymerase errors.

Comment 2-10. Line 249: “This indicates that the size of the early mtDNA bottleneck was much higher than 1.” I’m less familiar with this aspect of mtDNA selection, can the authors elaborate further (it can be in the Methods) as to what a bottleneck size corresponds to (the units?) and provide some intuition?

Answer 2-10. We apologize for not conveying the complete information about the early mtDNA bottleneck. We use ‘early embryonic mtDNA bottleneck’ to express the minimum mtDNA copy number per embryonic cell. Because mtDNA replication does not occur during cleavage in embryogenesis, the mtDNA copy number per cell readily decreases. In the original manuscript, we assumed an extreme scenario where the mtDNA copy number per cell is 1. In such a circumstance, all the Het_{FE} variants would become homoplasmic in the downstream lineages, which does not agree with our observation.

To clarify our description, we have revised the section.

Original [#234-#238]

The foundation of the early bottleneck is the lack of mtDNA replication until a certain stage of embryonic cleavage, leading to a continuous reduction in the mtDNA copy number per embryonic cell^{59,60}. If each embryonic cell has one or only a few mtDNA copies at a certain stage, the heteroplasmy level in each embryonic cell (and its downstream somatic lineage) can be quantized, thus becoming biased from the original level (early bottleneck).

Revised [Results, page 8-9, Section “MtDNA turnover and drift in somatic lineages”]

The foundation of the early embryonic mtDNA bottleneck is caused by the lack of mtDNA replication until a certain stage of embryogenesis (Fig. 3c)^{57,58}. If each embryonic cell has one or only a few mtDNA copies at a certain stage, the heteroplasmy level can be quantized according to the composition of founder mtDNAs in each embryonic cell.

Comment 2-11. Regarding the SVAf calculations around lines 280-296, I’m confused about the difference between SVAfs in old and young individuals. While there is not a statistically significant association between age and sporadic mutations, it seems like this is still a confounded. Is it possible to control for this in a more principled analysis using a linear model with number of sporadic mutations

as a covariate?

Answer 2-11. In lines 280-296, we showed that age is not significantly correlated to the number of sporadic mutations but the sum of the clone-VAFs of all sporadic mutations in a clone (S_{VAF}). We compared S_{VAF} s between old and young individuals and found that old individuals have much higher S_{VAF} s than young individuals. Of note, the S_{VAF} s of each clone were calculated using clone-specific sporadic mutations, not Het_{FE} variants or hotspot mutations. In contrast, the number of ‘detectable’ sporadic mutations between old and young individuals is not significantly different, as shown in the new **Extended Data Fig. 6c** in the revised manuscript. Therefore, we think the number of ‘detectable’ sporadic mutations is not a major confounder for S_{VAF} s.

Newly added [Results, page 10-11, Section “Post-zygotic mutations in the mtDNA”]

*Of note, there was no significant difference in the crude number of sporadic mutations between the clones of two individuals (**Extended Data Fig. 6c**).*

< New **Extended Data Fig. 6c** >

Extended Data Fig. 6c. Boxplots illustrating the mtDNA sporadic mutation counts in clones of DB8 (93 years old) and HC10 (37 years old). Two-sided Wilcoxon test; ns, not significant.

Comment 2-12. Figure 2f: I found this confusing and was hoping the authors could clarify. The figure

annotates m.12417 as a hotspot but in the figure legend it is labeled as Het_{FE}. Not sure what I am looking at, and how this shows a phasing error.

Answer 2-12. We apologize for the critical error in the Fig. 2f legend of the original manuscript. First, m.12,417 CA>C is a hotspot, not a Het_{FE} mutation. After that, we observe two variants in the vicinity of each other, m.12,417 CA>C and m.15,623 G>A (a sporadic mutation). These two variants can be phased by sequencing reads, and we observed four different allele combinations (wt-wt, mutant-wt, wt-mutant, mutant-mutant). Applying the four-gamete test suggests (1) a repeated mutation and (2) recombination between the two loci. As the likelihood of mtDNA recombination is low and we found the m.12,417 CA>C in multiple clones of DB6 and other individuals, the most plausible scenario for the four gametes is that m.12,417 CA>C is the repeated mutation. We have updated the sentences and the figures accordingly to clarify it. This issue was also discussed in **Supplementary Discussion 5**.

Newly added [Supplementary Discussion, page 7, Section “mtDNA hotspot mutations”]

*Consistent with independent multiple acquisitions, specific hotspot mutations often demonstrated an inconsistent phasing pattern with other variants (**Supplementary Fig. 3c**). For instance, we observed two variants in the vicinity of each other, m.12,417 CA>C and m.15,623 G>A, which are a hotspot mutation and a sporadic mutation, respectively (**Supplementary Fig. 3a**). According to the pigeonhole principle, these two variants can be phased by sequencing reads. We then observed four different allele combinations: wild-type–wild-type; wild-type–mutant; mutant–wild-type; and mutant–mutant (**Supplementary Fig. 3c**). Application of the four-gamete test indicates the potential for 1) repeated mutation and 2) recombination between the two loci. Considering the low likelihood of mtDNA recombination and our observation of m.12,417 CA>C in multiple clones of DB6 and other individuals, the most plausible scenario for the four gametes involves m.12,417 CA>C as the repeated mutation. The phasing pattern of the four gametes further supported the fact that the classified hotspot mutations occur recurrently and independently.*

< New **Supplementary Figs. 3a, 3c** >

Supplementary Figs. 3a, 3c (3a modified; 3c new). **a**, An example of mtDNA hotspot mutation (m.12,417 CA>C) recurrently found in clones of the individual and in other individuals. Clone-VAFs of the mutation are depicted in bar plots in the middle, and clone-VAFs of a unique sporadic mutation (m.15,623 G>A) are presented in bar plots at the bottom. **c**, Illustrations depicting the observed VAFs and genotypes in two clones of DB6. Each panel shows a bar plot representing the clone-VAF of m.12,417 CA>C and m.15,623 G>A on the left. The top right of each panel displays the observed genotypes from IGV, and the bottom right shows an IGV screenshot. Despite Clone 1 and Clone 2 being obtained from the same individual, Clone 1 exhibited m.12,417 CA>C but not m.15,623 G>A. In contrast, Clone 2 harbored m.12,417 CA>C and m.15,623 G>A, with m.15,623 G>A in a higher proportion of mtDNA. This indicates the existence of four genotypes within the cells of this individual.

Comment 2-13. I'm having trouble reconciling two ideas about the clone-averaged VAF (caVAF): early in the manuscript, it is used to summarize a smooth unimodal distribution of heteroplasmies across the cells of a clone. But later in the manuscript, it is revealed that sometimes caVAF summarizes a variant which has been pushed to extinction (0% VAF) or homoplasmy (100% VAF). Is the caVAF then useful?

Answer 2-13. We appreciate the comment and apologize for not conveying the clear features of clone-averaged VAF. The distribution of a Het_{FE} variant's VAFs across clones ('clone-VAF') was dichotomized towards 0 and 100% as donors age, as seen in **Figs. 3a, 3b, 3d, and 3e**. However, when these

dichotomized clone-VAFs were averaged (clone-averaged VAF, or caVAF), the caVAF value seemed to be stable regardless of the age and was close to the VAF of the Het_{FE} variant in the bulk blood tissue (blood-VAF; **Fig. 2e**). Therefore, we believe that the caVAF would be a proxy to its heteroplasmy level in the fertilized egg, as we further demonstrated in the **Extended Data Figs. 4d-4f** in the revised manuscript.

Throughout this manuscript, the caVAF is an important concept because it enables us to estimate the initial heteroplasmy level of a Het_{FE} variant in the fertilized egg. It further led us to calculate mtDNA turnover rates in the somatic lineages and then measure the absolute mtDNA mutation rate.

We believe that the confusion may be caused by many different VAF types introduced in the original manuscript. Therefore, we have put specific terminologies in the revised manuscript, such as caVAF, clone-VAF, and blood-VAF. In addition, as mentioned above, we have clarified the concept and utility of caVAF in the revised manuscript. The revised sections of the manuscript and the newly added figures, **Extended Data Figs. 4d-4f**, have been addressed in **Answer 2-8**.

Comment 2-14. Figure 5A: What do the colors represent?

Answer 2-14. We apologize for the confusion in **Fig. 5a** of the original manuscript. We have updated the figure and the figure legend for its clarification. Briefly, each color represents the fraction of an mtDNA type carrying a specific mutation (its heteroplasmy level) in a somatic cell in a somatic lineage. For example, the red color illustrates the mtDNA type carrying the red variant. On top of that, we illustrated two possible pressures that induce fluctuation of heteroplasmy levels over time, i.e., 1) early embryonic bottleneck and 2) lifetime drift by mtDNA turnovers.

< Original **Fig. 5a** >

< Revised Fig. 7a >

Fig. 7a. Schematic diagram illustrating the origin and dynamics of mtDNA heteroplasmy across the lifetime. The mtDNA variants originating from the fertilized egg undergo drift in each somatic lineage and are often fixed to homoplasmy. In every mtDNA turnover, a new mtDNA mutation can be acquired sporadically, which can also be expanded through mtDNA turnovers. Mutation and turnover rates (mitotic and homeostatic turnover rates), two fundamental elements of lifetime drift, are shown at the top. Stars indicate newly occurred sporadic mutations. The middle panel shows how the heteroplasmy of each mtDNA variant changes over a lifetime. Each Het_{FE} variant and sporadic mutation is represented in different colors, and the bars located on the leftmost, middle, and rightmost depict the clone-VAF distribution within the cell at that moment. This can be reconfirmed by the illustration at the bottom, showing how variants exist within the mtDNA of the cell.

Comment 2-15. Line 411: "...this is the first comprehensive study to reveal mtDNA heteroplasmy in fertilized eggs." This is a vast overstatement, there is no direct profiling of fertilized eggs.

Answer 2-15. We appreciate the comment. We estimated the mtDNA heteroplasmy within the fertilized egg by tracing information obtained from somatic cells. However, we did not directly sequence the fertilized egg to identify the mtDNA heteroplasmy. Therefore, we have toned down the sentence.

Original [#411-#412]

To the best of our knowledge, this is the first comprehensive study to reveal mtDNA heteroplasmy in fertilized eggs.

Revised [Discussion, page 15]

To the best of our knowledge, this is the first study to systematically estimate the amounts of mtDNA heteroplasmies in human fertilized eggs. Similar approaches with a higher number of clones from an individual than those used in this study will provide a more comprehensive landscape of Het_{FE} variants in humans.

Methodological comments (mostly related to Methods: "Calling and filtering mtDNA mutations")

Comment 2-16. How the locus-specific background noise matrix is constructed is not clear to me. It is mentioned that it is locus specific but based on the supplementary table it appears to me position and alternate allele specific so I would specify this. How do you test specifically for a variant to be "significantly greater than the background noise"?

Answer 2-16. We agree with the reviewer's comment. We have updated our manuscript with more descriptions of the methods. We specifically calculated background noise for each mtDNA position. The locus-specific background noise level can be estimated by the fraction of erroneous base calls (all the alternate allele supporting reads) at a specific position from all the clones presumed to be wild-type homoplasmy.

From the practical point of view, to determine background noise in the whole-genome sequences from a clone, we piled up all the reads for each mtDNA position. Then, we calculate the allele fraction of all the three alternate alleles. Two criteria were used to determine whether the VAF of an allele in a clone was significantly higher than the background noise. First, we calculated the average and standard deviation of the VAFs in each position, computing the one-sided 95% confidence interval. Clones having VAF beyond the interval were then determined to be mutant in the locus.

In parallel, we sorted VAFs of clones in a specific position in ascending order and focused on the gap between adjacent VAFs. The second approach was conducted because we realized that the quantum jump of the gap was a helpful clue to determine the cutoff value between the true signals and background noises. To this end, we calculated the relative gap between adjacent VAFs as follows:

$$\text{relative gap} = \frac{VAF' - VAF}{VAF}$$

where VAF and VAF' denote the lower and higher adjacent VAFs from an mtDNA locus. We only consider VAFs between 0.05 and 15% when calculating the relative gap. Following this calculation, we identified the adjacent VAFs, which showed the largest gap among those with a relative gap of 0.33 or higher. We considered this gap the boundary between the background noises and true signals. The actual cutoff value was set as a smaller value between 1) the average of VAF and VAF' or 2) $VAF \times 1.33$. Variants failing to surpass this cutoff, even after being called, are categorized as false positives and consequently excluded from the variant set. The background noise cutoffs were manually confirmed after their computation.

Original [#857-#865]

A background noise matrix was generated locus-by-locus using all whole-genome sequences of normal clones to establish high-confidence mutation sets. Because of the highly repetitive nature of mtDNA, the background noise rate is variable across loci of mtDNA. We systematically measured VAFs across all loci of mtDNA within every normal clone and subsequently constructed VAF distributions for each locus. Mutations were considered true positives only if their VAFs significantly surpassed the background levels inferred from the VAF distributions. Any mutations that were not called initially were rescued. Conversely, mutations with VAFs similar to or lower than the background noise were classified as false positives. They were consequently excluded from mutation sets.

→

Revised [Methods, page 32, Section “Calling and filtering of mtDNA mutations”]

A background noise matrix was generated for each locus of alternate alleles using all whole-genome sequences of normal clones to establish high-confidence mutation sets. Because of the highly repetitive nature of mtDNA, the background noise rate varies for alternate alleles at each locus of mtDNA. We systematically measured VAFs for all loci and alternate alleles of mtDNA within every normal clone. Subsequently, we constructed VAF distributions for each locus and alternate allele, considering the background noise matrix. Then, we overlaid the called variant set onto the background noise matrix.

To determine the background noise criteria, we first calculated the average and standard deviation of the VAFs in each position, computing the one-sided 95% confidence interval. Clones with VAF beyond the interval were considered mutants in the locus. In parallel, the VAFs of the background noise matrix in a specific position were sorted in ascending order, and the gaps between adjacent VAFs were examined. We considered the quantum jump of the gap as the cutoff value between the true signals and background noises. To this end, we calculated the relative gap between adjacent VAFs as follows:

$$\text{relative gap} = \frac{VAF' - VAF}{VAF}$$

where VAF and VAF' denote the lower and higher adjacent VAFs from an mtDNA locus. When calculating the relative gap, we only use VAFs between 0.05%-15%. After calculating the relative gap, we identified the adjacent VAFs, which showed the largest gap among those with a relative gap of 0.33 or higher. We considered this gap the boundary between the background noises and true signals. The actual threshold value was set as a smaller value between 1) the average of VAF and VAF' or 2) VAF × 1.33. If a variant did not exceed this threshold even after being called, it was considered a false-positive and excluded from the variant set. Conversely, if a variant was not called but exceeded this threshold, it was considered a false-negative and rescued.

Comment 2-17. On line 863 it is stated that “any mutations that were not called initially were rescued. Do you mean that if a mutation was not called using HaplotypeCaller2 or VarScan2 but its VAF was higher than the background noise then it was considered a variant? I am curious as to how many mutations were detected in this fashion and the justification of using these when they were not called by neither HaplotypeCaller2 nor VarScan2.

Answer 2-17. As HaplotypeCaller2 and VarScan2 callers were developed for calling mutations in diploid genomes, mtDNA variants showing insignificant VAFs (i.e., <5%) were often neglected. Therefore, we aimed to rescue the variants using the cutoff specified in the background noise matrix,

as described above. Since we piled up the VAF for each alternate allele at every locus in all clones, we could discern the VAFs of other clones even if they were not called. After determining the background noise cutoff, we rescued variants, despite not being called, that exhibited VAFs surpassing this cutoff. Although the variant callers did not call rescued variants, they were included in the variant set as false negatives, considering their higher VAFs than other errors. We rescued 19 variants with VAF ranging from 0.3% to 1%. An example of the rescued variants has been added as **Extended Data Fig. 1f**.

< New **Extended Data Fig. 1f** >

Extended Data Fig. 1f. Detailed illustrations of the filtering process based on locus and alternate allele. Clone-VAFs for each mutation in normal clones are aligned in descending order. The yellow bar denotes the clone-VAF of the called mutation in the specific clone, while the surrounding gray bars indicate VAFs of the same mutation in other clones. **f**, A representative example demonstrating a rescued variant (m.16,384 G>A).

Reviewer #3:

Comment 3-1. This manuscript from An and colleagues describes an interesting and unusual approach to studying mitochondrial DNA mosaicism in human somatic mitotic cells. The paper is understandable, but would benefit from reordering and rewriting for lucidity and communicability. The authors use data from 'clones' mostly derived from healthy tissues (with a few colorectal polyps and carcinomas included), which were cultured, sequenced and the data analysed.

Answer 3-1. We appreciate the reviewer's valuable and constructive comments. As Reviewer #3 (and also Reviewer #1; see Comment 1-10) suggested, we reordered the topics to make our manuscript more readable. Briefly, the recurrent mutation part was ordered in the later part as it interrupts the manuscripts' narrative.

Comment 3-2. There are a few fundamental issues with the approach the authors conclusions are based upon. Chief among these is the diversity of ways in which the 'clones' were derived and/or cultured. Assuming that single crypt organoid derivation vs hsc culture vs expansion of cancer cells will demonstrate similar selective pressures to one another or to the pressures present in situ is, at best, hopeful. It is widely known that culture conditions alone in 2D monoculture of cytoplasmic hybrids can induce dramatic shifts in heteroplasmy. As such, the unknown selective pressures being applied by the authors choice of sampling weakens the foundation of their conclusions.

Answer 3-2. We understand the Reviewer #3's concern. To establish clones from three different tissues, we employed three methodologies. For the colorectal epithelium, we used the organoid technique. In contrast, for fibroblasts and HSCs, conventional cell culture techniques were applied to clonalization. To confirm that mtDNA heteroplasmy levels frequently shift during cell culture, we have conducted two validation analyses: 1) tracking the fluctuation of clone-VAFs of an mtDNA mutation during longitudinal cell culture, and 2) profiling the mtDNA mosaicism without cell clonalization and seeing whether it is substantially different from the landscape obtained from the clones.

First, to investigate how culture conditions might influence VAF, serial clonalization was made from 47 lineages in vitro over approximately 10.5 weeks (**Extended Data Fig. 3a**), which made it possible to trace the VAF fluctuation in 49 mtDNA mutations. As shown in **Extended Data Figs. 3b and 3c** (shown below), although a few mutations showed some shifts as Reviewer #3 was concerned, we found that VAFs of most mtDNA mutations were correlating almost perfectly during cell culture ($R=0.984$). In line with this, few artifacts were detected during the culture, suggesting no apparent alteration in the overall

trend due to culture.

< New Extended Data Figs. 3a-3c >

Extended Data Figs. 3a-3c (3a modified; 3b,3c new). **a**, The experimental design for assessing the clone-VAF changes of mtDNA mutations during cell culture, as previously reported^{5,6}. Briefly, an isolated single cell (or crypt) was cultured for whole-genome sequencing of clones (mother clone). Subsequently, a second clonalization was conducted after isolating a single cell of the mother clone to assess the impact of culture on mtDNA variants (daughter clone). A total of 47 pairs of mother-daughter clones were obtained from ten mother clones. **b**, Linear correlation between

clone-VAFs of mother clones and median clone-VAFs of daughter clones. The Pearson's correlation coefficient and p-value from Pearson's correlation test are presented. The gray diagonal line represents a trendline indicating a one-to-one correspondence between the same x and y values. Different types of mother clones are color-coded. c, Comparison between clone-VAFs of mother and daughter clones of the same variant. The name of the mother clone and variants are provided above each panel. The clone-VAFs of the mother clone and daughter clone are indicated in different colors, and boxplots are also shown for the daughter clone.

Next, we explored mtDNA mosaicism in 484 patches obtained by laser-capture microdissection (LCM) of colon crypts (Lee-Six et al., Nature 2019). As only a few stem cells compete with each other over time, colon crypts are known to be mono-clonal or at least dominant clonal (Griffiths et al., Nature 1988; Kozar et al., Cell Stem Cell 2013). Moreover, LCM is free from the 'selective pressure' because it does not involve cell culture. From the effort, we confirmed that the landscapes of mtDNA heteroplasmies were similar in most aspects, including 1) the pattern of S_{VAF} increasing with age, 2) mutational signatures, and 3) the proportion of variants based on functional consequences (synonymous/missense/truncating). The VAF distribution was also similar between organoids and LCM samples, except for the absence of 100% VAF due to a small fraction of contaminating non-clonal cells in LCM samples.

< New **Extended Data Figs. 3d-3g** >

Extended Data Figs. 3d-3g. d-g. Comparing the results of clones obtained from single-crypt-derived organoids and patches obtained by laser capture microdissection (LCM) of colon crypts. LCM samples are obtained without cell culture. Results include linear correlation between age and the average S_{VAF} of the clones from the individual (**d**), the mutational spectrum in mtDNA (**e**), the proportion of mutations classified based on functional consequences (intergenic, synonymous, missense, and truncating; **f**), and clone-VAF distribution (**g**). S_{VAF} , the sum of the clone-VAFs of all detected sporadic mtDNA mutations in a clone.

To sum up, although substantial changes in mtDNA VAF may occur during the culture process, as Reviewer #3 indicated, 1) the event is not common, and 2) minor VAF fluctuations do not affect our observations and change this study's conclusions. We discussed the issue in **Supplementary Discussion 2** in the revised manuscript.

Original [#121-#123]

From the serial culture of the clones, we estimated that >93% of the mtDNA alterations were consistent over the duration of the cell culture (Extended Data Figs. 1f, 1g).

Revised [Results, page 5, Section “Landscape of mtDNA heteroplasmy in normal cells”]

The VAF of mtDNA alterations in each clone (clone-VAF) were predominantly (>93%) consistent throughout the cell culture, suggesting that our findings were not culture-associated artifacts (Extended Data Figs. 3a-3c; Supplementary Discussion 2).

Newly added [Discussion, page 15]

As mentioned above, we believe that flaws from culture-associated artifacts are minimal in this study (Extended Data Figs. 3a-3c; Supplementary Discussion 2). To further validate the robustness of the conclusion of this study, we explored mtDNAs from WGSs of micropatches collected by laser-capture microdissection (LCM)⁴³. Although the LCM technique does not always provide clonal resolution data, the mtDNA mutational profiles from LCM were mostly similar to those from our clones (Extended Data Figs. 3d-3g).

Comment 3-3. A further conceptual issue for me is the assumption that shared variants from an individual must result from such an early stage as egg fertilisation. If e.g. colonic crypts are being derived from a small area of the gut, the authors must also consider it likely that clonal competition and expansion over time is at least as likely a reason for shared variants as these being directly derived from the earliest stage of development. A similar comment could be made of the HSC data presented. Accepting this reality also weakens the foundation of the conclusions, which while still interesting, are not as far-reaching as the authors are suggesting at present.

Answer 3-3. We appreciate Reviewer #3 for the insightful comment and apologize for the confusion in the original manuscript. As mentioned by the reviewer, we constructed the phylogenetic tree using clones obtained from a single tissue in an individual. However, the phylogenetic trees we reconstructed represent that our clones were diverged from the earliest timing of our life (embryogenesis). These issues were discussed in previous publications (Coorens et al., Nature 2021; Park et al., Nature 2021; Nam et al., Nature 2023; Lee-Six et al., Nature 2018).

To validate the MRCA (most recent common ancestor) of our clones derived from the colorectal epithelium and fibroblast clones in this study, we employed clonal phylogenies again and tracked the lineage-defining mutations in the matched bulk blood tissues, which should have branched out in early embryogenesis. Of note, if the lineage-defining mutations were acquired early in embryogenesis before embryonic cell fate decision timing, these mutations should be found in the blood to substantial variant

allele fractions. Indeed, we confirmed that somatic mutations comprising the initial branches of all phylogenetic trees were also identified in the corresponding bulk blood samples (**Extended Data Fig. 2**). Furthermore, by merging the VAFs of mutations from Lineages 1 and 2, which are the earliest branches diverging from the MRCA cell, we observed that they were close to 50% in most individuals (**Fig. 1b**). This indicates that these two lineages collectively comprised the entire blood cells. Therefore, these findings suggest that the earliest two lineages in the trees were branched out before colon(or fibroblast)-blood differentiation in the embryogenesis, likely to be the first cell of our life or the fertilized eggs prior to gastrulation.

As Reviewer #3 suggested, we observed evidence of clonal competition at the late stage (during aging). In our phylogenies, these clones diverged in the very late stage, characterized by many more lineage-specific mutations, usually >100+. This distinct pattern allows us to distinguish them from embryonic lineages easily. We collapsed these variants, counting them as non-recurrent events in analyzing Het_{FE} variants and/or hotspot mutations.

As the issue is important for understanding our work, we have added descriptions for clone branching time in the revised manuscript in **Fig. 1b, Extended Data Fig. 2, and Supplementary Discussion 1** as shown below.

Newly added [Results, page 4-5, Section “Landscape of mtDNA heteroplasmy in normal cells”]

*The VAFs of lineage-defining variants in the matched bulk blood tissues suggest that the first branching in each of the phylogenies was early embryonic, likely to the first cell division in life, as reported previously³ (**Fig. 1b; Extended Data Fig. 2; Supplementary Discussion 1**).*

< New **Extended Data Fig. 2** >

Extended Data Fig. 2. The phylogenetic trees of 26 individuals colored by the VAF of lineage-defining early embryonic mutations (EEMs) in bulk blood tissues. Phylogenetic trees of each individual are reconstructed using somatic mutations in the nuclear genome, with branch lengths

proportional to the number of somatic mutations. After identifying the EEMs constituting the first two branches diverging from the MRCA (referred to as Lineage 1 and Lineage 2), their VAF in bulk blood was calculated. Then, the average VAF of each lineage was provided on the phylogenetic tree. In cases where branching from the MRCA resulted in more than two lineages, two with the highest VAFs in bulk blood were selected as Lineage 1 and Lineage 2. The sum of the average VAFs for Lineages 1 and 2 is shown in Fig. 1b. The phylogenetic trees display up to 30 EEMs, and each donor's name is provided at the top right of each panel. nDNA, nuclear DNA.

< New Fig. 1b >

b

Fig. 1b. Bar graphs showing the VAFs of early lineage-defining nDNA (nuclear DNA) mutations in bulk blood. Lineage-defining mutations refer to the somatic mutations in the nuclear genome that constitute the first two branches (designated as Lineage 1 and Lineage 2) branching from the MRCA (most recent common ancestor) in the phylogenetic tree. Cells carrying Lineage 1 mutations and Lineage 2 mutations are mutually exclusive, and the mutation VAF for each lineage is depicted in a cumulative bar graph. The top illustration explains the interpretation of this graph. If the VAF of mutations corresponding to each lineage in the bulk tissue totals 50%, it suggests that cells carrying Lineages 1 and 2 mutations comprise the entire cell population. This indicates that Lineages 1 and 2 represent lineages at the 2-cell stage. If the combined mutation VAF for Lineages 1 and 2 is 25%, it implies that Lineages 1 and 2 represent lineages at the 4-cell stage, as they constitute 50% of the entire cell population. Different color bars at the bottom represent

the original tissues from which each individual's clone was obtained.

Minor issues include:

Comment 3-4. The description of POLG as an error prone polymerase – POLG is the highest fidelity polymerase in humans.

Answer 3-4. We revised the sentences accordingly.

Original [#69-#72]

Conventionally, mitochondrial genomes are considered to have a higher mutation rate than nuclear genomes as their replication enzyme (polymerase gamma) is more prone to error^{26,30}, they lack a DNA repair system³¹ and protective histone proteins³², and their physical proximity to reactive oxygen species³³.

Revised [Introduction, page 3]

Conventionally, mitochondrial genomes are considered to have a higher mutation rate than nuclear genomes due to the error in their replication enzyme (polymerase gamma [POLG])^{26,31}, lack of a DNA repair system³² and protective histone proteins³³, and their physical proximity to reactive oxygen species³⁴.

Comment 3-5. The description of the mtDNA bottleneck occurring during embryo cleavage - this is part of the bottleneck, however the massive expansion of mtDNA copies from primordial germ cell to immature/mature oocyte represents a major part of the bottleneck process.

Answer 3-5. We appreciate the comment. Not to confuse our readers, we modified the terminology to 'early embryonic bottleneck' and revised the sentence to clarify this point.

Original [#230-#233]

Two possible scenarios include: 1) the mtDNA bottleneck during progressive mtDNA copy number reduction in the cleavage of early embryogenesis (referred to as early bottleneck)^{55,56}, and 2) the continuous mtDNA turnovers in each somatic lineage for a lifetime (referred to as lifetime drift; Fig. 3c)^{57,58}.

**Revised** [Results, page 8-9, Section "MtDNA turnover and drift in somatic lineages"]

*Two possible scenarios include: 1) early embryonic mtDNA bottleneck during progressive mtDNA copy number reduction in the cleavage of early embryogenesis^{53,54}, and 2) lifetime drift through the continuous mtDNA turnovers in each somatic lineage for a lifetime (**Fig. 3c**)^{55,56}.*

Comment 3-6. For clarity, despite the shortcomings I have illustrated and while many of the results presented here have been described elsewhere in other settings, I believe some of the data here are of interest and represent novel findings. However the authors need to robustly address the issues raised for the study to be convincing.

Answer 3-6. We appreciate the reviewer's constructive comments. As addressed above, we have tried to clarify the issues using experimental validation, in silico simulation, and in deeper depth description. We have revised our manuscript accordingly.

Decision Letter, first revision:

16th Apr 2024

Dear Professor Ju,

Please accept my apologies for the delay in returning this decision to you. Thank you for your patience.

Your Article, "Mitochondrial DNA mosaicism in human normal somatic cells" has now been seen by 3 referees. You will see from their comments below that while they find your work of interest, some important points are raised. We are interested in the possibility of publishing your study in Nature Genetics, but would like to consider your response to these concerns in the form of a revised

manuscript before we make a final decision on publication.

We therefore invite you to revise your manuscript taking into account all reviewer and editor comments. Please highlight all changes in the manuscript text file. At this stage we will need you to upload a copy of the manuscript in MS Word .docx or similar editable format.

Our plan is to assess the revisions in house. However, depending on your response, we might have to return to one or more reviewers. Please be assured that we are keen to avoid further delays and will only do this if we deem it to be absolutely necessary.

*2) If you have not done so already please begin to revise your manuscript so that it conforms to our Article format instructions, available here.

*3) Include a revised version of any required Reporting Summary:

Please be aware of our guidelines on digital image standards.

[redacted]

We hope to receive your revised manuscript within four to eight weeks. If you cannot send it within this time, please let us know.

Nature Genetics is committed to improving transparency in authorship. As part of our efforts in this

direction, we are now requesting that all authors identified as 'corresponding author' on published papers create and link their Open Researcher and Contributor Identifier (ORCID) with their account on the Manuscript Tracking System (MTS), prior to acceptance. ORCID helps the scientific community achieve unambiguous attribution of all scholarly contributions. You can create and link your ORCID from the home page of the MTS by clicking on 'Modify my Springer Nature account'. For more information please visit www.springernature.com/orcid.

I also look forward to meeting you in Edinburgh next week!

Sincerely,

Safia Danovi, PhD
Senior Editor, Nature Genetics
ORCID: 0009-0007-7822-5479

Reviewers' Comments:

Reviewer #1:

Remarks to the Author:

I thank the authors for their careful revisions which fully address my questions. I find the addition of the homeostatic model very helpful, as the paper now presents a more comprehensive theoretical basis to aid with the interpretation of the data. I also find the manuscript much easier to read now. Many thanks.

Reviewer #2:

Remarks to the Author:

Thank you to the authors for a very thorough and detailed revision. In my view the paper is now substantially more clear and rigorous, and the additional text clarifies many of the new concepts and methods that the authors have introduced to reach their conclusions. My major concerns have been addressed and I have only one minor suggestion remaining.

1. Some of the terminology the authors employ to describe mutations is potentially problematic. For example, the use of sporadic vs. hotspot mutations in the context of post-zygotic acquired mutations. In the genetics literature, sporadic is often employed to describe non-inherited (e.g. post-zygotic) mutations, just as the authors have employed it. However, the phrase "hotspot" refers not to the timing of the development of the mutation, but rather whether it is recurrent across a population. The authors are free to do with this comment what they will, but I would suggest retaining the phrase "post-zygotic", dropping the phrase "sporadic", and replacing "sporadic" with something like "non-recurrent."

Reviewer #3:

Remarks to the Author:

3-1) the modifications made by the authors to the manuscript have improved the flow, although the paper is still a challenging read in places.

3-2) I appreciate the authors' response. The key issue is of selection for fitness at the initial stage of biological material → successful adaptation to culture is the majority of my concern, as the authors did indeed show that the majority of the variants they cultured in the various systems are not subject to shifts in heteroplasmy. Are sequencing data of the original biological materials available, to ensure that the diversity of the original material is represented in the cultured material? The concern here is that the authors are only studying a minor subset of the total variants (the survivors), which has potential to confound the conclusions.

3-3) I appreciate the authors' response to my comment. However, it remains unclear whether this occurs in the 1 cell stage embryo as the authors assert, or later. As it is impossible to determine the answer to this with the data in hand, I would suggest the authors moderate their language to be less certain. Prior to endo/meso/ectoderm differentiation, a substantial proportion of the pre-implantation embryo does not go on to give rise to the foetus (placenta, yolk sac etc). It is entirely plausible that variants are selected or arise de novo between fertilisation and epiblast differentiation, and the novelty and interest of the authors findings do not rest on this distinction.

3-4) the modifications made to this sentence are still not correct. The authors should refer to Lynch M. The lower bound to the evolution of mutation rates. *Genome Biol Evol.* 2011;3:1107–18 and modify accordingly. The sentence still reads as saying that POLG is inherently error-prone and this is not correct. More broadly, this sentence asserts several largely debunked theories on the origin of high mtDNA mutation rate (e.g. ROS, lack of packaging proteins), and it should be moderated.

3-5) This change of terminology is good.

Author Rebuttal, first revision:

Reviewers' Comments:

Reviewer #1:

Comment 1-1. I thank the authors for their careful revisions which fully address my questions. I find the addition of the homeostatic model very helpful, as the paper now presents a more comprehensive theoretical basis to aid with the interpretation of the data. I also find the manuscript much easier to read now. Many thanks.

Answer 1-1. We are glad that Reviewer #1 is satisfied with our revised manuscript. We are grateful to the reviewer for the constructive comments that made our manuscript more meaningful.

Reviewer #2:

Comment 2-1. Thank you to the authors for a very thorough and detailed revision. In my view the paper is now substantially more clear and rigorous, and the additional text clarifies many of the new concepts and methods that the authors have introduced to reach their conclusions. My major concerns have been addressed and I have only one minor suggestion remaining.

Answer 2-1. We are glad that Reviewer #2 is satisfied with our revised manuscript. We deeply appreciate the reviewer's insightful comments.

Comment 2-2. Some of the terminology the authors employ to describe mutations is potentially problematic. For example, the use of sporadic vs. hotspot mutations in the context of post-zygotic acquired mutations. In the genetics literature, sporadic is often employed to describe non-inherited (e.g. post-zygotic) mutations, just as the authors have employed it. However, the phrase "hotspot" refers not to the timing of the development of the mutation, but rather whether it is recurrent across a population. The authors are free to do with this comment what they will, but I would suggest retaining the phrase "post-zygotic", dropping the phrase "sporadic", and replacing "sporadic" with something like "non-recurrent."

Answer 2-2. We appreciate the reviewer for the comment. To abide by the suggestion of Reviewer #2 and clarify the manuscript, we revised the terms: we now use post-zygotic simple (PZ_{simple}) and post-zygotic recurrent ($PZ_{\text{recurrent}}$) mutations to describe the phrase sporadic and (post-zygotic) hotspot mutations in the previous manuscript. We have updated the manuscript and figures accordingly.

Original [#166-#169]

In contrast, the vast majority of post-zygotic mutations were sporadic (singletons [$n = 5,276$] or coincidentally recurrent [$n = 376$]). Some post-zygotic mutations, however, were recurrently found

in multiple clones established from multiple donors, indicating hotspot mutations ($n = 390$; Figs. 2a, 2b).

→

Revised [Results, page 6, Section “Two origins of mtDNA alterations”]

In contrast, the vast majority of post-zygotic mutations were confined to a clone or two ($n = 5,652$; either as singletons [$n = 5,276$] or coincidentally recurrent mutations [$n = 376$]; referred to as post-zygotic simple [PZ_{simple}] mutations hereafter). However, some post-zygotic mutations ($n = 390$) occurred frequently in 32 sites across multiple clones established from multiple donors, suggesting a higher mutation rate in the sites compared to the background (referred to as post-zygotic recurrent [$PZ_{recurrent}$] mutations; Figs. 2a, 2b).

< Revised Figs. 2a, 2b >

Figs. 2a, 2b. a, A simple diagram showing how the origins of the identified mosaic mtDNA variants were categorized. **b**, Schematic diagram illustrating different origins of shared mtDNA variants. Different shared mtDNA variants are color-coded.

Reviewer #3:

Comment 3-1. The modifications made by the authors to the manuscript have improved the flow, although the paper is still a challenging read in places.

Answer 3-1. We appreciate the comment. We apologize for difficulties in understanding (if any), as our manuscript describes a broad field of mtDNA somatic mutations in normal cells.

Comment 3-2. I appreciate the authors' response. The key issue is of selection for fitness at the initial stage of biological material → successful adaptation to culture is the majority of my concern, as the authors did indeed show that the majority of the variants they cultured in the various systems are not subject to shifts in heteroplasmy. Are sequencing data of the original biological materials available, to ensure that the diversity of the original material is represented in the cultured material? The concern here is that the authors are only studying a minor subset of the total variants (the survivors), which has potential to confound the conclusions.

Answer 3-2. We appreciate the reviewer's careful comment. Unfortunately, the original biological materials are unavailable at the moment. In addition, even in the case of their availability, we carefully expect that sensitive detection of mtDNA variants in the bulk original materials would not be technically feasible given the inter-cellular and inter-mitochondrial heterogeneity (as described in the revised manuscript, #75-#78 and #86-#92 in Introduction). Therefore, proper comparisons between the original and cultured materials are mostly impossible.

We agree that our study only captured a subset of mtDNA mutations in original bulk tissues due to the limited number of cells explored within each tissue sample. It is important to recognize that detecting all mutations is only feasible if every cell within a tissue sample is sequenced. It is beyond the relevance, as increasing number of sequenced clones would inevitably reveal mutations across all mtDNA loci. Nevertheless, our mutation set remains unbiased by selection pressures or culture-mediated artifacts, as we showed in the previous revision. As such, our downstream conclusions, including insights into mtDNA heteroplasmy in fertilized eggs, mtDNA turnovers in somatic lineages, and the absolute mtDNA mutation rate, retain significance.

We would like to reiterate that we have already presented evidence representing only minimal bias in our system (Detailed description in **Supplementary Discussion 2**). First, when comparing clones obtained by longitudinal clonalization, we found that VAFs of most mtDNA mutations were maintained during cell culture (**Extended Data Figs. 3a-3c**). It suggests that even through the cell culture, clones can faithfully represent the mutational profiles of the original cells. Second, clone-like patches obtained without culture showed a similar mtDNA heteroplasmy landscape with cultured clones. We obtained patches of colon crypts by laser-capture microdissection (LCM) and explored their mtDNA mosaicism. Then, we confirmed that most patterns of mtDNA heteroplasmy were similar, including 1) the pattern of S_{VAF} increasing with age, 2) mutational signatures, 3) the proportion of variants based on functional consequences (synonymous/missense/truncating), and 4) VAF distributions (**Extended Data Figs.**

3d-3g).

Comment 3-3. I appreciate the authors' response to my comment. However, it remains unclear whether this occurs in the 1 cell stage embryo as the authors assert, or later. As it is impossible to determine the answer to this with the data in hand, I would suggest the authors moderate their language to be less certain. Prior to endo/meso/ectoderm differentiation, a substantial proportion of the pre-implantation embryo does not go on to give rise to the foetus (placenta, yolk sac etc). It is entirely plausible that variants are selected or arise de novo between fertilisation and epiblast differentiation, and the novelty and interest of the authors findings do not rest on this distinction.

Answer 3-3. We agree with the reviewer's comment. The Het_{FE} variants we identified were indeed present before gastrulation, but it is difficult to assert with certainty that the first node of the phylogenetic tree is the fertilized egg, as the reviewer mentioned. Therefore, we have moderated our language accordingly.

Original [#124-#127]

The VAFs of lineage-defining variants in the matched bulk blood tissues suggest that the first branching in each of the phylogenies was early embryonic, likely to the first cell division in life, as reported previously³ (Fig. 1b; Extended Data Fig. 2; Supplementary Discussion 1).

→

Revised [Results, page 4, Section "Landscape of mtDNA heteroplasmy in normal cells"]

The VAFs of lineage-defining variants in the matched bulk blood tissues suggest that the first branching in each of the phylogenies was early embryonic, equivalent and/or close to the first cell division in life, as reported previously³ (Fig. 1b; Extended Data Fig. 2; Supplementary Discussion 1).

Original [#178-#180]

As mentioned above (Fig. 1b), the first nodes in the phylogenies were likely the first founder cell of the donors, our observation strongly suggested that the fertilized eggs carried the heteroplasmic variants.

→

Revised [Results, page 6, Section “MtDNA heteroplasmy in the fertilized egg”]

As mentioned above (Fig. 1b), the first node is the most recent common ancestor (MRCA) cell of all/most somatic cells of a donor. Our observation strongly suggested that the MRCA cell, potentially the fertilized egg (if not, a pre-gastrulation stage early embryonic cell), carried the heteroplasmic variants.

Comment 3-4. The modifications made to this sentence are still not correct. The authors should refer to Lynch M. The lower bound to the evolution of mutation rates. *Genome Biol Evol.* 2011;3:1107–18 and modify accordingly. The sentence still reads as saying that POLG is inherently error-prone and this is not correct. More broadly, this sentence asserts several largely debunked theories on the origin of high mtDNA mutation rate (e.g. ROS, lack of packaging proteins), and it should be moderated.

Answer 3-4. We appreciate the comment. Not to mislead the readers, we revised the sentence accordingly.

Original [#70-#73]

Conventionally, mitochondrial genomes are considered to have a higher mutation rate than nuclear genomes due to the error in their replication enzyme (polymerase gamma [POLG])^{26,31}, lack of a DNA repair system³² and protective histone proteins³³, and their physical proximity to reactive oxygen species³⁴.

→

Revised [Introduction, page 3]

Conventionally, mitochondrial genomes are considered to have a higher mutation rate than nuclear genomes³¹.

Comment 3-5. This change of terminology is good.

Answer 3-5. We appreciate the comment.

Decision Letter, second revision:

1st May 2024

Dear Dr Ju,

It was a pleasure to meet you and your team in Edinburgh last week.

Thank you for submitting your revised manuscript "Mitochondrial DNA mosaicism in human normal somatic cells" (NG-A63896R2). The Nature Genetics editorial team assessed your revisions in house and I'm delighted to say that we'll be happy in principle to publish it in Nature Genetics, pending minor revisions to satisfy our editorial and formatting guidelines.

Sincerely,

Safia Danovi, PhD
Senior Editor, Nature Genetics
ORCID: 0009-0007-7822-5479

Author Rebuttal, first revision:**Reviewers' Comments:****Reviewer #3:**

Comment 3-2. I appreciate the authors' response. The key issue is of selection for fitness at the initial stage of biological material → successful adaptation to culture is the majority of my concern, as the authors did indeed show that the majority of the variants they cultured in the various systems are not subject to shifts in heteroplasmy. Are sequencing data of the original biological materials available, to ensure that the diversity of the original material is represented in the cultured material?

The concern here is that the authors are only studying a minor subset of the total variants (the survivors), which has potential to confound the conclusions.

Answer 3-2. We appreciate the reviewer's careful comment. While we understand the question raised by the reviewer, providing direct evidence for the absence of selection bias is quite challenging due to the nature of single-cell resolution genome sequencing; the original founder cells are no longer available. We believe that adjacent bulk tissues are not appropriate original material since they do not contain the specific single cell with the mtDNA post-zygotic mutation identified through genome sequencing. Additionally, as the reviewer may agree, detecting mtDNA mutations confined to a single cell from bulk-tissue DNA sequencing is technically challenging.

However, we believe we have presented some indirect evidence:

The dN/dS ratio analysis (**Fig. 5a**) suggests no substantial positive or negative selection. If specific mutations were advantageous or disadvantageous during cell culture, the ratio would be higher or lower than 1, respectively.

Laser capture microdissection of colon crypts followed by whole-genome sequencing revealed an mtDNA mutational landscape highly similar to the clones, indicating minimal culture-associated bias (**Extended Data Figs. 3d-g**). Especially, similar mtDNA mutation counts, the proportion of missense/truncating mutations, and the absence of recurrent mutations strongly support that our mutation set remains unbiased by selection pressures. Further details can be found in **Supplementary Note 2**.

Final Decision Letter:

In reply please quote: NG-A63896R3 Ju

21st Jun 2024

Dear Dr Ju,

I am delighted to say that your manuscript "Mitochondrial DNA mosaicism in normal human somatic cells" has been accepted for publication in an upcoming issue of Nature Genetics.

Your paper will be published online after we receive your corrections and will appear in print in the next available issue. You can find out your date of online publication by contacting the Nature Press Office (press@nature.com) after sending your e-proof corrections.

Please note that *Nature Genetics* is a Transformative Journal (TJ). Authors may publish their research with us through the traditional subscription access route or make their paper immediately open access through payment of an article-processing charge (APC). Authors will not be required to make a final decision about access to their article until it has been accepted. Find out more about Transformative Journals

Authors may need to take specific actions to achieve compliance with funder and institutional open access mandates. If your research is supported by a funder that requires immediate open access (e.g. according to Plan S principles) then you should select the gold OA route, and we will direct you to the compliant route where possible. For authors selecting the subscription publication route, the journal's standard licensing terms will need to be accepted, including [a href="https://www.nature.com/nature-portfolio/editorial-policies/self-archiving-and-license-to-publish"](https://www.nature.com/nature-portfolio/editorial-policies/self-archiving-and-license-to-publish). Those licensing terms will supersede any other terms that the author or any third party may

assert apply to any version of the manuscript.

If you have not already done so, we strongly recommend that you upload the step-by-step protocols used in this manuscript to protocols.io. protocols.io is an open online resource that allows researchers to share their detailed experimental know-how. All uploaded protocols are made freely available and are assigned DOIs for ease of citation. Protocols can be linked to any publications in which they are used and will be linked to from your article. You can also establish a dedicated workspace to collect all your lab Protocols. By uploading your Protocols to protocols.io, you are enabling researchers to more readily reproduce or adapt the methodology you use, as well as increasing the visibility of your protocols and papers. Upload your Protocols at <https://protocols.io>. Further information can be found at <https://www.protocols.io/help/publish-articles>.

Sincerely,

Safia Danovi, PhD
Senior Editor, Nature Genetics
ORCID: 0009-0007-7822-5479